# A Theoretical Analysis of In-context Task Retrieval and Learning

## Abstract

In-context learning (ICL) can be used for two different purposes: task retrieval and task learning. Task retrieval focuses on recalling a pre-trained task using examples from the task that closely approximates the target pre-trained task, while task learning involves learning a task using in-context examples. To rigorously analyze these two modes, we propose generative models for both pretraining data and in-context samples. Assuming we use our proposed models and consider the mean squared error as a risk measure, we demonstrate that in-context prediction using a Bayes-optimal next-token predictor equates to the posterior mean of the label, conditioned on in-context samples. From this equivalence, we derive risk upper bounds for in-context learning. We reveal a unique phenomenon in task retrieval: as the number of in-context samples increases, the risk upper bound decreases initially and then increases subsequently. This implies that more in-context examples could potentially worsen task retrieval. We validate our analysis with numerical computations in various scenarios and validate that our findings are replicable in the actual Transformer model implementation.

## 1 Introduction

Large language models (LLMs) exhibit a significant improvement in predictive performance when provided with in-context samples (Brown et al., 2020; Zhao et al., 2021; Liu et al., 2022a; Lyu et al., 2023). To explain such phenomenon, namely *in-context learning (ICL)*, researchers have introduced both theoretical and empirical analyses drawing on concepts like Bayesian inference and gradient descent (Olsson et al., 2022; Min et al., 2022; Yoo et al., 2022; Han et al., 2023b). Recently, two distinct modes of ICL have been proposed by Pan et al. (2023). Slightly different from Pan et al. (2023), we define two modes of ICL as follows. The first mode aims to *learns* a new task, specifically a mapping function from $x$ to $y$, which is termed as *in-context task learning* (Xie et al., 2022; von Oswald et al., 2022; Zhang et al., 2023b). The second mode aims to *retrieves* the most relevant task learned during pretraining, referred to as *in-context task retrieval* (Kojima et al., 2022; Kim et al., 2022; Liu et al., 2022a; Rubin et al., 2022; Liu et al., 2022b; Lyu et al., 2023).

Current analyses often neglect the two modes, lacking a comprehensive explanation. Garg et al. (2022); Akyürek et al. (2023); von Oswald et al. (2022); Dai et al. (2023) suggest Transformers inherently use gradient descent with in-context samples, implying task learning with in-context samples rather than task retrieval. Xie et al. (2022); Raventos et al. (2023); Wang et al. (2023) link ICL with Bayesian inference, suggesting it learns arbitrary concepts with continuous concept space and retrieves the closest concept with discrete concept space, failing to explain both retrieval and learning within the same prior distribution. Meanwhile, the majority of these theoretical explanations overlook the impact of pre-training distributions. Conversely, several empirical studies explore the influence of pre-training distribution on ICL. Chan et al. (2022) offer an empirical analysis of how the properties of pre-training distributions affect ICL. Raventos et al. (2023) empirically examine the effect of pre-training distribution diversity on ICL.

In our study, we propose data generative models for pretraining data, leveraging the properties of the pre-training distribution to explain task retrieval and learning. Extending Raventos et al. (2023), we assume that the pretraining data is drawn from a mixture of noisy task distributions, each centered around a *clean task*. We further give definitions to task learning and retrieval: (i) task learning aims to learn a task using in-context examples, and (ii) task retrieval aims to recall a pre-trained *clean task* with in-context examples from the task that approximates the target task. We assume the model

is pretrained with mean squared error and show inferencing a Bayes-optimal next-token predictor is equivalent to the posterior mean of the label. We further prove that the upper bound for task learning shows a quadratically decaying rate as $k$ increases, while the bound for task retrieval follows a U-shaped curve. This suggests that task retrieval performance may be optimal with a specific number of in-context examples, rather than with an infinite number of samples. Beyond theoretical bounds, we perform numerical computations to validate our analyses and conduct experiments to show our findings are replicable in actual Transformer model implementations.

We outline our contributions as follows:

- We introduce the generative models for both pretraining data and in-context samples to explain the two modes of ICL: task retrieval and task learning.

- We show that using a pre-trained Bayes-optimal next-token predictor is equivalent to the posterior mean of the label, conditioned on in-context samples. We then derive upper bounds for task retrieval and learning risks. We show that the risk bound of task learning follows a quadratic decreasing pattern and uncover a U-shaped pattern for the risk bound of task retrieval. These findings are further illustrated through numerical computations alongside mathematical derivation.

- We finally conduct experiments to demonstrate that our findings are not just supported by numerical calculations, but can also be reproduced in a real-world Transformer setup.

We put the comprehensive notation table to Appendix B.

## 2 RELATED WORK

**Two Modes of ICL** A significant body of research (Akyürek et al., 2023; von Oswald et al., 2022; Garg et al., 2022; Li et al., 2023a; Zhang et al., 2023b) delves into the mechanisms behind the success of ICL in learning the task from in-context samples. Yet, they often overlook the two distinct scenarios of ICL: task retrieval and task learning, recently introduced by Pan et al. (2023). In contrast to existing literature, our research delves into both these task categories.

**Explaining ICL via Bayesian Inference** Xie et al. (2022) first analyze ICL via the perspective of Bayesian inference, assuming that the next token is generated via Hidden Markov Model (HMM) (Ghahramani & Jordan, 1995; Rabiner, 1989). Wang et al. (2023) further stand on the Bayesian lens and proposes an in-context sample selection algorithm to improve the few-shot performance. These works explain that the pretrained LLM can learn the exact task of the in-context samples when the number of in-context samples is large enough. However, Xie et al. (2022) and Wang et al. (2023) ignore the effect of the pretraining concept/task distribution on the ICL phenomenon and failed to explain the two modes of ICL in the same setting. Differently, we explain two modes of ICL in the same setting with the help of further assumptions on the pretraining data distribution.

**Explaining ICL via Gradient Descent** Beyond Bayesian Inference, Garg et al. (2022); Akyürek et al. (2023); von Oswald et al. (2022); Dai et al. (2023) understand ICL from the perspective of "gradient descent". Garg et al. (2022) hinted at the possibility that under ICL, the Transformer might be implicitly executing gradient descent. This notion was subsequently expanded by several works (Garg et al., 2022; Akyürek et al., 2023; von Oswald et al., 2022; Dai et al., 2023), which provided a more detailed examination of the topic. Lastly, Mahankali et al. (2023); Ahn et al. (2023); Zhang et al. (2023a) indicate that the global minimizer of the pretraining loss implements gradient descent.

**Explaining ICL (Others)** In addition to Bayesian inference and gradient descent, Han et al. (2023a) explain ICL as kernel regression with the attention mechanism. Some recent works study ICL via the lens of the algorithm. Giannou et al. (2023) show Transformer can be constructed to emulate in-context learning algorithms, such as SGD. Li et al. (2023b) study the aspects of generalization and stability. Bai et al. (2023) show transformers can perform in-context algorithm selection, *i.e.*, adaptively selecting different ICL algorithms such as gradient descent, least square, or ridge regression.

**Analyzing ICL via Pretraining Dataset Distribution** Researchers also study the property of the pretraining dataset of language models (Chan et al., 2022; Raventos et al., 2023). Chan et al. (2022) identifies three properties of natural language datasets: (i) burstiness, (ii) label multiplicity, and (iii) a long-tailed class distribution. Raventos et al. (2023) show as the diversity of the pretraining dataset increases, the Transformer model transitions from doing task retrieval to learning the underlying task structure and becoming capable of doing task learning.

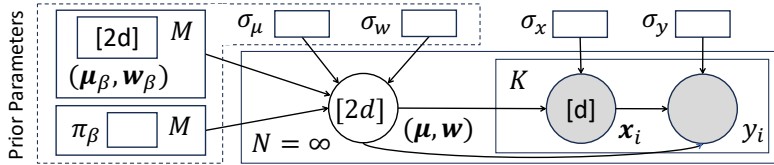

Figure 1: The probabilistic graphical model (PGM) for pretraining data generation process, assuming a particular Gaussian mixture model. (See Sec. 3.3 for more details.) The task $(\boldsymbol{\mu}, \boldsymbol{w})$ is drawn $N$ times from the task prior, then for each time, $K$ labeled samples are drawn from the chosen task.

## 3 PRETRAINING: DATA GENERATIVE MODELS AND OPTIMAL MODEL

Consider a next-token prediction model $\mathcal{F}$, pretrained on a dataset to produce $\hat{\mathcal{F}}$. The next-token prediction model can take $2k + 1$ elements (odd-numbered elements with dimension $d$ and even-numbered elements with dimension 1) for $k \geq 0$, where the first $2k$ elements are $k$ labeled samples and the last element is an unlabeled test sample. The goal of the next-token prediction model is to predict the $(2k + 2)^{\text{th}}$ token, which corresponds to the label of the last element. This prediction is possible if the model can discern hidden patterns within the "in-context" $k$ samples.

### 3.1 GENERATIVE MODELS FOR PRETRAINING DATA

In the pretraining phase, we assume the next-token prediction model is pretrained on diverse tasks, each representing a joint (continuous) distribution of $(\boldsymbol{x}, y)$. Each task is defined by vectors $\boldsymbol{\mu}$ (for the $\boldsymbol{x}$ distribution) and $\boldsymbol{w}$ (for the conditional distribution $y \mid \boldsymbol{x}$), together forming the joint distribution of $(\boldsymbol{x}, y)$. Unlike our model, Raventos et al. (2023) assumed identical $\boldsymbol{x}$ distributions across tasks (same $\boldsymbol{\mu}$, different $\boldsymbol{w}$). The pretraining data consists of a large number of sequences, and each sequence consists of $K$ labeled samples drawn from a certain task. Our generative model can be formally described in Assumption 1 and Fig. 1.

**Assumption 1** (Pretraining Data Generative Model). *For a given integer $K > 0$, a task* prior *$\mathcal{D}^{prior} = \mathcal{D}_{\boldsymbol{\mu}, \boldsymbol{w}}$, and a joint conditioned sampler $\mathcal{D}_{\boldsymbol{x}, y}(\boldsymbol{\mu}, \boldsymbol{w})$, we generate a sequence $\mathcal{S}_K$ as follows:*
*(a) Sample a task: $(\boldsymbol{\mu}, \boldsymbol{w}) \sim \mathcal{D}^{prior}$;*
*(b) Sample K labeled examples: $\forall i \in \{1, 2, \ldots, K\}$, $(\boldsymbol{x}_i, y_i) \sim \mathcal{D}_{\boldsymbol{x}, y}(\boldsymbol{\mu}, \boldsymbol{w})$.*
*(c) Define a sequence: $\mathcal{S}_K = [\boldsymbol{x}_1, y_1, \ldots, \boldsymbol{x}_K, y_K]$.*

The sequence of the first $2k$ elements of $\mathcal{S}_K$ will be denoted by $\mathcal{S}_k$, and the sequence of the first $2k + 1$ elements will be indicated by $\mathcal{S}_k \oplus \boldsymbol{x}_{k+1}$, e.g., $\mathcal{S}_0 = [\,]$, and $\mathcal{S}_1 \oplus \boldsymbol{x}_2 = [\boldsymbol{x}_1, y_1, \boldsymbol{x}_2]$.

### 3.2 PRETRAINING OBJECTIVE AND BAYES-OPTIMAL NEXT-TOKEN PREDICTOR

We consider the following pretraining objective: $\mathcal{L}(\mathcal{F}) = \mathbb{E}_{\mathcal{S}_K} \left[ \frac{1}{K} \sum_{k=0}^{K-1} (\mathcal{F}(\mathcal{S}_k \oplus \boldsymbol{x}_{k+1}) - y_{k+1})^2 \right]$. In other words, for each sequence, we pretrain $\mathcal{F}$ to predict each intermediate label based on preceding samples, measuring risk with the squared loss. Due to the linearity of expectation, we have: $\mathcal{L}(\mathcal{F}) = \frac{1}{K} \sum_{k=0}^{K-1} \mathbb{E}_{\mathcal{S}_K} \left[ (\mathcal{F}(\mathcal{S}_k \oplus \boldsymbol{x}_{k+1}) - y_{k+1})^2 \right]$. A highly expressive $\mathcal{F}$ can be viewed as $K$ separate models $\mathcal{F}_0, \ldots, \mathcal{F}_{K-1}$, where $\mathcal{F}_k$ takes exactly $2k + 1$ tokens as input. Thus, pretraining can be decomposed into $K$ separate optimization problems:

$$\mathcal{F}_k^* = \underset{\mathcal{F}_k}{\arg\min} \, \mathbb{E}_{\mathcal{S}_K} [(\mathcal{F}_k(\mathcal{S}_k \oplus \boldsymbol{x}_{k+1}) - y_{k+1})^2], \, \forall k \in \{0, \ldots, K-1\}.$$

The solution denoted $\mathcal{F}_k^*$ is an MMSE estimator (Van Trees, 2004, page 63) to each $k$, and thus the prediction $\mathcal{F}^*(\mathcal{S}_k \oplus \boldsymbol{x}_{k+1}) = \mathcal{F}_k^*(\mathcal{S}_k \oplus \boldsymbol{x}_{k+1})$ satisfies:

$$\mathcal{F}^*(\mathcal{S}_k \oplus \boldsymbol{x}_{k+1}) = \mathbb{E}_{\mathcal{S}_K} [y_{k+1} \mid \mathcal{S}_k \oplus \boldsymbol{x}_{k+1}] = \mathbb{E}_{\boldsymbol{\mu}, \boldsymbol{w}} \left[ \mathbb{E}_{y_{k+1}} [y_{k+1} \mid \boldsymbol{w}, \boldsymbol{x}_{k+1}] \Big| \mathcal{S}_k \oplus \boldsymbol{x}_{k+1} \right]. \quad (1)$$

$\mathcal{F}^*(\mathcal{S}_k \oplus \boldsymbol{x}_{k+1})$ is the expectation of $\mathbb{E}_{y_{k+1}} [y_{k+1} \mid \boldsymbol{w}, \boldsymbol{x}_{k+1}]$ on task posterior observing $\mathcal{S}_k \oplus \boldsymbol{x}_{k+1}$.

### 3.3 GAUSSIAN/LINEAR ASSUMPTIONS ON DATA GENERATIVE MODEL

Let us now elaborate on assumptions extending the generation process 1 for a tractable posterior and insights into ICL behaviors. Assumption 2 defines the task prior distribution of $(\boldsymbol{\mu}, \boldsymbol{w})$ using a Gaussian mixture and the noisy linear prediction task. Assumption 3 provides further clarifications for ICL phenomena. Assumption 4 specifies the distribution of test-time in-context samples.

**Assumption 2** (Gaussian/Linear Model for Pretraining Data).
*(a)* $(\boldsymbol{\mu}, \boldsymbol{w}) \sim \mathcal{D}^{prior} : P(\boldsymbol{\mu}, \boldsymbol{w}) = \sum_{\beta=1}^{M} \pi_\beta T_\beta = \sum_{\beta=1}^{M} \pi_\beta \mathcal{N}(\boldsymbol{\mu} \mid \boldsymbol{\mu}_\beta, \sigma_\mu^2 \boldsymbol{I}) \mathcal{N}(\boldsymbol{w} \mid \boldsymbol{w}_\beta, \sigma_w^2 \boldsymbol{I})$,
*where $\pi_\beta$ is the mixture weight of the $\beta^{th}$ mixture component $T_\beta$, $0 < \pi_\beta < 1$, $\sum_{\beta=1}^{M} \pi_\beta = 1$,
$(\boldsymbol{\mu}_\beta, \boldsymbol{w}_\beta)$ is the center of the $\beta^{th}$ mixture component, and all components share the same covariance
matrix controlled by $\sigma_\mu^2$ and $\sigma_w^2$;*
*(b) $\boldsymbol{x} \sim \mathcal{D}_{\boldsymbol{x}}(\boldsymbol{\mu}) : P(\boldsymbol{x} \mid \boldsymbol{\mu}) = \mathcal{N}(\boldsymbol{x} \mid \boldsymbol{\mu}, \sigma_x^2 \boldsymbol{I})$;*
*(c) $y \mid \boldsymbol{x} \sim \mathcal{D}_{y|\boldsymbol{x}}(\boldsymbol{w}) : P(y \mid \boldsymbol{x}, \boldsymbol{w}) = \mathcal{N}(y \mid \boldsymbol{w}^\top \boldsymbol{x}, \sigma_y^2)$;*
*(d) $\boldsymbol{x}, \boldsymbol{\mu}, \boldsymbol{\mu}_\beta, \boldsymbol{w}, \boldsymbol{w}_\beta \in \mathbb{R}^d, \boldsymbol{I} \in \mathbb{R}^{d \times d}$.*

We outline the rationale behind our modeling for pretraining task distribution. Assumption 2(a) implies training the next-token prediction model on $M$ components, reflecting real-world LLMs trained across diverse tasks. The center of the mixture component signifies a primary **clean** task, with variance indicating interpretative deviations, *i.e.*, different data sources such as different labelers interpret a clean task differently, resulting in generating data based on "noisy" versions of the clean task, namely **noisy** tasks. This noise is modeled by a Gaussian distribution, creating a Gaussian mixture task prior. Assumptions 2(b) and 2(c) dictate that interpreters generate $(\boldsymbol{x}, y)$ pairs from these noisy tasks. Specifically, Assumption 2(b) posits $\boldsymbol{x}$ distribution of each task as a multivariate Gaussian, sharing common covariance across tasks. Assumption 2(c) treats tasks as noisy linear regressions with shared Gaussian noise in labels.

**Assumption 3.** *Further assumptions on the pretraining distribution:*
*(a) $\forall \beta, \|\boldsymbol{\mu}_\beta\| = \|\boldsymbol{w}_\beta\| = 1$; (b) $\exists r > 1$ that $\forall \alpha \neq \beta, \frac{1}{r} \leq \frac{\pi_\alpha}{\pi_\beta} \leq r$; (c) $\sigma_\mu \ll \sigma_x, \sigma_w \ll \sigma_y$.*

Assumption 3(a) simplifies subsequent analysis. Assumption 3(b) posits comparable probabilities $\pi$ across different mixture components. Assumption 3(c) states the variances of $\boldsymbol{\mu}$ and $\boldsymbol{w}$, the task noises, are notably less than sample noises of $\boldsymbol{x}$ and $y$, implying high-quality training data.

**Assumption 4** (Gaussian/Linear Model for Demonstration Data at Test Time). *At test time:*
*(a) $\boldsymbol{x}_i \sim \mathcal{N}(\boldsymbol{\mu}^*, \tau_x^2 \boldsymbol{I})$, $y_i = \langle \boldsymbol{x}_i, \boldsymbol{w}^* \rangle$, $\forall i$; (b) $\|\boldsymbol{\mu}^*\| = \|\boldsymbol{w}^*\| = 1$.*

The Assumption 4(a) says that in-context sample $(\boldsymbol{x}, y)$ follows task $(\boldsymbol{\mu}^*, \boldsymbol{w}^*)$, and there is no noise in the labels. The Assumption 4(b) is used to simplify the subsequent analysis.

## 4 ANALYSIS OF THE POSTERIOR DISTRIBUTION

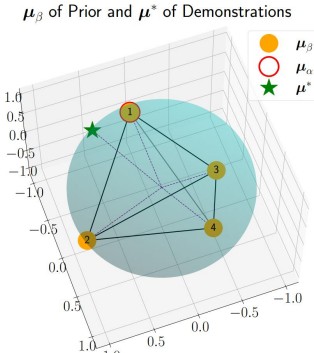

We have shown performing ICL with the optimally pretrained model is equivalent to computing the posterior mean in Sec.3.2. With assumptions in Sec. 3.3, we are able to derive a closed-form expression for the posterior $\mathcal{D}^{post}$ given $\mathcal{S}_k \oplus \boldsymbol{x}_{k+1}$ in Sec. 4.1. The posterior will be utilized to simplify the prediction of $\mathcal{F}^*(\mathcal{S}_k \oplus \boldsymbol{x}_{k+1})$ to a closed-form expression in Sec. 4.2. Then we analyze the two factors that determine the posterior: (i) Component Re-weighting (CR) in Sec. 4.3 and (ii) Component Shifting (CS) in Sec. 4.4. We further compare these two factors under different pretraining task noises in Sec. 4.5 to show how the task noises of the pretraining distribution affect these two factors. Along with mathematical derivations, we also perform numerical computations using the Tetrahedron setting with 4 mixture components as shown in Fig. 9 with detailed descriptions in Appendix C.1.

Figure 2: The illustration of the prior distribution. We set $\boldsymbol{\mu}_\beta = \boldsymbol{w}_\beta \forall \beta \in \{1, 2, 3, 4\}$.

### 4.1 POSTERIOR

In this section, we give the closed-form expressions of the posterior derived from the prior after observing ICL input $\mathcal{S}_k \oplus \boldsymbol{x}_{k+1}$. Consequently, we have the following Lemma 1 for the posterior:

**Lemma 1.** *Under Assumption 2, the posterior probability of task $(\boldsymbol{\mu}, \boldsymbol{w})$ given $\mathcal{S}_k \oplus \boldsymbol{x}_{k+1}$ is:*

$$P(\boldsymbol{\mu}, \boldsymbol{w} \mid \mathcal{S}_k \oplus \boldsymbol{x}_{k+1}) = \sum_{\beta=1}^{M} \tilde{\pi}_\beta P(\boldsymbol{\mu}, \boldsymbol{w} \mid \widetilde{T}_\beta),$$

*where the mixture component $T_\beta$ in the prior is mapped to the mixture component $\widetilde{T}_\beta$ in the posterior and the probability of the mixture component is changed from $\pi_\beta$ to $\tilde{\pi}_\beta$. Specifically, we have:*

$$\tilde{\pi}_\beta = \pi_\beta C_1 c_\beta^{\boldsymbol{\mu}} c_\beta^{\boldsymbol{w}}, \tag{2}$$
$$c_\beta^{\boldsymbol{\mu}} = \exp(-(\|\boldsymbol{\mu}_\beta\|^2 - \|\boldsymbol{\mu}_\beta + (k+1)\delta_{\boldsymbol{\mu}} \bar{\boldsymbol{\mu}}\|^2_{(\boldsymbol{I}+(k+1)\delta_{\boldsymbol{\mu}} \bar{\Sigma}_{\boldsymbol{\mu}})^{-1}})/(2\sigma_\mu^2)),$$
$$c_\beta^{\boldsymbol{w}} = \exp(-(\|\boldsymbol{w}_\beta\|^2 - \|\boldsymbol{w}_\beta + k\delta_{\boldsymbol{w}} \bar{\boldsymbol{w}}\|^2_{(\boldsymbol{I}+k\delta_{\boldsymbol{w}} \bar{\Sigma}_{\boldsymbol{w}})^{-1}})/(2\sigma_w^2)),$$
$$P(\boldsymbol{\mu}, \boldsymbol{w} \mid \widetilde{T}_\beta) = \mathcal{N}(\boldsymbol{\mu} \mid \tilde{\boldsymbol{\mu}}_\beta, \sigma_\mu^2(\boldsymbol{I}+(k+1)\delta_{\boldsymbol{\mu}} \bar{\Sigma}_{\boldsymbol{\mu}})^{-1}) \mathcal{N}(\boldsymbol{w} \mid \tilde{\boldsymbol{w}}_\beta, \sigma_w^2(\boldsymbol{I}+k\delta_{\boldsymbol{w}} \bar{\Sigma}_{\boldsymbol{w}})^{-1}), \tag{3}$$
$$\tilde{\boldsymbol{\mu}}_\beta = (\boldsymbol{I}+(k+1)\delta_{\boldsymbol{\mu}} \bar{\Sigma}_{\boldsymbol{\mu}})^{-1}(\boldsymbol{\mu}_\beta + (k+1)\delta_{\boldsymbol{\mu}} \bar{\boldsymbol{\mu}}),$$
$$\tilde{\boldsymbol{w}}_\beta = (\boldsymbol{I}+k\delta_{\boldsymbol{w}} \bar{\Sigma}_{\boldsymbol{w}})^{-1}(\boldsymbol{w}_\beta + k\delta_{\boldsymbol{w}} \bar{\boldsymbol{w}}),$$

*where $C_1$ normalizes the mixture weights such that $\sum_\beta \tilde{\pi}_\beta = 1$. See Appendix D for proof details. For notations, we have $\delta_{\boldsymbol{\mu}} = \frac{\sigma_\mu^2}{\sigma_x^2}$, $\delta_{\boldsymbol{w}} = \frac{\sigma_w^2}{\sigma_y^2}$, and $\bar{\Sigma}_{\boldsymbol{\mu}} = \boldsymbol{I}$, $\bar{\boldsymbol{\mu}} = \frac{\sum_{i=1}^{k+1} \boldsymbol{x}_i}{k+1}$, $\bar{\Sigma}_{\boldsymbol{w}} = \frac{\sum_{i=1}^{k} \boldsymbol{x}_i \boldsymbol{x}_i^\top}{k}$, $\bar{\boldsymbol{w}} = \frac{\sum_{i=1}^{k} \boldsymbol{x}_i y_i}{k}$. (Notice $(\boldsymbol{I}+(k+1)\delta_{\boldsymbol{\mu}} \bar{\Sigma}_{\boldsymbol{\mu}})$ and $(\boldsymbol{I}+k\delta_{\boldsymbol{w}} \bar{\Sigma}_{\boldsymbol{w}})$ are positive definite and invertible.)*

Lemma 1 states that the posterior remains a Gaussian mixture, with its components from the prior being shifted and re-weighted. Therefore, to understand the effect of demonstrations on ICL, it is essential to understand how demonstrations affect the following two factors:

- **Component Re-weighting (CR).** Eq. 2: the $\beta^{\text{th}}$ component is re-weighted by a re-weighting coefficient $\exp\left(-\frac{\|\boldsymbol{\mu}_\beta\|^2 - \|\boldsymbol{\mu}_\beta + (k+1)\delta_{\boldsymbol{\mu}} \bar{\boldsymbol{\mu}}\|^2_{(\boldsymbol{I}+(k+1)\delta_{\boldsymbol{\mu}} \bar{\Sigma}_{\boldsymbol{\mu}})^{-1}}}{2\sigma_\mu^2}\right) \exp\left(-\frac{\|\boldsymbol{w}_\beta\|^2 - \|\boldsymbol{w}_\beta + k\delta_{\boldsymbol{w}} \bar{\boldsymbol{w}}\|^2_{(\boldsymbol{I}+k\delta_{\boldsymbol{w}} \bar{\Sigma}_{\boldsymbol{w}})^{-1}}}{2\sigma_w^2}\right)$.
- **Component Shifting (CS).** Eq. 3: the $\beta^{\text{th}}$ component center is shifted from $(\boldsymbol{\mu}_\beta, \boldsymbol{w}_\beta)$ to $(\tilde{\boldsymbol{\mu}}_\beta, \tilde{\boldsymbol{w}}_\beta)$, where $\tilde{\boldsymbol{\mu}}_\beta = (\boldsymbol{I}+(k+1)\delta_{\boldsymbol{\mu}} \bar{\Sigma}_{\boldsymbol{\mu}})^{-1}(\boldsymbol{\mu}_\beta + (k+1)\delta_{\boldsymbol{\mu}} \bar{\boldsymbol{\mu}})$, $\tilde{\boldsymbol{w}}_\beta = (\boldsymbol{I}+k\delta_{\boldsymbol{w}} \bar{\Sigma}_{\boldsymbol{w}})^{-1}(\boldsymbol{w}_\beta + k\delta_{\boldsymbol{w}} \bar{\boldsymbol{w}})$.

### 4.2 PREDICTION WITH IN-CONTEXT DEMONSTRATIONS

With Assumption 2 and Lemma 1, we have the following Corollary for the prediction $\mathcal{F}^*(\mathcal{S}_k \oplus \boldsymbol{x}_{k+1})$:

**Corollary 2.** *Let $\tilde{\boldsymbol{w}} = \sum_{\beta=1}^{M} \tilde{\pi}_\beta \tilde{\boldsymbol{w}}_\beta$. Under sample generation process 1 and Assumption 2, if the pretrained model $\mathcal{F}^*$ minimizes the pretraining loss, then the prediction on any sequence $\mathcal{S}_k \oplus \boldsymbol{x}_{k+1}$ by $\mathcal{F}^*$ is as follows: $\mathcal{F}^*(\mathcal{S}_k \oplus \boldsymbol{x}_{k+1}) = \left\langle \sum_{\beta=1}^{M} \tilde{\pi}_\beta \tilde{\boldsymbol{w}}_\beta, \boldsymbol{x}_{k+1} \right\rangle = \langle \tilde{\boldsymbol{w}}, \boldsymbol{x}_{k+1} \rangle$.*

*Proof.* By applying Assumption 1 to Eq. 1, $\mathcal{F}^*(\mathcal{S}_k \oplus \boldsymbol{x}_{k+1}) = \mathbb{E}_{(\boldsymbol{\mu}, \boldsymbol{w}) \sim \mathcal{D}^{\text{prior}}}[\langle \boldsymbol{w}, \boldsymbol{x}_{k+1} \rangle \mid \mathcal{S}_k \oplus \boldsymbol{x}_{k+1}]$. Using Lemma 1, this reduces to $\sum_{\beta=1}^{M} \tilde{\pi}_\beta \mathbb{E}_{(\boldsymbol{\mu}, \boldsymbol{w}) \sim \widetilde{T}_\beta}[\langle \boldsymbol{w}, \boldsymbol{x}_{k+1} \rangle]$. Due to the linearity of expectation and that of the inner product, it can be simplified as $\langle \sum_{\beta=1}^{M} \tilde{\pi}_\beta \tilde{\boldsymbol{w}}_\beta, \boldsymbol{x}_{k+1} \rangle = \langle \tilde{\boldsymbol{w}}, \boldsymbol{x}_{k+1} \rangle$. $\square$

Thus the prediction is a convex combination of predictions by the centers of those re-weighted and shifted mixture components in the posterior. We are interested in, how $\pi_\beta$ and $\boldsymbol{w}_\beta$ change to $\tilde{\pi}_\beta$ and $\tilde{\boldsymbol{w}}_\beta$ with increasing $k$, and how the pretraining distribution properties affect these changes.

### 4.3 ANALYSIS OF COMPONENT RE-WEIGHTING

In this section, we analyze the CR effect on $\tilde{\pi}_\beta$ as $k$ increases. We focus on whether $\tilde{\pi}_\alpha$ of $\widetilde{T}_\alpha$ surpasses $\tilde{\pi}_\beta$ of any other $\widetilde{T}_\beta$ with $\beta \neq \alpha$, where $\alpha$ is the index of the closest clean task to the task

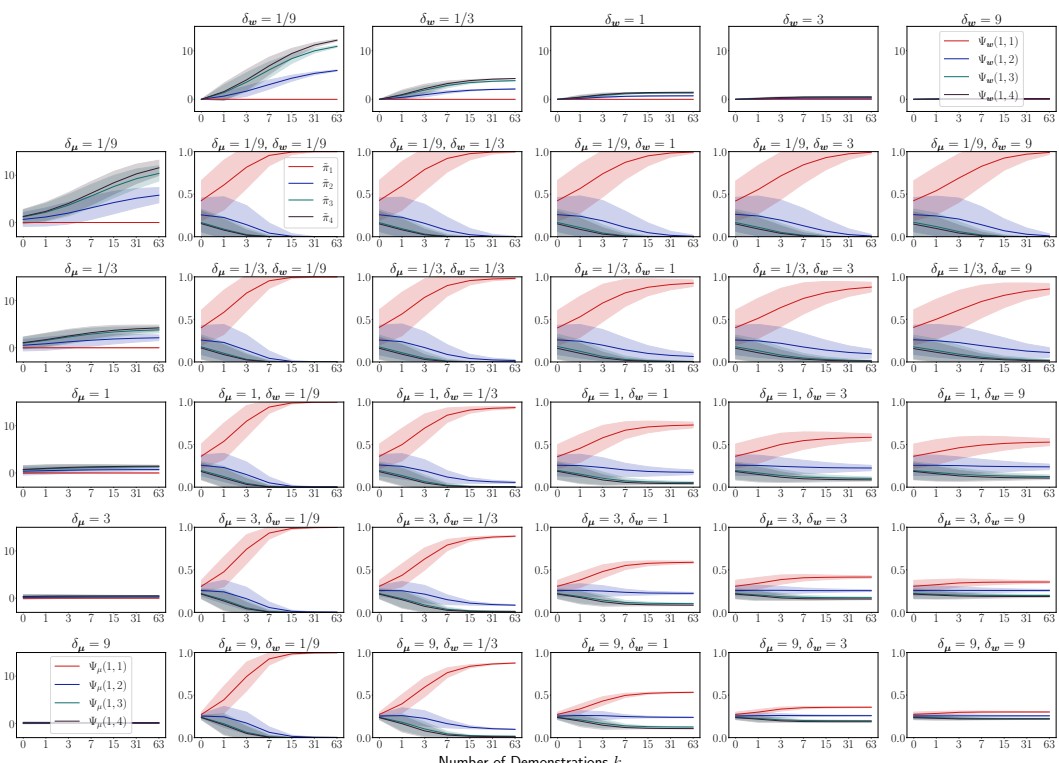

Figure 3: Numerical computation of $\Psi_{\boldsymbol{\mu}}$, $\Psi_{\boldsymbol{w}}$, and $\pi$ for CR with varying task noise parameters.

which in-context samples follow as Assumption 4. We assess this via the ratio $r(\alpha, \beta)$ of $\tilde{\pi}_\alpha$ to $\tilde{\pi}_\beta$:

$$r(\alpha, \beta) = \frac{\tilde{\pi}_\alpha}{\tilde{\pi}_\beta} = \frac{\pi_\alpha C_1 c_\alpha^{\boldsymbol{\mu}} c_\alpha^{\boldsymbol{w}}}{\pi_\beta C_1 c_\beta^{\boldsymbol{\mu}} c_\beta^{\boldsymbol{w}}} = \frac{\pi_\alpha}{\pi_\beta} \exp(\Psi_{\boldsymbol{\mu}}(\alpha, \beta) + \Psi_{\boldsymbol{w}}(\alpha, \beta)), \tag{4}$$

where we define two functions $\Psi_{\boldsymbol{\mu}}(\alpha, \beta) = \log(c_\alpha^{\boldsymbol{\mu}}/c_\beta^{\boldsymbol{\mu}})$ and $\Psi_{\boldsymbol{w}}(\alpha, \beta) = \log(c_\alpha^{\boldsymbol{w}}/c_\beta^{\boldsymbol{w}})$, and aim to analyze whether/how $r(\alpha, \beta)$ changes with increasing $k$.

**Analysis of $\Psi_{\boldsymbol{\mu}}(\alpha, \beta)$**   We further simplify the function $\Psi_{\boldsymbol{\mu}}(\alpha, \beta)$ as follows:

$$\Psi_{\boldsymbol{\mu}}(\alpha, \beta) = \left(\sum_{i=1}^{k+1} \|\boldsymbol{\mu}_\beta - \boldsymbol{x}_i\|^2 - \sum_{i=1}^{k+1} \|\boldsymbol{\mu}_\alpha - \boldsymbol{x}_i\|^2\right)/(2\sigma_x^2(1 + (k+1)\delta_{\boldsymbol{\mu}}^2)). \tag{5}$$

(See Appendix E.1 for derivation.) Since $\boldsymbol{x}_i \sim \mathcal{N}(\boldsymbol{\mu}^*, \tau_x^2 \boldsymbol{I})$ and $\boldsymbol{\mu}^*$ is closer to $\boldsymbol{\mu}_\alpha$, $\Psi_{\boldsymbol{\mu}}(\alpha, \beta)$ tends to be positive, contributing to $r(\alpha, \beta) > 1$. Yet, as $k$ grows large, $\Psi_{\boldsymbol{\mu}}(\alpha, \beta)$ stabilizes rather than increasing infinitely: $\lim_{k \to \infty} \Psi_{\boldsymbol{\mu}}(\alpha, \beta) = (\|\boldsymbol{\mu}_\beta - \boldsymbol{\mu}^*\|^2 - \|\boldsymbol{\mu}_\alpha - \boldsymbol{\mu}^*\|^2)/2\sigma_x^2$. The left side column of Fig. 3 shows the numerical computation of $\Psi_{\boldsymbol{\mu}}(\alpha, \beta)$ with varied task noises under Tetrahedron setting (Appendix C.1). The smaller $\delta_{\boldsymbol{\mu}} = \frac{\sigma_\mu^2}{\sigma_x^2}$, the easier for $\Psi_{\boldsymbol{\mu}}(\alpha, \beta)$ to grow large as $k$ increases.

**Analysis of $\Psi_{\boldsymbol{w}}(\alpha, \beta)$**   We further simplify the function $\Psi_{\boldsymbol{w}}(\alpha, \beta)$ as follows:

$$\Psi_{\boldsymbol{w}}(\alpha, \beta) = (\|\boldsymbol{w}_\beta - \boldsymbol{w}^*\|_{\boldsymbol{I} - (\boldsymbol{I} + k\delta_{\boldsymbol{w}} \bar{\Sigma}_{\boldsymbol{w}})^{-1}}^2 - \|\boldsymbol{w}_\alpha - \boldsymbol{w}^*\|_{\boldsymbol{I} - (\boldsymbol{I} + k\delta_{\boldsymbol{w}} \bar{\Sigma}_{\boldsymbol{w}})^{-1}}^2)/2\sigma_w^2. \tag{6}$$

(See Appendix E.2 for derivation.)  Since $k\delta_{\boldsymbol{w}} \bar{\Sigma}_{\boldsymbol{w}} = \delta_{\boldsymbol{w}} \sum_{i=1}^{k} \boldsymbol{x}_i \boldsymbol{x}_i^\top$, (see definition of $\bar{\Sigma}_{\boldsymbol{w}}$ in Lemma 1.) which is at least semi-positive definite, thus choosing $\boldsymbol{w}^*$ closer to $\boldsymbol{w}_\alpha$ tends to make the whole term positive. However, one should be caution that $\|\boldsymbol{w}_\beta - \boldsymbol{w}^*\|^2 \geq \|\boldsymbol{w}_\alpha - \boldsymbol{w}^*\|^2$ does not necessarily imply $\|\boldsymbol{w}_\beta - \boldsymbol{w}^*\|_{\boldsymbol{I} - (\boldsymbol{I} + k\delta_{\boldsymbol{w}} \bar{\Sigma}_{\boldsymbol{w}})^{-1}}^2 \geq \|\boldsymbol{w}_\alpha - \boldsymbol{w}^*\|_{\boldsymbol{I} - (\boldsymbol{I} + k\delta_{\boldsymbol{w}} \bar{\Sigma}_{\boldsymbol{w}})^{-1}}^2$. As $k$ approaches infinity, $\lim_{k \to \infty} k\delta_{\boldsymbol{w}} \bar{\Sigma}_{\boldsymbol{w}} = \lim_{k \to \infty} k\delta_{\boldsymbol{w}} \frac{\sum_{i=1}^{k} \boldsymbol{x}_i \boldsymbol{x}_i^\top}{k} = k\delta_{\boldsymbol{w}}(\boldsymbol{\mu}^* \boldsymbol{\mu}^{*\top} + \tau_x^2 \boldsymbol{I})$. Thus $\lim_{k \to \infty} \boldsymbol{I} - (\boldsymbol{I} + k\delta_{\boldsymbol{w}} \bar{\Sigma}_{\boldsymbol{w}})^{-1} \to \boldsymbol{I}$ and $\Psi_{\boldsymbol{w}}(\alpha, \beta)$ stabilizes rather than increasing infinitely: $\lim_{k \to \infty} \Psi_{\boldsymbol{w}}(\alpha, \beta) = (\|\boldsymbol{w}_\beta - \boldsymbol{w}^*\|^2 - \|\boldsymbol{w}_\alpha - \boldsymbol{w}^*\|^2)/2\sigma_w^2$ The top side row of Fig. 3 shows the numerical computation of $\Psi_{\boldsymbol{w}}(\alpha, \beta)$ with varied task noises under Tetrahedron setting (Appendix C.1). The smaller $\delta_{\boldsymbol{w}} = \frac{\sigma_w^2}{\sigma_y^2}$, the easier for $\Psi_{\boldsymbol{w}}(\alpha, \beta)$ to grow large with increasing $k$.

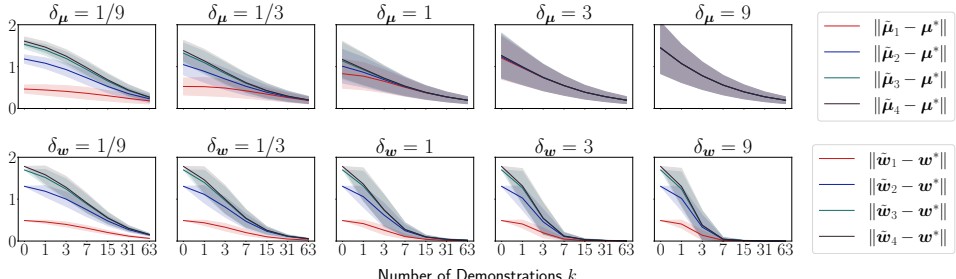

Figure 4: Numerical computations of $\|\tilde{\boldsymbol{\mu}}_\beta - \boldsymbol{\mu}^*\|$, $\|\tilde{\boldsymbol{w}}_\beta - \boldsymbol{w}^*\|$ for Component Shifting (CS).

**Numerical Computations of Component Re-weighting** We have seen how noises $\sigma_\mu$ and $\sigma_w$ of the task prior affect the influence of number $k$ of in-context samples on $\Psi_{\boldsymbol{\mu}}$ and $\Psi_{\boldsymbol{w}}$. We further show the numerical computation of $\tilde{\pi}_\beta$ in the center of Fig. 3. The figure shows the smaller $\delta_{\boldsymbol{\mu}}$ and $\delta_{\boldsymbol{w}}$ are, the larger $\Psi_{\boldsymbol{\mu}}(\alpha, \beta)$ and $\Psi_{\boldsymbol{w}}(\alpha, \beta)$ will be, and therefore the easier for the mixture component $\widetilde{T}_\alpha$ dominates the other components with an increasing number of in-context samples.

### 4.4 ANALYSIS OF COMPONENT SHIFTING

The Component Shifting in Eq. 3 of Lemma 1 consists of shifting parts for $\tilde{\boldsymbol{\mu}}_\beta$ and $\tilde{\boldsymbol{w}}_\beta$ separately:

$$\tilde{\boldsymbol{\mu}}_\beta = (\boldsymbol{I} + (k+1)\delta_{\boldsymbol{\mu}}\bar{\Sigma}_{\boldsymbol{\mu}})^{-1}(\boldsymbol{\mu}_\beta + (k+1)\delta_{\boldsymbol{\mu}}\bar{\boldsymbol{\mu}}), \tag{7}$$

$$\tilde{\boldsymbol{w}}_\beta = (\boldsymbol{I} + k\delta_{\boldsymbol{w}}\bar{\Sigma}_{\boldsymbol{w}})^{-1}(\boldsymbol{w}_\beta + k\delta_{\boldsymbol{w}}\bar{\boldsymbol{w}}). \tag{8}$$

In the following analysis, we examine these two shiftings with increasing $k$.

$\tilde{\boldsymbol{\mu}}_\beta$ **Analysis** Eq. 7 indicates the shifting of any $\tilde{\boldsymbol{\mu}}_\beta$. We further derive (see Appendix F.1) $\tilde{\boldsymbol{\mu}}_\beta$ as:

$$\tilde{\boldsymbol{\mu}}_\beta = (\boldsymbol{\mu}_\beta + k\delta_{\boldsymbol{\mu}}\bar{\boldsymbol{\mu}})/(1 + (k+1)\delta_{\boldsymbol{\mu}}). \tag{9}$$

Thus when $k$ increases, $\tilde{\boldsymbol{\mu}}_\beta$ moves close to the value of $\frac{\sum_{i=1}^k \boldsymbol{x}_i}{k}$ and $\lim_{k\to\infty} \tilde{\boldsymbol{\mu}}_\beta = \boldsymbol{\mu}^*$. We also show the numerical computation of the distance between shifted $\tilde{\boldsymbol{\mu}}_\beta$ and $\boldsymbol{\mu}^*$ in the first row of Fig. 4.

$\tilde{\boldsymbol{w}}_\beta$ **Analysis** Eq. 8 indicates the shifting of any $\tilde{\boldsymbol{w}}_\beta$. We further derive (see Appendix F.2) $\tilde{\boldsymbol{w}}_\beta$ as:

$$\tilde{\boldsymbol{w}}_\beta = (\boldsymbol{I} + k\delta_{\boldsymbol{w}}\bar{\Sigma}_{\boldsymbol{w}})^{-1}(\boldsymbol{w}_\beta - \boldsymbol{w}^*) + \boldsymbol{w}^*. \tag{10}$$

Notice when $k \to \infty$, $k\delta_{\boldsymbol{w}}\bar{\Sigma}_{\boldsymbol{w}} = k\delta_{\boldsymbol{w}} \frac{\sum_{i=1}^k \boldsymbol{x}_i \boldsymbol{x}_i^\top}{k} \to k\delta_{\boldsymbol{w}}(\tau_x^2 \boldsymbol{I} + \boldsymbol{w}^* \boldsymbol{w}^{*\top})$, thus $\lambda_d(k\delta_{\boldsymbol{w}}\bar{\Sigma}_{\boldsymbol{w}}) \to \infty$, and $\lim_{k\to\infty} \tilde{\boldsymbol{w}}_\beta = \boldsymbol{w}^*$, where $\lambda_d(\boldsymbol{A})$ indicates the minimum eigenvalue of $\boldsymbol{A}$. We also show the numerical computation of the distance between shifted $\tilde{\boldsymbol{w}}_\beta$ and $\boldsymbol{w}^*$ in the second row of Fig. 4.

### 4.5 THE RELATIONSHIP BETWEEN CR/CS AND TASK NOISES

Fig. 3, shows the factor of Component Shifting (CS), we observe that the smaller the value of $\sigma_\mu$ and $\sigma_w$, the faster the value of $\pi_\alpha$ becomes close to 1 thus dominates the other $\pi_\beta$. Fig. 4 shows, the smaller the value of $\sigma_\mu$ ($\sigma_w$) is, the slower the center $\tilde{\boldsymbol{\mu}}_\beta$ ($\tilde{\boldsymbol{w}}_\beta$) shifts towards to $\boldsymbol{\mu}^*$ ($\boldsymbol{w}^*$).

## 5 RISK BOUNDS FOR TWO MODES OF ICL

We give definitions to task retrieval and learning (Sec. 5.1) and derive risk bounds (Sec. 5.2).

### 5.1 FORMAL DEFINITIONS OF TASK RETRIEVAL AND LEARNING

Shown in Table 1 are the formal definitions of task retrieval and learning. Further, we present further assumptions for our analysis.

**Assumption 5.** *The target task of task retrieval and learning in addition to Assumption 4:*
*(a) task retrieval assumes* $\forall \beta \neq \alpha, \|\boldsymbol{\mu}_\beta - \boldsymbol{\mu}^*\|^2 - \|\boldsymbol{\mu}_\alpha - \boldsymbol{\mu}^*\|^2 \geq d_{\boldsymbol{\mu}}^2, \|\boldsymbol{w}_\beta - \boldsymbol{w}^*\|^2 - \|\boldsymbol{w}_\alpha - \boldsymbol{w}^*\|^2 \geq d_{\boldsymbol{w}}^2$, and $\tau_x^2\|\boldsymbol{w}_\beta - \boldsymbol{w}^*\|^2 - (1+\tau_x^2)\|\boldsymbol{w}_\alpha - \boldsymbol{w}^*\|^2 \geq \tau_x^2 u_{\boldsymbol{w}}^2$, and aims at $\mathcal{F}^*(\mathcal{S}_k \oplus \boldsymbol{x}_{k+1}) = \langle \boldsymbol{x}_{k+1}, \boldsymbol{w}_\alpha \rangle$;
*(b) task learning aims to have prediction* $\mathcal{F}^*(\mathcal{S}_k \oplus \boldsymbol{x}_{k+1}) = \langle \boldsymbol{x}_{k+1}, \boldsymbol{w}^* \rangle$ *and* $\forall \beta, \boldsymbol{w}^* \neq \boldsymbol{w}_\beta$.

Table 1: The learning goal and performance of two modes in ICL.

| Mode | Task Retrieval | Task Learning |
|---|---|---|
| In-context sample distribution | $(\boldsymbol{x}, y)$ pairs approximately follow the target clean task, i.e., $\boldsymbol{\mu}^* \approx \boldsymbol{\mu}_\alpha$, $\boldsymbol{w}^* \approx \boldsymbol{w}_\alpha$, and $\forall i$ $\boldsymbol{x}_i \sim \mathcal{N}(\boldsymbol{\mu}^*, \tau_x^2 \boldsymbol{I})$, $y_i = \langle \boldsymbol{x}_i, \boldsymbol{w} \rangle$. | $(\boldsymbol{x}, y)$ pairs exactly follow the target new (noisy) task, i.e., the target task is exactly $(\boldsymbol{\mu}^*, \boldsymbol{w}^*)$ and $\forall i$, $\boldsymbol{x} \sim \mathcal{N}(\boldsymbol{\mu}^*, \tau_x^2 \boldsymbol{I})$, $y_i = \langle \boldsymbol{x}_i, \boldsymbol{w} \rangle$. |
| Goal of the mode | retrieve $\boldsymbol{w}_\alpha$ of a clean task, i.e., aiming at $\mathcal{F}(S) = \langle \boldsymbol{w}_\alpha, \boldsymbol{x}_{k+1} \rangle$. | learn $\boldsymbol{w}^*$ of a new (noisy) task, i.e., aiming at $\mathcal{F}(S) = \langle \boldsymbol{w}^*, \boldsymbol{x}_{k+1} \rangle$ |
| Performance metric | The risk of input $\mathcal{S}_k \oplus \boldsymbol{x}_{k+1}$ is $(\mathcal{F}^*(\mathcal{S}_k \oplus \boldsymbol{x}_{k+1}) - \langle \boldsymbol{w}_\alpha, \boldsymbol{x}_{k+1} \rangle)^2$ | The risk of input $\mathcal{S}_k \oplus \boldsymbol{x}_{k+1}$ is $(\mathcal{F}^*(\mathcal{S}_k \oplus \boldsymbol{x}_{k+1}) - \langle \boldsymbol{w}^*, \boldsymbol{x}_{k+1} \rangle)^2$ |

Figure 5: Numerical computations of the squared losses of task learning/retrieval under varied task noise conditions with increasing $k$. For the fifth row, $\tilde{\boldsymbol{w}} = \sum_{\beta=1}^{4} \tilde{\pi}_\beta \tilde{\boldsymbol{w}}_\beta$. In the sixth row, $\mathcal{F}^*$ is the abbreviation of $\mathcal{F}^*(\mathcal{S}_k \oplus \boldsymbol{x}_{k+1})$, $y_{k+1}^L = \langle \boldsymbol{x}_{k+1}, \boldsymbol{w}^* \rangle$ indicates the desired prediction of $\mathcal{S}_k \oplus \boldsymbol{x}_{k+1}$ on task learning, and $y_{k+1}^R = \langle \boldsymbol{x}_{k+1}, \boldsymbol{w}_\alpha \rangle$ indicates the desired prediction of $\mathcal{S}_k \oplus \boldsymbol{x}_{k+1}$ on task retrieval.

Assumption 5(a) states that task retrieval aims to retrieve a clean task, with demonstrations' parameters $(\boldsymbol{\mu}^*, \boldsymbol{w}^*)$ closer to $(\boldsymbol{\mu}_\alpha, \boldsymbol{w}_\alpha)$. Assumption 5(b) suggests task learning focuses on learning a new task.

**Numerical Computations of the Prediction** With these new definitions, we numerically compute the squared losses of task learning and retrieval. Fig 5 shows the computed risks under varied task noise conditions with increasing $k$. The first row in Fig 5 shows the effect of CR. The second to the fourth rows in Fig 5 show the effect of CS. The fifth and sixth rows in Fig 5 show the simulated squared error of the predicted $\tilde{\boldsymbol{w}}$ and prediction measured on task learning and retrieval. A *unique phenomenon* is that when $\delta_\mu$ and $\delta_w$ are small, the squared error of task retrieval decreases and then increases with increasing $k$. This is because when variances are small and $k$ is small, the CR effect dominates first, and then when $k$ is large, the CS effect dominates.

## 5.2 RISK BOUNDS

We show upper bounds on the risks of task learning and retrieval with the following theorems:

**Theorem 3.** *Consider a next-token prediction model attaining the optimal pretraining risk. Then, the task learning risk is upper bounded by:*

$$\mathbb{E}_{\mathcal{S}_k}[\mathcal{L}_k^L] < \frac{4(1 + d\tau_x^2)}{\tau_x^4 \delta_{\boldsymbol{w}}^2 k^2} + O(k^{\delta - \frac{5}{2}})$$

*where $\delta$ is an arbitrarily small positive constant. The bound decreases as the square of the inverse of $k$ when $k$ is large.*

The notations $\delta_{\boldsymbol{\mu}}$, $\delta_{\boldsymbol{w}}$ and $k$ are colored for easier observation. (See Appendix G.2 for proof details)

**Theorem 4.** *Consider a next-token prediction model attaining the optimal pretraining risk. Then, the task retrieval risk is upper bounded by:*

$$\mathbb{E}_{\mathcal{S}_k}[\mathcal{L}_k^R] < 16rMC_{k=0}\exp(-(\frac{d_{\boldsymbol{\mu}}^2}{2\sigma_\mu^2} + \frac{d_{\boldsymbol{w}}^2}{2\sigma_w^2}))(1 + \frac{2\tau_x\sqrt{d}k^{\frac{\delta}{2}-\frac{3}{4}}}{\sigma_\mu^2}) + 4(1 + d\tau_x^2) + O(k^{-1}),$$

*(See Appendix G.3 for proof details) where $\delta$ is an arbitrarily small positive number and $C_{k=0}$ is a constant depends on the setting as Eq. 17. We also show that when $\delta_{\boldsymbol{\mu}}$ and $\delta_{\boldsymbol{w}}$ are sufficiently small (training data has high quality), there is a special region for $k$ such that when $k$ in that region:*

$$\mathbb{E}_{\mathcal{S}_k}[\mathcal{L}_k^R] < 16rMC_{k=0}\exp(-k(\frac{d_{\boldsymbol{\mu}}^2}{8\sigma_x^2} + \frac{u_{\boldsymbol{w}}^2\tau_x^2}{8\sigma_y^2})) + 16(1 + d\tau_x^2)((1 + \tau_x^2)^2\delta_{\boldsymbol{w}}^2k^2 + 3\exp(-\frac{k^{\frac{1}{2}}}{8})).$$

*(See Appendix G.3 for proof details.) We observe that in this region when $k$ is small, the first and third terms dominate and exponential decay, and then when $k$ is large, the second term dominates, therefore we observe a U-shaped pattern in Fig. 5 and Fig. 6.*

Lemma 5, a simple variation of the above theorem, can be used to explain zero-shot ICL, an ICL algorithm that oeprates with random or no labels (Lyu et al., 2023).

**Lemma 5.** *Assuming a next-token prediction model attains the optimal pretraining risk, and Assumptions 2 with only two mixture component with centers $(\boldsymbol{\mu}_\alpha, \boldsymbol{w}_\alpha)$ and $(\boldsymbol{\mu}_\beta, \boldsymbol{w}_\beta) = (-\boldsymbol{\mu}_\alpha, -\boldsymbol{\mu}_\alpha)$, when performing ICL with in-context samples following $\boldsymbol{x}_i \sim \mathcal{N}(\boldsymbol{\mu}^* \mid \tau_x^2\boldsymbol{I})$ and $y_i = 0$, thus $y$ does not provide information to the task. When $\|\boldsymbol{\mu}^* - \boldsymbol{\mu}_\beta\|^2 - \|\boldsymbol{\mu}^* - \boldsymbol{\mu}_\alpha\|^2 = d_{\boldsymbol{\mu}}^2$, the task learning risk is upper bounded by:*

$$\mathbb{E}_{\mathcal{S}_k}[\mathcal{L}_k^R] < 16rC_{k=0}\exp(-\frac{d_{\boldsymbol{\mu}}^2}{2\sigma_\mu^2})(1 + \frac{2\tau_x\sqrt{d}k^{\frac{\delta}{2}-\frac{3}{4}}}{\sigma_\mu^2}) + 4(1 + d\tau_x^2) + O(k^{-1})$$

*(See Appendix H for proof details) where $\delta$ is an arbitrarily small positive number and $C_{k=0}$ is a constant depends on the setting as Eq. 17. We also show that when $\delta_{\boldsymbol{\mu}}$ and $\delta_{\boldsymbol{w}}$ are sufficiently small (training data has high quality), there is a special region for $k$ such that when $k$ in that region:*

$$\mathbb{E}_{\mathcal{S}_k}[\mathcal{L}_k^R] < 16rC_{k=0}\exp(-\frac{d_{\boldsymbol{\mu}}^2k}{8\sigma_x^2}) + 4(1 + d\tau_x^2)\max\{1, 4k^2\delta_{\boldsymbol{w}}^2(1 + \tau_x^2)^2\} + 48(1 + d\tau_x^2)\exp(-\frac{k^{\frac{1}{2}}}{8}).$$

*(See Appendix H for proof details.) We observe that in this region when $k$ is small, the first and third terms dominate and exponential decay, and then when $k$ is large, the second term dominates.*

## 6 TRANSFORMER SIMULATION

We further examine if a Transformer trained on samples from generative model 6 following Assumption 2, matches the performance of Bayesian inference. We consider three factors of the prior in our experiment: *task noise*, *number of components*, and *feature dimension*. For scalar $y$, we transform it to a $d$-dimensional vector $[y, 0, \ldots, 0]$. Thus, input $\mathcal{S}_k \oplus \boldsymbol{x}_{k+1}$ forms a $(2k + 1) \times d$ matrix, comprising $k$ pairs of $(\boldsymbol{x}_i, y_i)$ and $\boldsymbol{x}_{k+1}$. The prediction of the Transformer is denoted as $\hat{\mathcal{F}}$ and the prediction of Bayesian inference is denoted as $\mathcal{F}^*$. From Fig 6, Fig 7 and Fig 8, we are able to observe that the pretrained Transformer model are able to approximate Baeys-optimal predictor under varied settings. (Due to the page limitation, we put Fig 6, Fig 7, Fig 8 and the training setting into Appendix A.)

## 7 CONCLUSION

We analyze in-context learning's two modes: task retrieval and task learning. Our findings show that with a Bayes-optimal pretrained next-token prediction model, in-context inference predicts the posterior distribution's mean given in-context samples. Task learning risk decreases quadratically with more in-context samples, while task retrieval risk follows a U-shaped pattern. These findings are validated through numerical computations and Transformer experiments.

We conclude our paper with the limitations of our current model: (i) the gap between linear regression tasks and complex, non-linear real-world NLP tasks; (ii) the unverified existence of a U-shaped phenomenon in real LLMs; and (iii) demonstration labels are assumed to be noiseless.

## 8 REPRODUCIBILITY STATEMENT

The code for all experiments reported in this paper is publicly accessible. For the purpose of reproducibility, the code can be found at the following anonymized GitHub repository: `https://anonymous.4open.science/r/ICLMODES-1B28`.

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

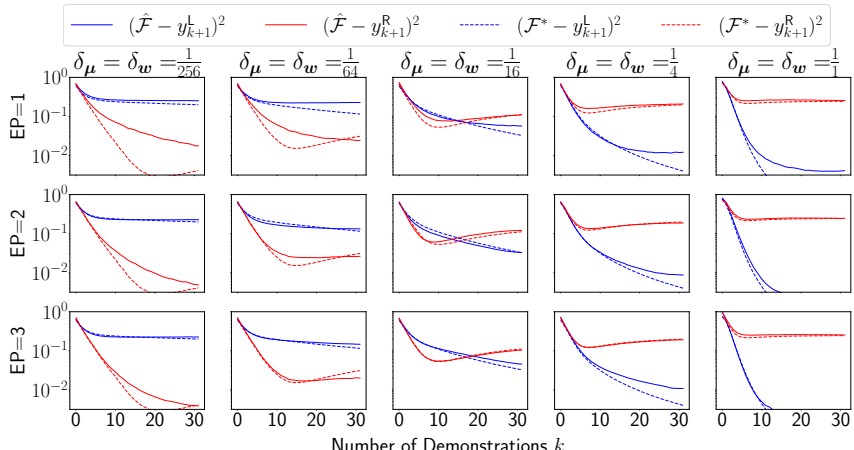

Figure 6: The figure shows the simulation of the Transformer under varied noise levels. $\mathcal{F}^*$ indicates the prediction of Bayesian inference while $\hat{\mathcal{F}}$ indicates the prediction of the trained Transformer. One can observe that the lower the values of $\delta_{\boldsymbol{\mu}}$ and $\delta_{\boldsymbol{w}}$, *i.e.*, the noise levels, the stronger the dip-and-rise phenomenon and the harder for the Transformer to approach the Bayesian inference.

Xinyi Wang, Wanrong Zhu, and William Yang Wang. Large language models are implicitly topic models: Explaining and finding good demonstrations for in-context learning. *arXiv preprint arXiv:2301.11916*, 2023.

Sang Michael Xie, Aditi Raghunathan, Percy Liang, and Tengyu Ma. An explanation of in-context learning as implicit Bayesian inference. In *International Conference on Learning Representations (ICLR)*. OpenReview.net, 2022.

Kang Min Yoo, Junyeob Kim, Hyuhng Joon Kim, Hyunsoo Cho, Hwiyeol Jo, Sang-Woo Lee, Sang-goo Lee, and Taeuk Kim. Ground-truth labels matter: A deeper look into input-label demonstrations. In Yoav Goldberg, Zornitsa Kozareva, and Yue Zhang (eds.), *Proceedings of the 2022 Conference on Empirical Methods in Natural Language Processing, EMNLP 2022, Abu Dhabi, United Arab Emirates, December 7-11, 2022*, pp. 2422–2437. Association for Computational Linguistics, 2022.

Ruiqi Zhang, Spencer Frei, and Peter L. Bartlett. Trained transformers learn linear models in-context. *CoRR*, abs/2306.09927, 2023a.

Yufeng Zhang, Fengzhuo Zhang, Zhuoran Yang, and Zhaoran Wang. What and how does in-context learning learn? bayesian model averaging, parameterization, and generalization. *arXiv preprint arXiv:2305.19420*, 2023b.

Zihao Zhao, Eric Wallace, Shi Feng, Dan Klein, and Sameer Singh. Calibrate before use: Improving few-shot performance of language models. PMLR, 2021.

## A ADDITIONAL RESULTS OF TRANSFORMER SIMULATION

The Fig 6, Fig. 7 and Fig. 8 of experimental results of the Transformer model discussed in Sec. 6 are shown here due to the page limitation.

In Fig 6, we consider the Tetrahedron setting (see Apendix C.1) under varied task noises ($\delta_{\boldsymbol{\mu}} = \delta_{\boldsymbol{w}} \in \{1/256, 1/64, 1/16, 1/4, 1\}$). The results show that the lower the variance, the stronger the dip-and-rise phenomenon in both Bayesian and Transformer inference, and it is also harder for the Transformer to capture the Bayesian prediction. It also takes more training epochs for the Transformer to catch the Bayesian prediction when task noises are small. In Fig 7, we consider settings of regular shapes (see Appendix C.1) with different numbers of vertices/components ($M \in \{4, 6, 8, 12, 20\}$). In Fig 8, we consider settings with varied dimensions (see Appendix C.2, $d \in \{2, 4, 8, 16, 32\}$).

We conduct experiments using a 10-layer, 8-head Transformer decoder with 1024-dimensional feedforward layers, and the input dimension is set to $d$, equivalent to the dimension of $\boldsymbol{x}$. In each training epoch, we train the model with 10000 batches and each batch contains 256 samples. We use

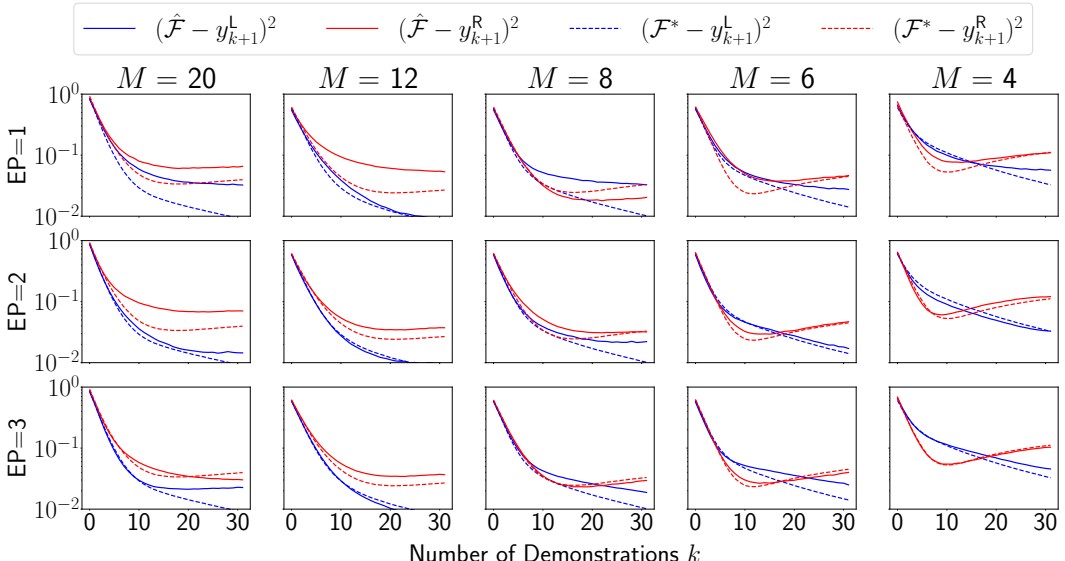

Figure 7: The figure shows the simulation of the Transformer under varied component densities. $\mathcal{F}^*$ indicates the prediction of Bayesian inference while $\hat{\mathcal{F}}$ indicates the prediction of the trained Transformer. $M$ indicates the number of mixture components on the sphere with radius 1, and $\delta_{\boldsymbol{\mu}} = \delta_{\boldsymbol{w}} = \frac{1}{16}$. It is observed that the higher the component density, the harder for the Transformer to approach the Bayesian inference.

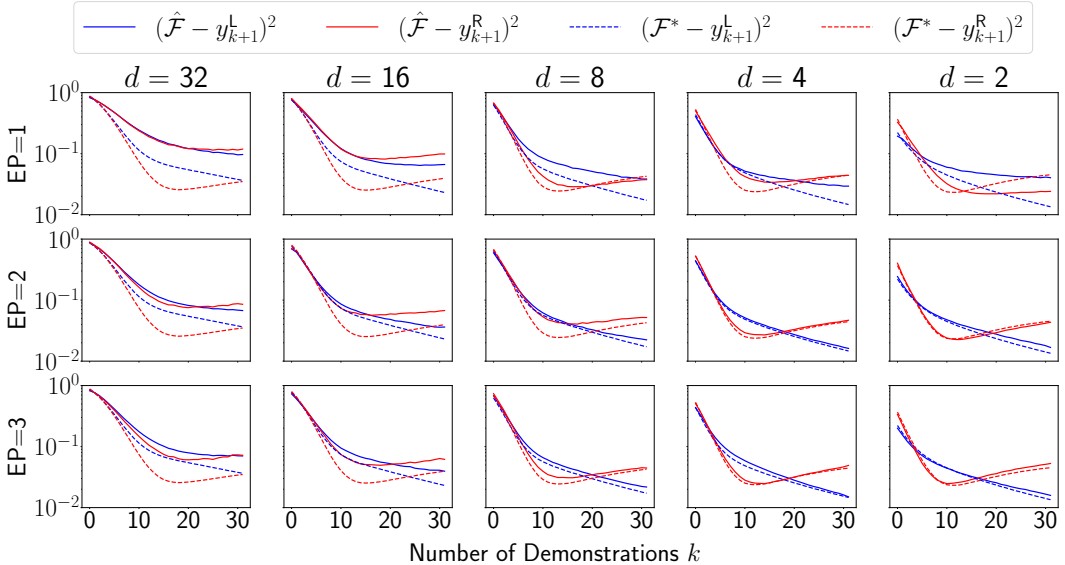

Figure 8: The figure shows the simulation of the Transformer under varied dimensions. $\mathcal{F}^*$ indicates the prediction of Bayesian inference while $\hat{\mathcal{F}}$ indicates the prediction of the trained Transformer. $d$ indicates the dimension as well as the number of mixture components on the sphere with radius 1, and $\delta_{\boldsymbol{\mu}} = \delta_{\boldsymbol{w}} = \frac{1}{16}$. It is observed that the higher the number of dimensions, the harder for the Transformer to approach the Bayesian inference.

AdamW Loshchilov & Hutter (2017) as the optimizer with weight decay as 0.00001, and the learning rate is set to 0.00001.

## B    NOTATIONS

This section collects all notations used in the main paper.

**Notations initially introduced in Sec. 3**

- $\mathcal{F}$: a next-token prediction model.
- $\hat{\mathcal{F}}$: a pretrained next-token prediction model.
- $\mathcal{F}^*$: a next-token prediction model that attains Bayes risk minimization.
- $\mathcal{F}_k$: a next-token prediction model for $k$ in-context samples.
- $\mathcal{F}_k^*$: a next-token prediction model that attains Bayes risk minimization for $k$ in-context samples.
- $\boldsymbol{x}$ and $y$: input and label for a task, *e.g.*, $\boldsymbol{x}$ and $y$ of a linear regression task $y = \boldsymbol{x}^\top \boldsymbol{w}$.
- $k$: the number of in-context samples.
- $K$: the max number of in-context samples in a sequence.
- $\mathcal{S}_k$: a sequence of $k$ in-context samples, $[\boldsymbol{x}_1, y_1, \ldots, \boldsymbol{x}_k, y_k]$.
- $\mathcal{S}_K$: a sequence of $K$ in-context samples, $[\boldsymbol{x}_1, y_1, \ldots, \boldsymbol{x}_K, y_K]$.
- $\mathcal{S}_k \oplus \boldsymbol{x}_{k+1}$:  a sequence of $k$ in-context samples and $\boldsymbol{x}_{k+1}$ pending to be predicted, $[\boldsymbol{x}_1, y_1, \ldots, \boldsymbol{x}_k, y_k, \boldsymbol{x}_{k+1}]$.
- $\boldsymbol{\mu}$ and $\boldsymbol{w}$: the parameters control a task. $\boldsymbol{\mu}$ controls the distribution of $\boldsymbol{x}$ and $\boldsymbol{w}$ controls how $\boldsymbol{x}$ maps to $y$.
- $\mathcal{D}^{\text{prior}}$ and $\mathcal{D}_{\boldsymbol{\mu}, \boldsymbol{w}}$: $\mathcal{D}^{\text{prior}} = \mathcal{D}_{\boldsymbol{\mu}, \boldsymbol{w}}$, and they represent the task prior distribution where each task is controlled by parameters $\boldsymbol{\mu}$ and $\boldsymbol{w}$.
- $\mathcal{D}_{\boldsymbol{x}}(\boldsymbol{\mu})$: the conditional distribution of $\boldsymbol{x}$ conditioned on $\boldsymbol{\mu}$ of the task $(\boldsymbol{\mu}, \boldsymbol{w})$.
- $\mathcal{D}_{\boldsymbol{x}, y}(\boldsymbol{\mu}, \boldsymbol{w})$: the joint distribution of $(\boldsymbol{x}, y)$ in the task $(\boldsymbol{\mu}, \boldsymbol{w})$.
- $\mathcal{D}_{y|\boldsymbol{x}}(\boldsymbol{w})$: the joint conditional distribution of $y$ conditioned on the input $\boldsymbol{x}$ and $\boldsymbol{w}$ of the task $(\boldsymbol{\mu}, \boldsymbol{w})$.
- $P(\boldsymbol{\mu}, \boldsymbol{w})$: the task probability of $(\boldsymbol{\mu}, \boldsymbol{w})$ in the task prior $\mathcal{D}^{\text{prior}}$.
- $P(\boldsymbol{x} \mid \mu)$: the probability of $\boldsymbol{x}$ in $\mathcal{D}_{\boldsymbol{x}}(\boldsymbol{\mu})$.
- $P(y \mid \boldsymbol{x}, \boldsymbol{w})$: the probability of $y$ in $\mathcal{D}_{y|\boldsymbol{x}}(\boldsymbol{w})$.
- $\mathcal{L}(\mathcal{F})$: the risk of $\mathcal{F}$ on samples generated from generation process 1.
- $M$: the number of mixture components in a Gaussian mixture prior.
- $\alpha, \beta$: the index of a mixture component in a Gaussian mixture prior.
- $T_\beta$: the $\beta^{\text{the}}$ mixture component in a Gaussian mixture prior.
- $\pi_\beta$: the mixture weight of the $\beta^{\text{th}}$ mixture component in a Gaussian mixture prior.
- $\boldsymbol{\mu}_\beta$ and $\boldsymbol{w}_\beta$: $(\boldsymbol{\mu}_\beta, \boldsymbol{w}_\beta)$ is the mean of the $\beta^{\text{th}}$ mixture component.
- $\sigma_\mu$ and $\sigma_w$: the task noises, *i.e.*, the noise scales of $\boldsymbol{\mu}$ and $\boldsymbol{w}$.
- $\sigma_x$ and $\sigma_y$: the sample noises, *i.e.*, the noise scales of $\boldsymbol{x}$ and $y$.
- $d$: the dimension of the $\boldsymbol{x}$.
- $r$: the max ratio of two mixture weights of two tasks.

**Notations initially introduced in Sec. 4**

- $\|\boldsymbol{x}\|^2$: for any vector $\boldsymbol{x}$, $\|\boldsymbol{x}\|^2 = \boldsymbol{x}^\top \boldsymbol{x}$.
- $\|\boldsymbol{x}\|_{\boldsymbol{A}}^2$: for any vector $\boldsymbol{x}$ and matrix $\boldsymbol{A}$, $\|\boldsymbol{x}\|_{\boldsymbol{A}}^2 = \boldsymbol{x}^\top \boldsymbol{A} \boldsymbol{x}$.
- $P(\boldsymbol{\mu}, \boldsymbol{w} \mid \mathcal{S}_k^+)$: the probability of task $(\boldsymbol{\mu}, \boldsymbol{w})$ in the posterior after observing $\mathcal{S}_k^+$.
- $\widetilde{T}_\beta$: the $\beta^{\text{th}}$ mixture component in the Gaussian mixture posterior.

- $\tilde{\pi}_\beta$: the mixture weight of the $\beta^{\text{th}}$ mixture component in the Gaussian mixture posterior.
- $\tilde{\boldsymbol{\mu}}_\beta$ and $\tilde{\boldsymbol{w}}_\beta$: the center of the $\beta^{\text{th}}$ mixture component in the Gaussian mixture posterior.
- $P(\boldsymbol{\mu}, \boldsymbol{w} \mid \widetilde{T}_\beta)$: the probability of task $(\boldsymbol{\mu}, \boldsymbol{w})$ in the $\beta^{\text{th}}$ mixture component.
- $\delta_{\boldsymbol{\mu}}$ and $\delta_{\boldsymbol{w}}$: the ratio of squared task noise over squared sample noise. $\sigma_{\boldsymbol{\mu}} = \frac{\sigma_\mu}{\sigma_x}$, and $\sigma_{\boldsymbol{w}} = \frac{\sigma_w}{\sigma_y}$.
- $\bar{\Sigma}_{\boldsymbol{\mu}}$: $\bar{\Sigma}_{\boldsymbol{\mu}} = \boldsymbol{I}$.
- $\bar{\Sigma}_{\boldsymbol{w}}$: $\bar{\Sigma}_{\boldsymbol{w}} = \frac{\sum_{i=1}^{k} \boldsymbol{x}_i \boldsymbol{x}_i^\top}{k}$.
- $\bar{\boldsymbol{\mu}}$: $\bar{\boldsymbol{\mu}} = \frac{\sum_{i=1}^{k+1} \boldsymbol{x}_i}{k+1}$.
- $\bar{\boldsymbol{w}}$: $\bar{\boldsymbol{w}} = \frac{\sum_{i=1}^{k} \boldsymbol{x}_i y_i}{k}$.
- $\tilde{\boldsymbol{w}}$: the mean of $\boldsymbol{w}$ in the task posterior.
- $c_\beta^{\boldsymbol{\mu}}$ and $c_\beta^{\boldsymbol{w}}$: parts of the re-weighting coefficient of Component Re-weighting.
- $\Psi_{\boldsymbol{\mu}}(\alpha, \beta)$ and $\Psi_{\boldsymbol{w}}(\alpha, \beta)$: functions defined to help analyze the phenomenon of Component Re-weighting.
- $r(\alpha, \beta)$: the ratio of the mixture weight $\tilde{\pi}_\alpha$ of $\widetilde{T}_\alpha$ over the mixture weight $\tilde{\pi}_\beta$ of $\widetilde{T}_\beta$.
- $\lambda_d(\boldsymbol{A})$: the $d^{\text{th}}$ largest eigenvalue of matrix $\boldsymbol{A}$. In this paper $\boldsymbol{A} \in \mathbb{R}^{d \times d}$, thus $\lambda_d(\boldsymbol{A})$ represents the smallest eigenvalue value of matrix $\boldsymbol{A}$.
- $\lambda_1(\boldsymbol{A})$: the $1^{\text{st}}$ largest eigenvalue of matrix $\boldsymbol{A}$.

**Notations initially introduced in Sec. 5**

- $d_{\boldsymbol{\mu}}^2$: $\forall \beta \neq \alpha, \|\boldsymbol{\mu}_\beta - \boldsymbol{\mu}^*\|^2 - \|\boldsymbol{\mu}_\alpha - \boldsymbol{\mu}^*\|^2 \geq d_{\boldsymbol{\mu}}^2$, the squared $\boldsymbol{\mu}$-margin of any other $\boldsymbol{\mu}_\beta$ over $\boldsymbol{\mu}_\alpha$.
- $d_{\boldsymbol{w}}^2$: $\forall \beta \neq \alpha, \|\boldsymbol{w}_\beta - \boldsymbol{w}^*\|^2 - \|\boldsymbol{w}_\alpha - \boldsymbol{w}^*\|^2 \geq d_{\boldsymbol{w}}^2$, the squared $\boldsymbol{w}$-margin of any other $\boldsymbol{w}_\beta$ over $\boldsymbol{w}_\alpha$.
- $u_{\boldsymbol{w}}^2$: $\forall \beta \neq \alpha, \tau_x^2 \|\boldsymbol{w}_\beta - \boldsymbol{w}^*\|^2 - (1 + \tau_x^2)\|\boldsymbol{w}_\alpha - \boldsymbol{w}^*\|^2 \geq \tau_x^2 u_{\boldsymbol{w}}^2$, the weighted squared $\boldsymbol{w}$-margin of any other $\boldsymbol{w}_\beta$ over $\boldsymbol{w}_\alpha$.
- $y_{k+1}^{\text{L}}$: the label or the target prediction of task learning. $y_{k+1}^{\text{L}} = \langle \boldsymbol{x}_{k+1}, \boldsymbol{w}^* \rangle$
- $y_{k+1}^{\text{R}}$: the label or the target prediction of task retrieval. $y_{k+1}^{\text{R}} = \langle \boldsymbol{x}_{k+1}, \boldsymbol{w}_\alpha \rangle$

## C  PRIOR EXAMPLE

In this section, we introduce some examples of prior we use in simulation experiments. We split the examples based on the shape of the centers of the topics in the priors. Those shapes include 3-dimensional regular polyhedrons in Sec. C.1, and $d$-dimensional examples in Sec. C.2.

### C.1  REGULAR POLYHEDRONS

For the abstract of the task prior, we consider 3-dimensional regular polyhedrons including Tetrahedron (4 vertices/centers), Octahedron (6 vertices/centers), Hexahedron (8 vertices/centers), Icosahedron (12 vertices/centers), and Dodecahedron (20 vertices/centers), listed with increasing density.

A regular polyhedron of a task prior with $M$ components is set as follows including all the parameters used Assumption 2:

- Dimension $d = 3$, number of mixture component $M = M$, and $\beta \in \{1, \ldots, M\}$;
- The probabilities of all topics are the same, $\pi_\beta = 1/M$, for all $\beta \in \{1, \ldots, M\}$;
- For noises of the input $\boldsymbol{x}$ and the label $y$, we have $\sigma_x = \sigma_y = 1$, and $\tau_x = 1$;
- For noises of $\boldsymbol{\mu}$ and $\boldsymbol{w}$, they are set equal to 0.25 if not specified;

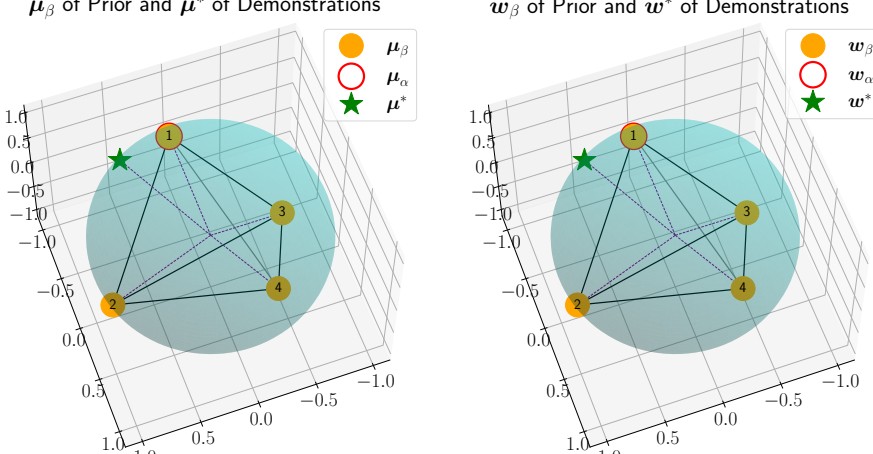

Figure 9: The illustration figure of the prior distribution. $\boldsymbol{\mu}_\beta$ and $\boldsymbol{w}_\beta$ for $\beta \in \{1,2,3,4\}$ are mixture component centers in the prior. $\boldsymbol{\mu}_\alpha$ and $\boldsymbol{w}_\alpha$ for $\alpha = 1$ (numbers are noted in the center of circles) are the centers of the target task for task learning. Both $\boldsymbol{\mu}^*$ and $\boldsymbol{w}^*$ govern the distribution of demonstrations. The dotted purple lines highlight the distance of 1 from the origin $(0,0,0)$ to any point represented by $\boldsymbol{\mu}$ or $\boldsymbol{w}$.

- The centers of mixture components shape a regular polyhedron with $M$ vertices, and for all $\beta$, $\boldsymbol{\mu}_\beta = \boldsymbol{w}_\beta$;

- For demonstrations, we have $\boldsymbol{\mu}^* = \frac{2\boldsymbol{\mu}_1 + \boldsymbol{\mu}_2}{\|2\boldsymbol{\mu}_1 + \boldsymbol{\mu}_2\|}$ and $\boldsymbol{w}^* = \frac{2\boldsymbol{w}_1 + \boldsymbol{w}_2}{\|2\boldsymbol{w}_1 + \boldsymbol{w}_2\|}$, where $\boldsymbol{\mu}_2 \in \arg\min_{\boldsymbol{\mu}_\beta} \|\boldsymbol{\mu}_\beta - \boldsymbol{\mu}_1\|$.

We will mainly use the Tetrahedron setting in the experiment, therefore, we further visualize the setting and give the description. We introduce the setting of the prior Tetrahedron including all the parameters used Assumption 2. The 3D visualization of the clean tasks in the prior and the task of demonstrations are shown in Fig. **??**. The exact parameters are shown as follows:

- Dimension $d = 3$, number of topics $M = 4$, and $\beta \in \{1,2,3,4\}$;

- The probabilities of all topics are the same, $\pi_\beta = 1/4$, *i.e.*, for all $\beta \in \{1,2,3,4\}$;

- For noise of the input $\boldsymbol{x}$ and the label $y$, we have $\sigma_x = \sigma_y = 1$, and $\tau_x = 1$;

- For noises of $\boldsymbol{\mu}$ and $\boldsymbol{w}$, they are set equal to $0.25$ if not specified;

- The centers of topics shape a tetrahedron as shown in Fig. **??** $\boldsymbol{\mu}_1 = \boldsymbol{w}_1 = [0, 0, -1]^\top$, $\boldsymbol{\mu}_2 = \boldsymbol{w}_2 = [\sqrt{\frac{8}{9}}, 0, \frac{1}{3}]^\top$, $\boldsymbol{\mu}_3 = \boldsymbol{w}_3 = [-\sqrt{\frac{2}{9}}, +\sqrt{\frac{2}{3}}, \frac{1}{3}]^\top$, and $\boldsymbol{\mu}_4 = \boldsymbol{w}_4 = [-\sqrt{\frac{2}{9}}, -\sqrt{\frac{2}{3}}, \frac{1}{3}]^\top$;

- For demonstrations, we have $\boldsymbol{\mu}^* = \frac{2\boldsymbol{\mu}_1 + \boldsymbol{\mu}_2 + 0.2\boldsymbol{\mu}_3}{\|2\boldsymbol{\mu}_1 + \boldsymbol{\mu}_2 + 0.2\boldsymbol{\mu}_3\|}$ and $\boldsymbol{w}^* = \frac{2\boldsymbol{w}_1 + \boldsymbol{w}_2 + 0.2\boldsymbol{w}_3}{\|2\boldsymbol{w}_1 + \boldsymbol{w}_2 + 0.2\boldsymbol{w}_3\|}$. (We slightly shift the center towards $(\boldsymbol{\mu}_3, \boldsymbol{w}_3)$ for visualization purpose, so that $\beta = 3$ and $\beta = 4$ could produce slightly different results.)

## C.2    $d$-DIMENSIONAL EXAMPLES

We consider $d$-dimensional examples with exact $d$ centers where $d \in \{2, 4, 8, 16, 32\}$. A $d$-dimensional example with $d$ vertices is set as follows:

- Dimension $d = d$, number of mixture component $M = d$, and $\beta \in \{1, \ldots, d\}$;

- The probabilities of all topics are the same, $\pi_\beta = 1/d$, for all $\beta \in \{1, \ldots, d\}$;

- For noise of the input $\boldsymbol{x}$ and the label $y$, we have $\sigma_x = \sigma_y = 1$, and $\tau_x = 1$;

- For noises of $\boldsymbol{\mu}$ and $\boldsymbol{w}$, they are set equal to $0.25$ if not specified;

- For all $\beta$, $\boldsymbol{\mu}_{\beta,i} = \boldsymbol{w}_{\beta,i} = \begin{cases} 1 & \text{if } i = \beta \\ 0 & \text{if } i \neq \beta \end{cases}$, *i.e.*, $\boldsymbol{\mu}_\beta$ is a vector with all elements $0$ except the $\beta^{\text{th}}$ element is $1$, and $\boldsymbol{w}_\beta$ is equal to $\boldsymbol{\mu}_\beta$.

- For demonstrations, we have $\boldsymbol{\mu}^* = \frac{2\boldsymbol{\mu}_1 + \boldsymbol{\mu}_2}{\|2\boldsymbol{\mu}_1 + \boldsymbol{\mu}_2\|}$ and $\boldsymbol{w}^* = \frac{2\boldsymbol{w}_1 + \boldsymbol{w}_2}{\|2\boldsymbol{w}_1 + \boldsymbol{w}_2\|}$.

## D    THE DERIVATION OF POSTERIOR

This section provides detailed derivations for Lemma 1. We begin by showing the posterior is potentially still a Gaussian mixture in Sec. D.1. Then in Sec. D.2 we show how Eq. 11 is proportion to Eq. 12, which is exactly still a Gaussian mixture.

### D.1    PRIOR TO POSTERIOR

We start by showing the posterior is potentially still a Gaussian mixture:

$$
\begin{aligned}
&P(\boldsymbol{\mu}, \boldsymbol{w} \mid \mathcal{S}_k \oplus \boldsymbol{x}_{k+1}) \\
&\propto P(\boldsymbol{\mu}, \boldsymbol{w} \mid \mathcal{S}_k \oplus \boldsymbol{x}_{k+1}) P(\mathcal{S}_k \oplus \boldsymbol{x}_{k+1}) \\
&= P(\boldsymbol{\mu}, \boldsymbol{w}, \mathcal{S}_k \oplus \boldsymbol{x}_{k+1}) \\
&= P(\boldsymbol{\mu}, \boldsymbol{w}) P(\mathcal{S}_k \oplus \boldsymbol{x}_{k+1} \mid \boldsymbol{\mu}, \boldsymbol{w}) \\
&= (\sum_{\beta=1}^{M} \pi_\beta P(\boldsymbol{\mu}, \boldsymbol{w} \mid T_\beta)) P(\mathcal{S}_k \oplus \boldsymbol{x}_{k+1} \mid \boldsymbol{\mu}, \boldsymbol{w}) \\
&= \sum_{\beta=1}^{M} \pi_\beta P(\boldsymbol{\mu}, \boldsymbol{w} \mid T_\beta) P(\mathcal{S}_k \oplus \boldsymbol{x}_{k+1} \mid \boldsymbol{\mu}, \boldsymbol{w}) \quad (11) \\
&\propto \sum_{\beta=1}^{M} \tilde{\pi}_\beta P(\boldsymbol{\mu}, \boldsymbol{w} \mid \widetilde{T}_\beta), \quad (12)
\end{aligned}
$$

### D.2    CLOSED-FORM SOLUTION FROM EQ. 11 TO EQ. 12

We analyze each component (indicated by a specific $\beta$) in Eq. 11. For all $\beta \in \{1, \ldots, M\}$ and all $(\boldsymbol{\mu}, \boldsymbol{w})$, we have:

$$
\begin{aligned}
&P(\boldsymbol{\mu}, \boldsymbol{w} \mid \widetilde{T}_\beta) P(\mathcal{S}_k \oplus \boldsymbol{x}_{k+1} \mid \boldsymbol{\mu}, \boldsymbol{w}) \\
&\propto \exp(-\frac{\|\boldsymbol{\mu}_\beta - \boldsymbol{\mu}\|^2}{2\sigma_\mu^2}) \exp(-\frac{\sum_{i=1}^{k+1} \|\boldsymbol{\mu} - \boldsymbol{x}_i\|^2}{2\sigma_x^2}) \exp(-\frac{\|\boldsymbol{w}_\beta - \boldsymbol{w}\|^2}{2\sigma_w^2}) \exp(-\frac{\sum_{i=1}^{k} \|\boldsymbol{x}_i^\top \boldsymbol{w} - y_i\|^2}{2\sigma_y^2}) \\
&(\text{let } \delta_{\boldsymbol{\mu}} = \frac{\sigma_\mu^2}{\sigma_x^2}, \delta_{\boldsymbol{w}} = \frac{\sigma_w^2}{\sigma_y^2}) \\
&= \exp(-\frac{(\|\boldsymbol{\mu}_\beta\|^2 - 2\boldsymbol{\mu}_\beta^\top \boldsymbol{\mu} + \|\boldsymbol{\mu}\|^2) + \delta_{\boldsymbol{\mu}}((k+1)\|\boldsymbol{\mu}\|^2 - 2\boldsymbol{\mu}^\top \sum_{i=1}^{k+1} \boldsymbol{x}_i + \sum_{i=1}^{k+1} \|\boldsymbol{x}_i\|^2)}{2\sigma_\mu^2}) \\
&\quad \exp(-\frac{(\|\boldsymbol{w}_\beta\|^2 - 2\boldsymbol{w}_\beta^\top \boldsymbol{w} + \|\boldsymbol{w}\|^2) + \delta_{\boldsymbol{w}}(\sum_{i=1}^{k} \boldsymbol{w}^\top \boldsymbol{x}_i \boldsymbol{x}_i^\top \boldsymbol{w} - 2\boldsymbol{w}^\top \sum_{i=1}^{k} \boldsymbol{x}_i y_i + \sum_{i=1}^{k+1} y_i^2)}{2\sigma_\mu^2}) \\
&\propto \exp(-\frac{\|\boldsymbol{\mu}_\beta\|^2 + (1 + (k+1)\delta_{\boldsymbol{\mu}})\|\boldsymbol{\mu}\|^2 - 2\boldsymbol{\mu}(\boldsymbol{\mu}_\beta + \delta_{\boldsymbol{\mu}} \sum_{i=1}^{k+1} \boldsymbol{x}_i)}{2\sigma_\mu^2}) \\
&\quad \exp(-\frac{\|\boldsymbol{w}_\beta\|^2 + \boldsymbol{w}^\top (\boldsymbol{I} + \delta_{\boldsymbol{w}} \sum_{i=1}^{k} \boldsymbol{x}_i \boldsymbol{x}_i^\top) \boldsymbol{w} - 2\boldsymbol{w}(\boldsymbol{w}_\beta + \delta_{\boldsymbol{w}} \sum_{i=1}^{k} \boldsymbol{x}_i y_i)}{2\sigma_w^2}) \\
&(\text{let } \bar{\Sigma}_{\boldsymbol{\mu}} = \boldsymbol{I}, \bar{\Sigma}_{\boldsymbol{w}} = \frac{\sum_{i=1}^{k} \boldsymbol{x}_i \boldsymbol{x}_i^\top}{k})
\end{aligned}
$$

$$= \exp(-\frac{\|\boldsymbol{\mu}_\beta\|^2 + \|\boldsymbol{\mu}\|^2_{\boldsymbol{I}+(k+1)\delta_{\boldsymbol{\mu}}\bar{\Sigma}_{\boldsymbol{\mu}}} - 2\boldsymbol{\mu}^\top(\boldsymbol{\mu}_\beta + \delta_{\boldsymbol{\mu}}\sum_{i=1}^{k+1}\boldsymbol{x}_i)}{2\sigma_{\boldsymbol{\mu}}^2})$$

$$\exp(-\frac{\|\boldsymbol{w}_\beta\|^2 + \|\boldsymbol{w}\|^2_{\boldsymbol{I}+k\delta_{\boldsymbol{w}}\bar{\Sigma}_{\boldsymbol{w}}} - 2\boldsymbol{w}^\top(\boldsymbol{w}_\beta + \delta_{\boldsymbol{w}}\sum_{i=1}^{k}\boldsymbol{x}_i y_i)}{2\sigma_w^2})$$

$$(\text{let } \bar{\boldsymbol{\mu}} = \sum_{i=1}^{k+1}\boldsymbol{x}_i, \bar{\boldsymbol{w}} = \frac{\sum_{i=1}^{k}\boldsymbol{x}_i y_i}{k})$$

$$= \exp(-\frac{\|\boldsymbol{\mu}_\beta\|^2 + \|\boldsymbol{\mu}\|^2_{\boldsymbol{I}+(k+1)\delta_{\boldsymbol{\mu}}\bar{\Sigma}_{\boldsymbol{\mu}}} - 2\boldsymbol{\mu}^\top(\boldsymbol{\mu}_\beta + (k+1)\delta_{\boldsymbol{\mu}}\bar{\boldsymbol{\mu}})}{2\sigma_{\boldsymbol{\mu}}^2})$$

$$\exp(-\frac{\|\boldsymbol{w}_\beta\|^2 + \|\boldsymbol{w}\|^2_{\boldsymbol{I}+k\delta_{\boldsymbol{w}}\bar{\Sigma}_{\boldsymbol{w}}} - 2\boldsymbol{w}^\top(\boldsymbol{w}_\beta + k\delta_{\boldsymbol{w}}\bar{\boldsymbol{w}})}{2\sigma_w^2})$$

$$(\text{Let } \Delta\bar{\boldsymbol{\mu}} = (k+1)\delta_{\boldsymbol{\mu}}\bar{\boldsymbol{\mu}}, \Delta\bar{\boldsymbol{w}} = k\delta_{\boldsymbol{w}}\bar{\boldsymbol{w}}, \Delta\bar{\Sigma}_{\boldsymbol{\mu}} = (k+1)\delta_{\boldsymbol{\mu}}\bar{\Sigma}_{\boldsymbol{\mu}}, \Delta\bar{\Sigma}_{\boldsymbol{w}} = k\delta_{\boldsymbol{w}}\bar{\Sigma}_{\boldsymbol{w}})$$

$$= \exp(-\frac{\|\boldsymbol{\mu}_\beta\|^2 + (\|\boldsymbol{\mu}\|^2_{\boldsymbol{I}+\Delta\bar{\boldsymbol{\mu}}} - 2\boldsymbol{\mu}^\top(\boldsymbol{\mu}_\beta + \Delta\bar{\boldsymbol{\mu}}) + \|\boldsymbol{\mu}_\beta + \Delta\bar{\boldsymbol{\mu}}\|^2_{(\boldsymbol{I}+\Delta\bar{\Sigma}_{\boldsymbol{\mu}})^{-1}}) - \|\boldsymbol{\mu}_\beta + \Delta\bar{\boldsymbol{\mu}}\|^2_{(\boldsymbol{I}+\Delta\bar{\Sigma}_{\boldsymbol{\mu}})^{-1}}}{2\sigma_w^2})$$

$$\exp(-\frac{\|\boldsymbol{w}_\beta\|^2 + (\|\boldsymbol{w}\|^2_{\boldsymbol{I}+\Delta\bar{\Sigma}_{\boldsymbol{w}}} - 2\boldsymbol{w}^\top(\boldsymbol{w}_\beta + \Delta\bar{\Sigma}_{\boldsymbol{w}}) + \|\boldsymbol{w}_\beta + \Delta\bar{\boldsymbol{w}}\|^2_{(\boldsymbol{I}+\Delta\bar{\Sigma}_{\boldsymbol{w}})^{-1}}) - \|\boldsymbol{w}_\beta + \Delta\bar{\boldsymbol{w}}\|^2_{(\boldsymbol{I}+\Delta\bar{\Sigma}_{\boldsymbol{w}})^{-1}}}{2\sigma_w^2})$$

$$= \exp(-\frac{\|\boldsymbol{\mu}_\beta\|^2 - \|\boldsymbol{\mu}_\beta + \Delta\bar{\boldsymbol{\mu}}\|^2_{(\boldsymbol{I}+\Delta\bar{\Sigma}_{\boldsymbol{\mu}})^{-1}}}{2\sigma_{\boldsymbol{\mu}}^2}) \cdot \exp(-\frac{\|\boldsymbol{\mu} - (\boldsymbol{I}+\Delta\bar{\Sigma}_{\boldsymbol{\mu}})^{-1}(\boldsymbol{\mu}_\beta + \Delta\bar{\boldsymbol{\mu}})\|^2_{\boldsymbol{I}+\Delta\bar{\Sigma}_{\boldsymbol{\mu}}}}{2\sigma_{\boldsymbol{\mu}}^2}) \cdot$$

$$\exp(-\frac{\|\boldsymbol{w}_\beta\|^2 - \|\boldsymbol{w}_\beta + \Delta\bar{\boldsymbol{w}}\|^2_{(\boldsymbol{I}+\Delta\bar{\Sigma}_{\boldsymbol{w}})^{-1}}}{2\sigma_w^2}) \cdot \exp(-\frac{\|\boldsymbol{w} - (\boldsymbol{I}+\Delta\bar{\Sigma}_{\boldsymbol{w}})^{-1}(\boldsymbol{w}_\beta + \Delta\bar{\boldsymbol{w}})\|^2_{\boldsymbol{I}+\Delta\bar{\Sigma}_{\boldsymbol{w}}}}{2\sigma_w^2})$$

$$\propto \exp(-\frac{\|\boldsymbol{\mu}_\beta\|^2 - \|\boldsymbol{\mu}_\beta + (k+1)\delta_{\boldsymbol{\mu}}\bar{\boldsymbol{\mu}}\|^2_{(\boldsymbol{I}+(k+1)\delta_{\boldsymbol{\mu}}\bar{\Sigma}_{\boldsymbol{\mu}})^{-1}}}{2\sigma_{\boldsymbol{\mu}}^2}) \exp(-\frac{\|\boldsymbol{w}_\beta\|^2 - \|\boldsymbol{w}_\beta + k\delta_{\boldsymbol{w}}\bar{\boldsymbol{w}}\|^2_{(\boldsymbol{I}+k\delta_{\boldsymbol{w}}\bar{\Sigma}_{\boldsymbol{w}})^{-1}}}{2\sigma_w^2}) \cdot$$

$$\mathcal{N}(\boldsymbol{\mu} \mid (\boldsymbol{I}+(k+1)\delta_{\boldsymbol{\mu}}\bar{\Sigma}_{\boldsymbol{\mu}})^{-1}(\boldsymbol{\mu}_\beta + (k+1)\delta_{\boldsymbol{\mu}}\bar{\boldsymbol{\mu}}), \sigma_{\boldsymbol{\mu}}^2(\boldsymbol{I}+(k+1)\delta_{\boldsymbol{\mu}}\bar{\Sigma}_{\boldsymbol{\mu}})^{-1}) \cdot$$

$$\mathcal{N}(\boldsymbol{w} \mid (\boldsymbol{I}+k\delta_{\boldsymbol{w}}\bar{\Sigma}_{\boldsymbol{w}})^{-1}(\boldsymbol{w}_\beta + k\delta_{\boldsymbol{w}}\bar{\boldsymbol{w}}), \sigma_{\boldsymbol{w}}^2(\boldsymbol{I}+k\delta_{\boldsymbol{w}}\bar{\Sigma}_{\boldsymbol{w}})^{-1})$$

# E   DERIVATION COLLECTION OF $\Psi_{\boldsymbol{\mu}}(\alpha, \beta)$ AND $\Psi_w(\alpha, \beta)$

This section collects derivations for $\Psi_{\boldsymbol{\mu}}(\alpha, \beta)$ and $\Psi_{\boldsymbol{w}}(\alpha, \beta)$. The derivation of $\Psi_{\boldsymbol{\mu}}(\alpha, \beta)$ is collected in Sec E.1 and the derivation of $\Psi_{\boldsymbol{w}}(\alpha, \beta)$ is collected in Sec E.2.

## E.1   DERIVATION OF $\Psi_{\boldsymbol{\mu}}(\alpha, \beta)$

This section collects the derivation of $\Psi_{\boldsymbol{\mu}}(\alpha, \beta)$ in Eq. 5 of Sec. 4.3:

$$\Psi_{\boldsymbol{\mu}}(\alpha, \beta)$$

$$= \log(\exp(-\frac{\|\boldsymbol{\mu}_\beta\|^2 - \|\boldsymbol{\mu}_\beta + (k+1)\delta_{\boldsymbol{\mu}}\bar{\boldsymbol{\mu}}\|^2_{(\boldsymbol{I}+(k+1)\delta_{\boldsymbol{\mu}}\bar{\Sigma}_{\boldsymbol{\mu}})^{-1}}}{2\sigma_{\boldsymbol{\mu}}^2}) / \exp(-\frac{\|\boldsymbol{\mu}_\alpha\|^2 - \|\boldsymbol{\mu}_\alpha + (k+1)\delta_{\boldsymbol{\mu}}\bar{\boldsymbol{\mu}}\|^2_{(\boldsymbol{I}+(k+1)\delta_{\boldsymbol{\mu}}\bar{\Sigma}_{\boldsymbol{\mu}})^{-1}}}{2\sigma_{\boldsymbol{\mu}}^2}))$$

$$= \frac{(1+(k+1)\delta_{\boldsymbol{\mu}})\|\boldsymbol{\mu}_\beta\|^2 - \|\boldsymbol{\mu}_\beta + \delta_{\boldsymbol{\mu}}\sum_{i=1}^{k+1}\boldsymbol{x}_i\|^2}{2\sigma_{\boldsymbol{\mu}}^2(1+(k+1)\delta_{\boldsymbol{\mu}})} - \frac{(1+(k+1)\delta_{\boldsymbol{\mu}})\|\boldsymbol{\mu}_\alpha\|^2 - \|\boldsymbol{\mu}_\alpha + \delta_{\boldsymbol{\mu}}\sum_{i=1}^{k+1}\boldsymbol{x}_i\|^2}{2\sigma_{\boldsymbol{\mu}}^2(1+(k+1)\delta_{\boldsymbol{\mu}})}$$

$$= \frac{-\|\boldsymbol{\mu}_\beta + \delta_{\boldsymbol{\mu}}\sum_{i=1}^{k+1}\boldsymbol{x}_i\|^2}{2\sigma_{\boldsymbol{\mu}}^2(1+(k+1)\delta_{\boldsymbol{\mu}})} - \frac{-\|\boldsymbol{\mu}_\alpha + \delta_{\boldsymbol{\mu}}\sum_{i=1}^{k+1}\boldsymbol{x}_i\|^2}{2\sigma_{\boldsymbol{\mu}}^2(1+(k+1)\delta_{\boldsymbol{\mu}})}$$

$$= \frac{-\|\boldsymbol{\mu}_\beta\|^2 - 2\boldsymbol{\mu}_\beta^\top(\delta_{\boldsymbol{\mu}}\sum_{i=1}^{k+1}\boldsymbol{x}_i) - \|\delta_{\boldsymbol{\mu}}\sum_{i=1}^{k+1}\boldsymbol{x}_i\|^2}{2\sigma_{\boldsymbol{\mu}}^2(1+(k+1)\delta_{\boldsymbol{\mu}})} - \frac{-\|\boldsymbol{\mu}_\alpha\|^2 - 2\boldsymbol{\mu}_\alpha^\top(\delta_{\boldsymbol{\mu}}\sum_{i=1}^{k+1}\boldsymbol{x}_i) - \|\delta_{\boldsymbol{\mu}}\sum_{i=1}^{k+1}\boldsymbol{x}_i\|^2}{2\sigma_{\boldsymbol{\mu}}^2(1+(k+1)\delta_{\boldsymbol{\mu}})}$$

$$= \frac{(k+1)\delta_{\boldsymbol{\mu}}\|\boldsymbol{\mu}_\beta\|^2 - 2\boldsymbol{\mu}_\beta^\top(\delta_{\boldsymbol{\mu}}\sum_{i=1}^{k+1}\boldsymbol{x}_i) + \delta_{\boldsymbol{\mu}}\sum_{i=1}^{k+1}\|\boldsymbol{x}_i\|^2}{2\sigma_{\boldsymbol{\mu}}^2(1+(k+1)\delta_{\boldsymbol{\mu}})} - \frac{(k+1)\delta_{\boldsymbol{\mu}}\|\boldsymbol{\mu}_\alpha\|^2 - 2\boldsymbol{\mu}_\alpha^\top(\delta_{\boldsymbol{\mu}}\sum_{i=1}^{k+1}\boldsymbol{x}_i) + \delta_{\boldsymbol{\mu}}\sum_{i=1}^{k+1}\|\boldsymbol{x}_i\|^2}{2\sigma_{\boldsymbol{\mu}}^2(1+(k+1)\delta_{\boldsymbol{\mu}})}$$

$$= \frac{\sum_{i=1}^{k+1} \delta_{\boldsymbol{\mu}} \|\boldsymbol{\mu}_\beta - \boldsymbol{x}_i\|^2}{2\sigma_{\boldsymbol{\mu}}^2(1 + (k+1)\delta_{\boldsymbol{\mu}})} - \frac{\sum_{i=1}^{k+1} \delta_{\boldsymbol{\mu}} \|\boldsymbol{\mu}_\alpha - \boldsymbol{x}_i\|^2}{2\sigma_{\boldsymbol{\mu}}^2(1 + (k+1)\delta_{\boldsymbol{\mu}})}$$

$$= \frac{\sum_{i=1}^{k+1} \|\boldsymbol{\mu}_\beta - \boldsymbol{x}_i\|^2 - \sum_{i=1}^{k+1} \|\boldsymbol{\mu}_\alpha - \boldsymbol{x}_i\|^2}{2\sigma_x^2(1 + (k+1)\delta_{\boldsymbol{\mu}}^2)}.$$

### E.2  DERIVATION OF $\Psi_w(\alpha, \beta)$

This section collects the derivation of $\Psi_{\boldsymbol{\mu}}(\alpha, \beta)$ in Eq. 6 of Sec. 4.3:

$$\Psi_{\boldsymbol{w}}(\alpha, \beta)$$

$$= \log(\exp(-\frac{\|\boldsymbol{w}_\alpha\|^2 - \|\boldsymbol{w}_\alpha + k\delta_{\boldsymbol{w}}\bar{\boldsymbol{w}}\|^2_{(\boldsymbol{I}+k\delta_{\boldsymbol{w}}\bar{\Sigma}_{\boldsymbol{w}})^{-1}}}{2\sigma_{\boldsymbol{w}}^2}) / \exp(-\frac{\|\boldsymbol{w}_\beta\|^2 - \|\boldsymbol{w}_\beta + k\delta_{\boldsymbol{w}}\bar{\boldsymbol{w}}\|^2_{(\boldsymbol{I}+k\delta_{\boldsymbol{w}}\bar{\Sigma}_{\boldsymbol{w}})^{-1}}}{2\sigma_{\boldsymbol{w}}^2}))$$

$$= \frac{\|\boldsymbol{w}_\beta\|^2 - \|\boldsymbol{w}_\beta + k\delta_{\boldsymbol{w}}\bar{\boldsymbol{w}}\|^2_{(\boldsymbol{I}+k\delta_{\boldsymbol{w}}\bar{\Sigma}_{\boldsymbol{w}})^{-1}}}{2\sigma_{\boldsymbol{w}}^2} - \frac{\|\boldsymbol{w}_\alpha\|^2 - \|\boldsymbol{w}_\alpha + k\delta_{\boldsymbol{w}}\bar{\boldsymbol{w}}\|^2_{(\boldsymbol{I}+k\delta_{\boldsymbol{w}}\bar{\Sigma}_{\boldsymbol{w}})^{-1}}}{2\sigma_{\boldsymbol{w}}^2}$$

$$(\text{Note } k\delta_{\boldsymbol{w}}\bar{\boldsymbol{w}} = \delta_{\boldsymbol{w}} \sum_{i=1}^{k} \boldsymbol{x}_i y_i = \delta_{\boldsymbol{w}} \sum_{i=1}^{k} \boldsymbol{x}_i \boldsymbol{x}_i^\top \boldsymbol{w}^* = k\delta_{\boldsymbol{w}}\bar{\Sigma}_{\boldsymbol{w}}\boldsymbol{w}^*)$$

$$= \frac{\|\boldsymbol{w}_\beta\|^2 - \|\boldsymbol{w}_\beta + k\delta_{\boldsymbol{w}}\bar{\Sigma}_{\boldsymbol{w}}\boldsymbol{w}^*\|^2_{(\boldsymbol{I}+k\delta_{\boldsymbol{w}}\bar{\Sigma}_{\boldsymbol{w}})^{-1}}}{2\sigma_{\boldsymbol{w}}^2} - \frac{\|\boldsymbol{w}_\alpha\| - \|\boldsymbol{w}_\alpha + k\delta_{\boldsymbol{w}}\bar{\Sigma}_{\boldsymbol{w}}\boldsymbol{w}^*\|^2_{(\boldsymbol{I}+k\delta_{\boldsymbol{w}}\bar{\Sigma}_{\boldsymbol{w}})^{-1}}}{2\sigma_{\boldsymbol{w}}^2}$$

$$= \frac{\|\boldsymbol{w}_\beta\|^2 - \|(\boldsymbol{w}_\beta - \boldsymbol{w}^*) + (\boldsymbol{I} + k\delta_{\boldsymbol{w}}\bar{\Sigma}_{\boldsymbol{w}})\boldsymbol{w}^*\|^2_{(\boldsymbol{I}+k\delta_{\boldsymbol{w}}\bar{\Sigma}_{\boldsymbol{w}})^{-1}}}{2\sigma_{\boldsymbol{w}}^2} - \frac{\|\boldsymbol{w}_\alpha\|^2 - \|(\boldsymbol{w}_\alpha - \boldsymbol{w}^*) + (\boldsymbol{I} + k\delta_{\boldsymbol{w}}\bar{\Sigma}_{\boldsymbol{w}})\boldsymbol{w}^*\|^2_{(\boldsymbol{I}+k\delta_{\boldsymbol{w}}\bar{\Sigma}_{\boldsymbol{w}})^{-1}}}{2\sigma_{\boldsymbol{w}}^2}$$

$$= \frac{\|\boldsymbol{w}_\beta\|^2 - \|\boldsymbol{w}_\beta - \boldsymbol{w}^*\|^2_{(\boldsymbol{I}+k\delta_{\boldsymbol{w}}\bar{\Sigma}_{\boldsymbol{w}})^{-1}} - 2(\boldsymbol{w}_\beta - \boldsymbol{w}^*)^\top \boldsymbol{w}^*}{2\sigma_{\boldsymbol{w}}^2} - \frac{\|\boldsymbol{w}_\alpha\|^2 - \|\boldsymbol{w}_\alpha - \boldsymbol{w}^*\|^2_{(\boldsymbol{I}+k\delta_{\boldsymbol{w}}\bar{\Sigma}_{\boldsymbol{w}})^{-1}} - 2(\boldsymbol{w}_\alpha - \boldsymbol{w}^*)^\top \boldsymbol{w}^*}{2\sigma_{\boldsymbol{w}}^2}$$

$$= \frac{\|\boldsymbol{w}_\beta - \boldsymbol{w}^*\|^2 - \|\boldsymbol{w}_\beta - \boldsymbol{w}^*\|^2_{(\boldsymbol{I}+k\delta_{\boldsymbol{w}}\bar{\Sigma}_{\boldsymbol{w}})^{-1}}}{2\sigma_{\boldsymbol{w}}^2} - \frac{\|\boldsymbol{w}_\alpha - \boldsymbol{w}^*\|^2 - \|\boldsymbol{w}_\alpha - \boldsymbol{w}^*\|^2_{(\boldsymbol{I}+k\delta_{\boldsymbol{w}}\bar{\Sigma}_{\boldsymbol{w}})^{-1}}}{2\sigma_{\boldsymbol{w}}^2}$$

$$= \frac{\|\boldsymbol{w}_\beta - \boldsymbol{w}^*\|^2_{\boldsymbol{I}-(\boldsymbol{I}+k\delta_{\boldsymbol{w}}\bar{\Sigma}_{\boldsymbol{w}})^{-1}} - \|\boldsymbol{w}_\alpha - \boldsymbol{w}^*\|^2_{\boldsymbol{I}-(\boldsymbol{I}+k\delta_{\boldsymbol{w}}\bar{\Sigma}_{\boldsymbol{w}})^{-1}}}{2\sigma_{\boldsymbol{w}}^2}$$

## F  DERIVATION COLLECTION OF $\tilde{\boldsymbol{\mu}}_\beta$ AND $\tilde{\boldsymbol{w}}_\beta$

This section collects derivations for $\tilde{\boldsymbol{\mu}}_\beta$ and $\tilde{\boldsymbol{w}}_\beta$. The derivation of $\tilde{\boldsymbol{\mu}}_\beta$ is collected in Sec F.1 and the derivation of $\Psi_{\boldsymbol{w}}$ is collected in Sec F.2.

### F.1  DERIVATION OF $\tilde{\boldsymbol{\mu}}_\beta$

This section collects the derivation of $\boldsymbol{\mu}_\beta$ in Eq. 9 of Sec. 4.3:

$$\tilde{\boldsymbol{\mu}}_\beta = (\boldsymbol{I} + (k+1)\delta_{\boldsymbol{\mu}}\bar{\Sigma}_{\boldsymbol{\mu}})^{-1}(\boldsymbol{\mu}_\beta + (k+1)\delta_{\boldsymbol{\mu}}\bar{\boldsymbol{\mu}})$$

$$= (\boldsymbol{I} + (k+1)\delta_{\boldsymbol{\mu}}\boldsymbol{I})^{-1}(\boldsymbol{\mu}_\beta + \delta_{\boldsymbol{\mu}} \sum_{i=1}^{k+1} \boldsymbol{x}_i)$$

$$= \frac{\boldsymbol{\mu}_\beta + \delta_{\boldsymbol{\mu}} \sum_{i=1}^{k+1} \boldsymbol{x}_i}{1 + (k+1)\delta_{\boldsymbol{\mu}}}$$

### F.2  DERIVATION OF $\tilde{\boldsymbol{w}}_\beta$

This section collects the derivation of $\boldsymbol{w}_\beta$ in Eq. 10 of Sec. 4.3:

$$\tilde{\boldsymbol{w}}_\beta = (\boldsymbol{I} + k\delta_{\boldsymbol{w}}\bar{\Sigma}_{\boldsymbol{w}})^{-1}(\boldsymbol{w}_\beta + k\delta_{\boldsymbol{w}}\bar{\boldsymbol{w}})$$

$$(\text{recall } k\delta_{\boldsymbol{w}}\bar{\boldsymbol{w}} = \delta_{\boldsymbol{w}} \sum_{i=1}^{k} \boldsymbol{x}_i y_i = \delta_{\boldsymbol{w}} \sum_{i=1}^{k} \boldsymbol{x}_i \boldsymbol{x}_i^\top \boldsymbol{w}^* = k\delta_{\boldsymbol{w}}\bar{\Sigma}_{\boldsymbol{w}}\boldsymbol{w}^*)$$

$$
\begin{aligned}
&= (\boldsymbol{I} + k\delta_{\boldsymbol{w}}\bar{\Sigma}_{\boldsymbol{w}})^{-1}(\boldsymbol{w}_\beta + k\delta_{\boldsymbol{w}}\bar{\Sigma}_{\boldsymbol{w}}\boldsymbol{w}^*) \\
&= (\boldsymbol{I} + k\delta_{\boldsymbol{w}}\bar{\Sigma}_{\boldsymbol{w}})^{-1}(\boldsymbol{w}_\beta - \boldsymbol{w}^* + (\boldsymbol{I} + k\delta_{\boldsymbol{w}}\bar{\Sigma}_{\boldsymbol{w}})\boldsymbol{w}^*) \\
&= (\boldsymbol{I} + k\delta_{\boldsymbol{w}}\bar{\Sigma}_{\boldsymbol{w}})^{-1}(\boldsymbol{w}_\beta - \boldsymbol{w}^*) + \boldsymbol{w}^*
\end{aligned}
\tag{13}
$$

## G  PROOF OF THEORIES FOR TWO MODES

### G.1  PROOF TOOLS

We use the following inequalities in our proofs:

#### G.1.1  GAUSSIAN TAIL BOUND

If $Z_i \sim \mathcal{N}(0,1)$, then for $t > 0$ we have:

$$
P(\frac{\sum_{i=1}^k Z_i}{k} > t) \le \exp(-\frac{kt^2}{2})
$$

$$
P(\frac{\sum_{i=1}^k Z_i}{k} < -t) \le \exp(-\frac{kt^2}{2})
$$

#### G.1.2  CHI-SQUARED TAIL BOUND

If $X \sim \chi(k)$, *i.e.*, $X = \sum_{i=1}^k Z_i^2$ where $Z_i \sim \mathcal{N}(0,1)$ then:

$$
P(\frac{X}{k} - 1 > 2\sqrt{t_1} + 2t_1) \le \exp(-kt_1^2)
$$

$$
P(\frac{X}{k} - 1 < -2\sqrt{t_1}) \le \exp(-kt_1^2)
$$

As a looser but symmetric bound, for $t > 0$ we have:

$$
P(\frac{X}{k} - 1 > t) \le \exp(-\frac{kt^2}{8})
$$

$$
P(\frac{X}{k} - 1 < -t) \le \exp(-\frac{kt^2}{8})
$$

(See Chi-square Tail Bound.)

#### G.1.3  NORM TAIL BOUND

If $\boldsymbol{\epsilon}_i \sim \mathcal{N}(\boldsymbol{0}, \tau_x^2 \boldsymbol{I})$, $\boldsymbol{\epsilon}_i \in \mathbb{R}^d$, $\boldsymbol{I} \in \mathbb{R}^{d \times d}$, then for $t > 0$ we have:

$$
P(\|\frac{\sum_{i=1}^k \boldsymbol{\epsilon}_i}{k}\| > \sqrt{\frac{\tau_x^2 d}{k}(1+t)}) \le \exp(-\frac{kt^2}{8})
$$

*Proof.*

$$
\begin{aligned}
&\|\frac{\sum_{i=1}^k \boldsymbol{\epsilon}_i}{k}\|^2 \\
&= \sum_{j=1}^d (\frac{\sum_{i=1}^k \epsilon_{i,j}}{k})^2 \\
&= \frac{\tau_x^2}{k} \sum_{j=1}^d (\frac{\sum_{i=1}^k \epsilon_{i,j}}{\tau_x \sqrt{k}})^2 \\
&\quad (\text{Notice } \epsilon_{i,j} \sim \mathcal{N}(0, \tau_x^2) \text{ and let } Z_j = \frac{\sum_{i=1}^k \epsilon_{i,j}}{\tau_x \sqrt{k}} \sim \mathcal{N}(0,1)) \\
&= \frac{\tau_x^2 d}{k} \frac{\sum_{i=1}^d Z_i^2}{d}
\end{aligned}
$$

therefore by section G.1.2 we have:

$$P(\frac{\tau_x^2 d}{k} \frac{\sum_{i=1}^d Z_i^2}{d} > \frac{\tau_x^2 d}{k}(1+t)) \leq \exp(-\frac{kt^2}{8})$$

$\square$

### G.1.4 EIGENVALUE CONCENTRATION BOUND

**Lemma 6.** *If $x_i \sim \mathcal{N}(\mu, \tau_x^2)$, $A = \frac{\sum_{i=1}^k x_i x_i^\top}{k}$, and $\frac{\sum_{i=1}^k \epsilon_i}{k} = \frac{\sum_{i=1}^k (x_i - \mu)}{k}$, we have $\forall t > 0$::*

$$P(L \leq \lambda_d(A) \leq \lambda_1(A) \leq U \text{ and } \|\frac{\sum_{i=1}^k \epsilon_i}{k}\| < \tau_x \sqrt{\gamma(1+t)}) > 1 - 3\exp(-\frac{kt^2}{8})$$

*where $L = \tau_x^2(1 - \frac{t}{2} - \gamma)^2 - 2\tau_x \gamma \sqrt{1+t}$, $U = 1 + \tau_x^2(1 + \frac{t}{2} + \gamma)^2 + 2\tau_x \gamma \sqrt{1+t}$ and $\lambda_i(A)$ is the $i^{th}$ biggest eigenvalue of the matrix $A$ and $\gamma = \sqrt{\frac{d}{k}}$.*

We begin with decomposing $A$ to three components $A = \frac{\sum_{i=1}^k x_i x_i^\top}{k} = \mu\mu^\top + \frac{\sum_{i=1}^k (\mu\epsilon_i^\top + \epsilon_i \mu^\top)}{k} + \frac{\sum_{i=1}^k \epsilon_i \epsilon_i^\top}{k}$, where $x_i = \mu + \epsilon_i$, then consider the eigenvalues of them.

Firstly, we have:

$$0 \leq \lambda_d(\mu\mu^\top) < \lambda_1(\mu\mu^\top) \leq 1$$

Then by Gaussian Case Covariance Estimation we have for $s > 0$:

$$P((1 - s - \sqrt{\frac{d}{k}})^2 \leq \frac{1}{\tau_x^2}\lambda_d(\frac{\sum_{i=1}^k \epsilon_i \epsilon_i^\top}{k}) < \frac{1}{\tau_x^2}\lambda_1(\frac{\sum_{i=1}^k \epsilon_i \epsilon_i^\top}{k}) \leq (1 + s + \sqrt{\frac{d}{k}})^2) > 1 - 2\exp(-\frac{ks^2}{2})$$

Finally we examine $\frac{\sum_{i=1}^k (\mu\epsilon_i^\top + \epsilon_i \mu^\top)}{k}$. For all $\|a\| = 1$ and $0 \leq t \leq 1$ we have:

$$|a^\top \frac{\sum_{i=1}^k (\mu\epsilon_i^\top + \epsilon_i \mu^\top)}{k} a| \leq 2\|\frac{\sum_{i=1}^k \epsilon_i}{k}\| \Longrightarrow$$

(Notice by Norm Tail Bound G.1.3, we have $P(\|\frac{\sum_{i=1}^k \epsilon_i}{k}\| > \sqrt{\frac{\tau_x^2 d}{k}(1+t)}) \leq \exp(-\frac{kt^2}{8})$)

$$P(-2\sqrt{\frac{\tau_x^2 d}{k}(1+t)} \leq \lambda_d(2\mu \frac{\sum_{i=1}^k \epsilon_i^\top}{k}) \leq \lambda_1(2\mu \frac{\sum_{i=1}^k \epsilon_i^\top}{k}) \leq 2\sqrt{\frac{\tau_x^2 d}{k}(1+t)}) > 1 - \exp(-\frac{kt^2}{8})$$

Let $\gamma = \sqrt{\frac{d}{k}}$ and $s = t/2$, we have:

$$P(\tau_x^2(1 - \frac{t}{2} - \gamma)^2 - 2\tau_x \gamma \sqrt{1+t} \leq \lambda_d(A) \leq \lambda_1(A) \leq 1 + \tau_x^2(1 + \frac{t}{2} + \gamma)^2 + 2\tau_x \gamma \sqrt{1+t}) > 1 - 3\exp(-\frac{kt^2}{8})$$

### G.2 TASK LEARNING

In this subsection, we introduce the proof of Theorem 3.

*Proof.* Assuming we are using in-context samples following Assumption 4(a), *i.e.*, $x_i \sim \mathcal{N}(\mu^*, \tau_x^2 I)$, $y_i = \langle x_i, w^* \rangle$, and we aim to have the prediction on $\mathcal{S}_k \oplus x_{k+1}$ as $\langle x_{k+1}, w^* \rangle$, *i.e.*, to learn the prediction of the demonstration task. Let $\mathcal{L}_k^{\text{L}}$ indicate the squared loss $(\mathcal{F}^*(\mathcal{S}_k \oplus x_{k+1}) - \langle x_{k+1}, w^* \rangle)^2$ on $\mathcal{S}_k \oplus x_{k+1}$. With the help of Lemma 1 and Corollary 2, we can derive the expected

squared loss on the prediction $\mathcal{F}^*(\mathcal{S}_k \oplus \boldsymbol{x}_{k+1})$ as follows:

$$\mathbb{E}_{\mathcal{S}_k \oplus \boldsymbol{x}_{k+1}}[\mathcal{L}_k^{\mathrm{L}}]$$
$$=\mathbb{E}_{\mathcal{S}_K}[(\mathcal{F}^*(\mathcal{S}_k \oplus \boldsymbol{x}_{k+1}) - \langle \boldsymbol{w}^*, \boldsymbol{x}_{k+1}\rangle)^2]$$
$$=\mathbb{E}_{\mathcal{S}_K}[(\sum_{\beta=1}^{M} \tilde{\pi}_\beta \langle \tilde{\boldsymbol{w}}_\beta, \boldsymbol{x}_{k+1}\rangle - \langle \boldsymbol{w}^*, \boldsymbol{x}_{k+1}\rangle)^2]$$
$$=\mathbb{E}_{\mathcal{S}_K}[(\langle \sum_{\beta=1}^{M} \tilde{\pi}_\beta (\tilde{\boldsymbol{w}}_\beta - \boldsymbol{w}^*), \boldsymbol{x}_{k+1}\rangle)^2]$$

(See Eq. 13 for the derivation of $\tilde{\boldsymbol{w}}_\beta$)

$$=\mathbb{E}_{\mathcal{S}_K}[(\langle \sum_{\beta=1}^{M} \tilde{\pi}_\beta ((\boldsymbol{I} + k\delta_{\boldsymbol{w}} \bar{\Sigma}_{\boldsymbol{w}})^{-1}(\boldsymbol{w}_\beta - \boldsymbol{w}^*) + \boldsymbol{w}^* - \boldsymbol{w}^*), \boldsymbol{x}_{k+1}\rangle)^2]$$

(Let $\Delta\bar{\Sigma}_{\boldsymbol{w}} = k\delta_{\boldsymbol{w}}\bar{\Sigma}_{\boldsymbol{w}}$)

$$=\mathbb{E}_{\mathcal{S}_K}[(\langle (\boldsymbol{I} + \Delta\bar{\Sigma}_{\boldsymbol{w}})^{-1} \sum_{\beta=1}^{M} \tilde{\pi}_\beta (\boldsymbol{w}_\beta - \boldsymbol{w}^*), \boldsymbol{x}_{k+1}\rangle)^2]$$
$$=\mathbb{E}_{\mathcal{S}_K}[(\langle (\boldsymbol{I} + \Delta\bar{\Sigma}_{\boldsymbol{w}})^{-1} (\sum_{\beta=1}^{M} \tilde{\pi}_\beta \boldsymbol{w}_\beta - \boldsymbol{w}^*), \boldsymbol{x}_{k+1}\rangle)^2]$$

(Let $\boldsymbol{A} = (\boldsymbol{I} + \Delta\bar{\Sigma}_{\boldsymbol{w}})^{-1}, \boldsymbol{b} = \sum_{\beta=1}^{M} \tilde{\pi}_\beta \boldsymbol{w}_\beta - \boldsymbol{w}^*.$)

$$=\mathbb{E}_{\mathcal{S}_K}[(\boldsymbol{b}^\top \boldsymbol{A} \boldsymbol{x}_{k+1})^2]$$

($\boldsymbol{A}$ is a random matrix only depending on $\mathcal{S}_k$, while $\boldsymbol{b}$ is a random vector depending on both $\mathcal{S}_k$ and $\boldsymbol{x}_{k+1}$.)

$$\leq\mathbb{E}_{\mathcal{S}_K}[(\|\boldsymbol{b}\|\lambda_1(\boldsymbol{A})\|\boldsymbol{x}_{k+1}\|)^2]$$

(Notice $\|\boldsymbol{b}\| \leq 2$)

$$\leq\mathbb{E}_{\mathcal{S}_K}[2^2 \lambda_1^2(\boldsymbol{A})\|\boldsymbol{x}_{k+1}\|^2]$$
$$\leq 4\mathbb{E}_{\mathcal{S}_K}[\lambda_1^2(\boldsymbol{A})]\mathbb{E}_{\boldsymbol{x}_{k+1}}[\|\boldsymbol{x}_{k+1}\|^2]$$
$$=4(1 + d\tau_x^2)\mathbb{E}_{\mathcal{S}_K}[\lambda_1^2(\boldsymbol{A})]$$

Apply Lemma 6, we have the upper bound on the expected loss we have:

$$\mathbb{E}_{\mathcal{S}_k \oplus \boldsymbol{x}_{k+1}}[\mathcal{L}_k^{\mathrm{L}}]$$
$$<4(1 + d\tau_x^2)\mathbb{E}_{\mathcal{S}_K}[\lambda_1^2(\boldsymbol{A})]$$
$$<4(1 + d\tau_x^2)\mathbb{E}_{\mathcal{S}_K}[(\frac{1}{1 + k\delta_{\boldsymbol{w}}\lambda_d(\frac{\sum_{i=1}^{k} \boldsymbol{x}_i \boldsymbol{x}_i^\top}{k})})^2]$$
$$<4(1 + d\tau_x^2)((\frac{1}{1 + k\delta_{\boldsymbol{w}}(\tau_x^2(1 - \frac{t}{2} - \gamma)^2 - 2\tau_x\gamma\sqrt{1+t})})^2 + 3\exp(-\frac{kt^2}{8}))$$

Let $t = k^{\delta - \frac{1}{2}}$, where $\frac{1}{2} > \delta > 0$ we have:

$$\mathbb{E}_{\mathcal{S}_k \oplus \boldsymbol{x}_{k+1}}[\mathcal{L}_k^{\mathrm{L}}] < \frac{4(1 + d\tau_x^2)}{\tau_x^4 \delta_{\boldsymbol{w}}^2 k^2} + O(k^{\delta - \frac{5}{2}})$$

$\square$

## G.3 TASK RETRIEVAL

In this subsection, we introduce the proof of Theorem 4.

*Proof.* Assuming we are using demonstrations following Assumption 4(a), *i.e.*, $\boldsymbol{x}_i \sim \mathcal{N}(\boldsymbol{\mu}^*, \tau_x^2 \boldsymbol{I}), y_i = \langle \boldsymbol{x}_i, \boldsymbol{w}^*\rangle$, and we aim to have the prediction on $\mathcal{S}_k \oplus \boldsymbol{x}_{k+1}$ as $\langle \boldsymbol{x}_{k+1}, \boldsymbol{w}_\alpha\rangle$, *i.e.*, to

retrieve the prediction of the clean task $\alpha$. In order to have an upper bound on the loss, we consider $\boldsymbol{x}_i \sim \mathcal{N}(\boldsymbol{\mu}^*, \tau_x^2 \boldsymbol{I})$ in two regions: (1) $\mathbf{C}$: $L < \lambda_d(\frac{\sum_{i=1}^{k} \boldsymbol{x}_i \boldsymbol{x}_i^\top}{k}) \leq \lambda_1(\frac{\sum_{i=1}^{k} \boldsymbol{x}_i \boldsymbol{x}_i^\top}{k}) < U$ (see Lemma 6 for L and U) and (2) $\neg \mathbf{C}$: either the previous inequality does not hold. The probability of $\neg \mathbf{C}$ is bounded by:

$$P(\neg \mathbf{C}) < 3 \exp(-\frac{kt^2}{8}).$$

Let $\mathcal{L}_k^{\mathrm{R}}$ indicate the squared loss $(\mathcal{F}^*(\mathcal{S}_k \oplus \boldsymbol{x}_{k+1}) - \langle \boldsymbol{x}_{k+1}, \boldsymbol{w}_\alpha \rangle)^2$ on $\mathcal{S}_k \oplus \boldsymbol{x}_{k+1}$. With the help of Lemma 1 and Corollary 2, we can derive the expected squared loss on the prediction $\mathcal{F}^*(\mathcal{S}_k \oplus \boldsymbol{x}_{k+1})$, and then based on $\mathbf{C}$ and the target task $\alpha$, we split the expected squared loss into three parts:

$$\mathbb{E}_{\mathcal{S}_k \oplus \boldsymbol{x}_{k+1}}[\mathcal{L}_k^{\mathrm{R}}]$$

$$= \mathbb{E}_{\mathcal{S}_K}[(\sum_{\beta=1}^{M} \tilde{\pi}_\beta \langle \tilde{\boldsymbol{w}}_\beta, \boldsymbol{x}_{k+1} \rangle - \langle \boldsymbol{w}_\alpha, \boldsymbol{x}_{k+1} \rangle)^2]$$

$$(\text{Notice } \sum_{\beta=1}^{M} \pi_\beta = 1)$$

$$= \mathbb{E}_{\mathcal{S}_K}[(\sum_{\beta=1}^{M} \tilde{\pi}_\beta(\langle \tilde{\boldsymbol{w}}_\beta, \boldsymbol{x}_{k+1} \rangle - \langle \boldsymbol{w}_\alpha, \boldsymbol{x}_{k+1} \rangle))^2]$$

$$(\text{Notice } (\sum_{\beta=1}^{M} \tilde{\pi}_\beta a_\beta)^2 \leq \sum_{\beta=1}^{M} \tilde{\pi}_\beta a_\beta^2, \text{ since } \mathbb{E}[a]^2 \leq \mathbb{E}[a^2])$$

$$\leq \mathbb{E}_{\mathcal{S}_K}[\sum_{\beta=1}^{M} \tilde{\pi}_\beta(\langle \tilde{\boldsymbol{w}}_\beta, \boldsymbol{x}_{k+1} \rangle - \langle \boldsymbol{w}_\alpha, \boldsymbol{x}_{k+1} \rangle)^2]$$

$$= \mathbb{E}_{\mathcal{S}_K}[\sum_{\beta=1}^{M} \tilde{\pi}_\beta(\langle \tilde{\boldsymbol{w}}_\beta - \boldsymbol{w}_\alpha, \boldsymbol{x}_{k+1} \rangle)^2]$$

$$= P(\mathbf{C}) \mathbb{E}_{\mathcal{S}_K}[\sum_{\beta=1}^{M} \tilde{\pi}_\beta(\langle \tilde{\boldsymbol{w}}_\beta - \boldsymbol{w}_\alpha, \boldsymbol{x}_{k+1} \rangle)^2 \mid \mathbf{C}] +$$

$$P(\neg \mathbf{C}) \mathbb{E}_{\mathcal{S}_K}[\sum_{\beta=1}^{M} \tilde{\pi}_\beta(\langle \tilde{\boldsymbol{w}}_\beta - \boldsymbol{w}_\alpha, \boldsymbol{x}_{k+1} \rangle)^2 \mid \neg \mathbf{C}]$$

$$= P(\mathbf{C}) \mathbb{E}_{\mathcal{S}_K}[\sum_{\beta \neq \alpha} \tilde{\pi}_\beta(\langle \tilde{\boldsymbol{w}}_\beta - \boldsymbol{w}_\alpha, \boldsymbol{x}_{k+1} \rangle)^2 \mid \mathbf{C}] + \tag{14}$$

$$P(\mathbf{C}) \mathbb{E}_{\mathcal{S}_K}[\tilde{\pi}_\alpha(\langle \tilde{\boldsymbol{w}}_\alpha - \boldsymbol{w}_\alpha, \boldsymbol{x}_{k+1} \rangle)^2 \mid \mathbf{C}] + \tag{15}$$

$$P(\neg \mathbf{C}) \mathbb{E}_{\mathcal{S}_K}[\sum_{\beta=1}^{M} \tilde{\pi}_\beta(\langle \tilde{\boldsymbol{w}}_\beta - \boldsymbol{w}_\alpha, \boldsymbol{x}_{k+1} \rangle)^2 \mid \neg \mathbf{C}] \tag{16}$$

We firstly analyze the first term $P(\mathbf{C}) \mathbb{E}_{\mathcal{S}_K}[\sum_{\beta \neq \alpha} \tilde{\pi}_\beta(\langle \tilde{\boldsymbol{w}}_\beta - \boldsymbol{w}_\alpha, \boldsymbol{x}_{k+1} \rangle)^2 \mid \mathbf{C}]$ in Part. 14, we have:

$$P(\mathbf{C}) \mathbb{E}_{\mathcal{S}_K}[\sum_{\beta \neq \alpha} \tilde{\pi}_\beta(\langle \tilde{\boldsymbol{w}}_\beta - \boldsymbol{w}_\alpha, \boldsymbol{x}_{k+1} \rangle)^2 \mid \mathbf{C}]$$

$$< P(\mathbf{C}) \mathbb{E}_{\mathcal{S}_K}[\sum_{\beta \neq \alpha} \frac{\tilde{\pi}_\beta}{\tilde{\pi}_\beta + \tilde{\pi}_\beta}(\langle \tilde{\boldsymbol{w}}_\beta - \boldsymbol{w}_\alpha, \boldsymbol{x}_{k+1} \rangle)^2 \mid \mathbf{C}]$$

$$= P(\mathbf{C}) \mathbb{E}_{\mathcal{S}_K}[\sum_{\beta \neq \alpha} \frac{\tilde{\pi}_\beta / \tilde{\pi}_\alpha}{1 + \tilde{\pi}_\beta / \tilde{\pi}_\alpha}(\langle \tilde{\boldsymbol{w}}_\beta - \boldsymbol{w}_\alpha, \boldsymbol{x}_{k+1} \rangle)^2 \mid \mathbf{C}]$$

$$< P(\mathbf{C}) \mathbb{E}_{\mathcal{S}_K}[\sum_{\beta \neq \alpha} \frac{\tilde{\pi}_\beta}{\tilde{\pi}_\alpha}(\langle \tilde{\boldsymbol{w}}_\beta - \boldsymbol{w}_\alpha, \boldsymbol{x}_{k+1} \rangle)^2 \mid \mathbf{C}]$$

$$< P(\mathbf{C}) \mathbb{E}_{\mathcal{S}_K} [\sum_{\beta \neq \alpha} \frac{\tilde{\pi}_\beta}{\tilde{\pi}_\alpha} \|\tilde{\boldsymbol{w}}_\beta - \boldsymbol{w}_\alpha\|^2 \|\boldsymbol{x}_{k+1}\|^2 \mid \mathbf{C}]$$

$$< P(\mathbf{C}) \mathbb{E}_{\mathcal{S}_K} [\sum_{\beta \neq \alpha} \frac{\tilde{\pi}_\beta}{\tilde{\pi}_\alpha} \|\tilde{\boldsymbol{w}}_\beta - \boldsymbol{w}_\alpha\|^2 \|\boldsymbol{x}_{k+1}\|^2 \mid \mathbf{C}]$$

(See Eq. 13 for the derivation of $\tilde{\boldsymbol{w}}_\beta$)

$$= P(\mathbf{C}) \mathbb{E}_{\mathcal{S}_K} [\sum_{\beta \neq \alpha} \frac{\tilde{\pi}_\beta}{\tilde{\pi}_\alpha} \|(\boldsymbol{I} + k\delta_{\boldsymbol{w}} \bar{\Sigma}_{\boldsymbol{w}})^{-1}(\boldsymbol{w}_\beta - \boldsymbol{w}^*) + \boldsymbol{w}^* - \boldsymbol{w}_\alpha\|^2 \|\boldsymbol{x}_{k+1}\|^2 \mid \mathbf{C}]$$

(Let $\Delta\bar{\Sigma}_{\boldsymbol{w}} = k\delta_{\boldsymbol{w}} \bar{\Sigma}_{\boldsymbol{w}}$)

$$= P(\mathbf{C}) \mathbb{E}_{\mathcal{S}_K} [\sum_{\beta \neq \alpha} \frac{\tilde{\pi}_\beta}{\tilde{\pi}_\alpha} \|(\boldsymbol{I} + \Delta\bar{\Sigma}_{\boldsymbol{w}})^{-1}(\boldsymbol{w}_\beta - \boldsymbol{w}^*) + \boldsymbol{w}^* - \boldsymbol{w}_\beta + \boldsymbol{w}_\beta - \boldsymbol{w}_\alpha\|^2 \|\boldsymbol{x}_{k+1}\|^2 \mid \mathbf{C}]$$

$$= P(\mathbf{C}) \mathbb{E}_{\mathcal{S}_K} [\sum_{\beta \neq \alpha} \frac{\tilde{\pi}_\beta}{\tilde{\pi}_\alpha} \| - (\boldsymbol{I} - (\boldsymbol{I} + \Delta\bar{\Sigma}_{\boldsymbol{w}})^{-1})(\boldsymbol{w}_\beta - \boldsymbol{w}^*) + (\boldsymbol{w}_\beta - \boldsymbol{w}_\alpha)\|^2 \|\boldsymbol{x}_{k+1}\|^2 \mid \mathbf{C}]$$

$$= P(\mathbf{C}) \mathbb{E}_{\mathcal{S}_K} [\sum_{\beta \neq \alpha} \frac{\tilde{\pi}_\beta}{\tilde{\pi}_\alpha} (2\|(\boldsymbol{I} - (\boldsymbol{I} + \Delta\bar{\Sigma}_{\boldsymbol{w}})^{-1})(\boldsymbol{w}_\beta - \boldsymbol{w}^*)\|^2 + 2\|(\boldsymbol{w}_\beta - \boldsymbol{w}_\alpha)\|^2) \|\boldsymbol{x}_{k+1}\|^2 \mid \mathbf{C}]$$

$$< 2 P(\mathbf{C}) \mathbb{E}_{\mathcal{S}_K} [\sum_{\beta \neq \alpha} \frac{\tilde{\pi}_\beta}{\tilde{\pi}_\alpha} (\lambda_1^2 (\boldsymbol{I} - (\boldsymbol{I} + \Delta\bar{\Sigma}_{\boldsymbol{w}})^{-1}) \|(\boldsymbol{w}_\beta - \boldsymbol{w}^*)\|^2 + \|(\boldsymbol{w}_\beta - \boldsymbol{w}_\alpha)\|^2) \|\boldsymbol{x}_{k+1}\|^2 \mid \mathbf{C}]$$

(where $\lambda_1(\boldsymbol{A})$ is the largest eigenvalue of matrix $\boldsymbol{A}$)

$$< 2 P(\mathbf{C}) \mathbb{E}_{\mathcal{S}_K} [\sum_{\beta \neq \alpha} \frac{\tilde{\pi}_\beta}{\tilde{\pi}_\alpha} (1^2 \cdot 4 + 4) \|\boldsymbol{x}_{k+1}\|^2 \mid \mathbf{C}]$$

$$< 16 P(\mathbf{C}) \mathbb{E}_{\mathcal{S}_K} [\sum_{\beta \neq \alpha} \frac{\tilde{\pi}_\beta}{\tilde{\pi}_\alpha} \|\boldsymbol{x}_{k+1}\|^2 \mid \mathbf{C}]$$

Apply Eqs. 4, 5, and 6 and Assumption 3(b) to $\frac{\tilde{\pi}_\beta}{\tilde{\pi}_\alpha}$, we have:

$$\mathbb{E}_{\mathcal{S}_K} [\sum_{\beta \neq \alpha} \frac{\tilde{\pi}_\beta}{\tilde{\pi}_\alpha} \|\boldsymbol{x}_{k+1}\|^2 \mid \mathbf{C}]$$

$$< 16 \mathbb{E}_{\mathcal{S}_K} [\sum_{\beta \neq \alpha} r \exp(\frac{-\sum_{i=1}^k \|\boldsymbol{\mu}_\beta - \boldsymbol{x}_i\|^2 + \sum_{i=1}^k \|\boldsymbol{\mu}_\alpha - \boldsymbol{x}_i\|^2}{2\sigma_x^2 (1 + (k+1)\delta_{\boldsymbol{\mu}})}) \cdot$$

$$\exp(\frac{-\|\boldsymbol{w}_\beta - \boldsymbol{w}^*\|_{\boldsymbol{I} - (\boldsymbol{I} + \Delta\bar{\Sigma}_w)^{-1}}^2 + \|\boldsymbol{w}_\alpha - \boldsymbol{w}^*\|_{\boldsymbol{I} - (\boldsymbol{I} + \Delta\bar{\Sigma}_w)^{-1}}^2}{2\sigma_w^2}) \cdot$$

$$\exp(\frac{-\|\boldsymbol{\mu}_\beta - \boldsymbol{x}_{k+1}\|^2 + \|\boldsymbol{\mu}_\alpha - \boldsymbol{x}_{k+1}\|^2}{2\sigma_x^2 (1 + (k+1)\delta_{\boldsymbol{\mu}})}) \|\boldsymbol{x}_{k+1}\|^2 \mid \mathbf{C}]$$

$$= 16 r \sum_{\beta \neq \alpha} \mathbb{E}_{\mathcal{S}_K} [\exp(\frac{-\sum_{i=1}^k \|\boldsymbol{\mu}_\beta - \boldsymbol{x}_i\|^2 + \sum_{i=1}^k \|\boldsymbol{\mu}_\alpha - \boldsymbol{x}_i\|^2}{2\sigma_x^2 (1 + (k+1)\delta_{\boldsymbol{\mu}})}) \cdot$$

$$\exp(\frac{-\|\boldsymbol{w}_\beta - \boldsymbol{w}^*\|_{\boldsymbol{I} - (\boldsymbol{I} + \Delta\bar{\Sigma}_w)^{-1}}^2 + \|\boldsymbol{w}_\alpha - \boldsymbol{w}^*\|_{\boldsymbol{I} - (\boldsymbol{I} + \Delta\bar{\Sigma}_w)^{-1}}^2}{2\sigma_w^2}) \cdot$$

$$\exp(\frac{-\|\boldsymbol{\mu}_\beta - \boldsymbol{x}_i\|^2 + \|\boldsymbol{\mu}_\alpha - \boldsymbol{x}_i\|^2}{2\sigma_x^2 (1 + (k+1)\delta_{\boldsymbol{\mu}})}) \|\boldsymbol{x}_{k+1}\|^2 \mid \mathbf{C}]$$

Recall in case $\mathbf{C}$ we have:

$$\|\frac{\sum_{i=1}^k \boldsymbol{\epsilon}_i}{k}\| < \tau_x \gamma \sqrt{1+t}$$

Therefore, when conditioned on case **C** we have:

$$\frac{\sum_{i=1}^{k}(-\|\boldsymbol{\mu}_\beta - \boldsymbol{x}_i\|^2 + \|\boldsymbol{\mu}_\alpha - \boldsymbol{x}_i\|^2)}{1 + (k+1)\delta_{\boldsymbol{\mu}}}$$

(Let $\boldsymbol{x}_i = \boldsymbol{\mu}^* + \boldsymbol{\epsilon}_i$)

$$= k\frac{\|\boldsymbol{\mu}_\alpha - \boldsymbol{\mu}^*\|^2 - \|\boldsymbol{\mu}_\beta - \boldsymbol{\mu}^*\|^2 + \frac{\sum_{i=1}^{k} 2\langle \boldsymbol{\mu}_\beta - \boldsymbol{\mu}_\alpha, \boldsymbol{\epsilon}_i \rangle}{k}}{1 + (k+1)\delta_{\boldsymbol{\mu}}}$$

$$= k\frac{\|\boldsymbol{\mu}_\alpha - \boldsymbol{\mu}^*\|^2 - \|\boldsymbol{\mu}_\beta - \boldsymbol{\mu}^*\|^2 + \langle 2(\boldsymbol{\mu}_\beta - \boldsymbol{\mu}_\alpha), \frac{\sum_{i=1}^{k} \boldsymbol{\epsilon}_i}{k} \rangle}{1 + (k+1)\delta_{\boldsymbol{\mu}}}$$

$$\leq k\frac{\|\boldsymbol{\mu}_\alpha - \boldsymbol{\mu}^*\|^2 - \|\boldsymbol{\mu}_\beta - \boldsymbol{\mu}^*\|^2 + 4\tau_x\gamma\sqrt{1+t}}{1 + (k+1)\delta_{\boldsymbol{\mu}}}$$

(Branch to purple for asymptotic bound or to orange for the bound for the U-shaped pattern.)

(Let $t = k^{\delta - \frac{1}{2}}$ and $\delta$ is small.)

$$= -\frac{d_{\boldsymbol{\mu}}^2}{\delta_{\boldsymbol{\mu}}} + \frac{4\tau_x\sqrt{d}}{\delta_{\boldsymbol{\mu}}}k^{\frac{\delta}{2} - \frac{3}{4}} + O(k^{-1})$$

(let $t = k^{-\frac{1}{4}}$, When $\delta_{\boldsymbol{\mu}} \ll 1$, such that $\exists k \leq \frac{1}{\delta_{\boldsymbol{\mu}}}$, s.t. $\frac{d_{\boldsymbol{\mu}}^2}{2} > 4\tau_x\gamma\sqrt{1 + k^{-\frac{1}{4}}}$)

$$< -\frac{d_{\boldsymbol{\mu}}^2}{4}$$

Recall in case **C** we have:

$$\mathrm{L} < \lambda_d\left(\frac{\sum_{i=1}^{k} \boldsymbol{x}_i\boldsymbol{x}_i^\top}{k}\right) < \lambda_1\left(\frac{\sum_{i=1}^{k} \boldsymbol{x}_i\boldsymbol{x}_i^\top}{k}\right) < \mathrm{U}$$

Therefore when conditioned on case **C** we also have:

$$-\|\boldsymbol{w}_\beta - \boldsymbol{w}^*\|_{\boldsymbol{I} - (\boldsymbol{I} + \Delta\bar{\Sigma}_{\boldsymbol{w}})^{-1}}^2 + \|\boldsymbol{w}_\alpha - \boldsymbol{w}^*\|_{\boldsymbol{I} - (\boldsymbol{I} + \Delta\bar{\Sigma}_{\boldsymbol{w}})^{-1}}^2$$

$$< -\|\boldsymbol{w}_\beta - \boldsymbol{w}^*\|^2 \lambda_d(\boldsymbol{I} - (\boldsymbol{I} + \Delta\bar{\Sigma}_{\boldsymbol{w}})^{-1}) + \|\boldsymbol{w}_\alpha - \boldsymbol{w}^*\|^2 \lambda_1(\boldsymbol{I} - (\boldsymbol{I} + \Delta\bar{\Sigma}_{\boldsymbol{w}})^{-1})$$

(where $\lambda_1(\boldsymbol{A})$ and $\lambda_d(\boldsymbol{A})$ indicate the maximal and minimal eigenvalues of the matrix $\boldsymbol{A} \in \mathbb{R}^{d \times d}$)

$$< -\|\boldsymbol{w}_\beta - \boldsymbol{w}^*\|^2\left(1 - \frac{1}{1 + k\delta_{\boldsymbol{w}}\mathrm{L}}\right) + \|\boldsymbol{w}_\alpha - \boldsymbol{w}^*\|^2\left(1 - \frac{1}{1 + k\delta_{\boldsymbol{w}}\mathrm{U}}\right)$$

(Branch to purple for asymptotic bound or to orange for the bound for the U-shaped pattern.)

$$= (-\|\boldsymbol{w}_\beta - \boldsymbol{w}^*\|^2 + \|\boldsymbol{w}_\alpha - \boldsymbol{w}^*\|^2) + \left(+\frac{\|\boldsymbol{w}_\beta - \boldsymbol{w}^*\|^2}{1 + k\delta_{\boldsymbol{w}}\mathrm{L}} - \frac{\|\boldsymbol{w}_\alpha - \boldsymbol{w}^*\|^2}{1 + k\delta_{\boldsymbol{w}}\mathrm{U}}\right)$$

(Let $t = k^{\delta - \frac{1}{2}}$ and $\delta$ is small.)

$$= -(\|\boldsymbol{w}_\beta - \boldsymbol{w}^*\|^2 - \|\boldsymbol{w}_\alpha - \boldsymbol{w}^*\|^2) + \left(\frac{\|\boldsymbol{w}_\beta - \boldsymbol{w}^*\|^2}{k\delta_{\boldsymbol{w}}\tau_x^2} - \frac{\|\boldsymbol{w}_\alpha - \boldsymbol{w}^*\|^2}{k\delta_{\boldsymbol{w}}(1 + \tau_x^2)}\right) + O(k^{\delta - \frac{3}{2}})$$

$$< -(d_{\boldsymbol{w}}^2) + \frac{4}{\delta_{\boldsymbol{w}}\tau_x^2}k^{-1} + O(k^{\delta - \frac{3}{2}})$$

$$= -\|\boldsymbol{w}_\beta - \boldsymbol{w}^*\|^2 \frac{k\delta_{\boldsymbol{w}}\mathrm{L}}{1 + k\delta_{\boldsymbol{w}}\mathrm{L}} + \|\boldsymbol{w}_\alpha - \boldsymbol{w}^*\|^2 \frac{k\delta_{\boldsymbol{w}}\mathrm{U}}{1 + k\delta_{\boldsymbol{w}}\mathrm{U}}$$

$$< -\|\boldsymbol{w}_\beta - \boldsymbol{w}^*\|^2 \frac{k\delta_{\boldsymbol{w}}\mathrm{L}}{1 + k\delta_{\boldsymbol{w}}\tau_x^2} + \|\boldsymbol{w}_\alpha - \boldsymbol{w}^*\|^2 \frac{k\delta_{\boldsymbol{w}}\mathrm{U}}{1 + k\delta_{\boldsymbol{w}}\tau_x^2}$$

(let $t = k^{-\frac{1}{4}}$, When $\delta_{\boldsymbol{w}} \ll 1$, such that $\exists k \leq \frac{1}{\delta_{\boldsymbol{w}}\tau_x^2}$, s.t. $\mathrm{L}\|\boldsymbol{w}_\beta - \boldsymbol{w}^*\|^2 - \mathrm{U}\|\boldsymbol{w}_\alpha - \boldsymbol{w}^*\|^2 > \tau_x^2 u_{\boldsymbol{w}}^2/2$)

$$< -k\delta_{\boldsymbol{w}}\frac{\tau_x^2 u_{\boldsymbol{w}}^2}{4}$$

Further, we have:

$$P(\mathbf{C})\mathbb{E}_{\mathcal{S}_K}[\exp(\frac{-\|\boldsymbol{\mu}_\beta - \boldsymbol{x}_{k+1}\|^2 + \|\boldsymbol{\mu}_\alpha - \boldsymbol{x}_{k+1}\|^2}{2\sigma_x^2(1 + (k+1)\delta_{\boldsymbol{\mu}})})\|\boldsymbol{x}_{k+1}\|^2 \mid \mathbf{C}]$$

$$< \mathbb{E}_{\mathcal{S}_K}[\exp(\frac{-\|\boldsymbol{\mu}_\beta - \boldsymbol{x}_{k+1}\|^2 + \|\boldsymbol{\mu}_\alpha - \boldsymbol{x}_{k+1}\|^2}{2\sigma_x^2(1 + (k+1)\delta_{\boldsymbol{\mu}})})\|\boldsymbol{x}_{k+1}\|^2]$$

$$(\text{Let } \boldsymbol{x}_{k+1} = \boldsymbol{\mu}^* + \boldsymbol{\epsilon})$$

$$= \mathbb{E}_{\mathcal{S}_K}[\exp(\frac{-\|\boldsymbol{\mu}_\beta - \boldsymbol{\mu}^* - \boldsymbol{\epsilon}\|^2 + \|\boldsymbol{\mu}_\alpha - \boldsymbol{\mu}^* - \boldsymbol{\epsilon}\|^2}{2\sigma_x^2(1 + (k+1)\delta_{\boldsymbol{\mu}})})\|\boldsymbol{x}_{k+1}\|^2]$$

$$= \mathbb{E}_{\mathcal{S}_K}[\exp(\frac{-\|\boldsymbol{\mu}_\beta - \boldsymbol{\mu}^*\|^2 + \|\boldsymbol{\mu}_\alpha - \boldsymbol{\mu}^*\|^2 + \langle 2(\boldsymbol{\mu}_\beta - \boldsymbol{\mu}_\alpha), \boldsymbol{\epsilon}\rangle}{2\sigma_x^2(1 + (k+1)\delta_{\boldsymbol{\mu}})})\|\boldsymbol{x}_{k+1}\|^2]$$

$$(\text{Let } -\|\boldsymbol{\mu}_\beta - \boldsymbol{\mu}^*\|^2 + \|\boldsymbol{\mu}_\alpha - \boldsymbol{\mu}^*\|^2 = -D, 2\sigma_x^2(1 + (k+1)\delta_{\boldsymbol{\mu}}) = E, \boldsymbol{b} = 2(\boldsymbol{\mu}_\beta - \boldsymbol{\mu}_\alpha))$$

$$= \mathbb{E}_{\mathcal{S}_K}[\exp(\frac{-D + \boldsymbol{b}^\top \boldsymbol{\epsilon}}{E})\|\boldsymbol{x}_{k+1}\|^2]$$

$$\leq \mathbb{E}_{\mathcal{S}_K}[\exp(\frac{-D + \boldsymbol{b}^\top \boldsymbol{\epsilon}}{E})(2\|\boldsymbol{\mu}^*\|^2 + 2\|\boldsymbol{\epsilon}\|^2)]$$

$$= 2(\mathbb{E}_{\mathcal{S}_K}[\exp(\frac{-D + \boldsymbol{b}^\top \boldsymbol{\epsilon}}{E})] + \mathbb{E}_{\mathcal{S}_K}[\exp(\frac{-D + \boldsymbol{b}^\top \boldsymbol{\epsilon}}{E})\|\boldsymbol{\epsilon}\|^2])$$

$$= 2(\exp(\frac{\tau_x^2\|\boldsymbol{b}\|^2}{2E^2} - \frac{D}{E}) + \mathbb{E}_{\mathcal{S}_K}[\exp(\frac{-D + \boldsymbol{b}^\top \boldsymbol{\epsilon}}{E})\|\boldsymbol{\epsilon}\|^2])$$

$$= 2(\exp(\frac{\tau_x^2\|\boldsymbol{b}\|^2}{2E^2} - \frac{D}{E}) + \tau_x^2(1 + \frac{\tau_x^2\|\boldsymbol{b}\|^2}{E^2})\exp(\frac{\tau_x^2\|\boldsymbol{b}\|^2}{2E^2} - \frac{D}{E}) + (d-1)\tau_x^2\exp(\frac{\tau_x^2\|\boldsymbol{b}\|^2}{2E^2} - \frac{D}{E}))$$

$$= 2(1 + \tau_x^2(d + \frac{\tau_x^2\|\boldsymbol{b}\|^2}{E^2}))\exp(\frac{\tau_x^2\|\boldsymbol{b}\|^2}{2E^2} - \frac{D}{E})$$

$$= C_{k=0} \tag{17}$$

Thus we have:

$$P(\mathbf{C})\mathbb{E}_{\mathcal{S}_K}[\sum_{\beta \neq \alpha} \frac{\tilde{\pi}_\beta}{\tilde{\pi}_\alpha}\|\boldsymbol{x}_{k+1}\|^2 \mid \mathbf{C}]$$

$$< 16r \sum_{\beta \neq \alpha} P(\mathbf{C})\mathbb{E}_{\mathcal{S}_K}[\exp(\frac{-\sum_{i=1}^k \|\boldsymbol{\mu}_\beta - \boldsymbol{x}_i\|^2 + \sum_{i=1}^k \|\boldsymbol{\mu}_\alpha - \boldsymbol{x}_i\|^2}{2\sigma_x^2(1 + (k+1)\delta_{\boldsymbol{\mu}})}) \cdot$$

$$\exp(\frac{-\|\boldsymbol{w}_\beta - \boldsymbol{w}^*\|_{\boldsymbol{I}-(\boldsymbol{I}+\Delta\bar{\Sigma}_{\boldsymbol{w}})^{-1}}^2 + \|\boldsymbol{w}_\alpha - \boldsymbol{w}^*\|_{\boldsymbol{I}-(\boldsymbol{I}+\Delta\bar{\Sigma}_{\boldsymbol{w}})^{-1}}^2}{2\sigma_w^2}) \cdot$$

$$\exp(\frac{-\|\boldsymbol{\mu}_\beta - \boldsymbol{x}_i\|^2 + \|\boldsymbol{\mu}_\alpha - \boldsymbol{x}_i\|^2}{2\sigma_x^2(1 + (k+1)\delta_{\boldsymbol{\mu}})})\|\boldsymbol{x}_{k+1}\|^2 \mid \mathbf{C}]$$

(Branch to purple for asymptotic bound or to orange for the bound for the U-shaped pattern.)

$$< 16r \sum_{\beta \neq \alpha} \exp(\frac{-\frac{d_{\boldsymbol{\mu}}^2}{\delta_{\boldsymbol{\mu}}} + \frac{4\tau_x\sqrt{d}}{\delta_{\boldsymbol{\mu}}}k^{\frac{\delta}{2}-\frac{3}{4}} + O(k^{-1})}{2\sigma_x^2})\exp(-\frac{-d_{\boldsymbol{w}}^2 + \frac{4}{\delta_{\boldsymbol{w}}\tau_x^2}k^{-1} + O(k^{\delta-\frac{3}{2}})}{2\sigma_w^2})C_{k=0}$$

$$= 16rMC_{k=0}\exp(\frac{-d_{\boldsymbol{\mu}}^2 + 4\tau_x\sqrt{d}k^{\frac{\delta}{2}-\frac{3}{4}} + O(k^{-1})}{2\sigma_{\boldsymbol{\mu}}^2})\exp(-\frac{-d_{\boldsymbol{w}}^2 + \frac{4}{\delta_{\boldsymbol{w}}\tau_x^2}k^{-1} + O(k^{\delta-\frac{3}{2}})}{2\sigma_w^2})$$

$$= 16rMC_{k=0}\exp(-\frac{d_{\boldsymbol{\mu}}^2 k}{8\sigma_x^2})\exp(-\frac{u_{\boldsymbol{w}}^2\tau_x^2 k}{8\sigma_y^2})$$

We then deal with the second term $P(\mathbf{C})\mathbb{E}_{\mathcal{S}_K}[\tilde{\pi}_\alpha(\langle \tilde{\boldsymbol{w}}_\beta - \boldsymbol{w}_\alpha, \boldsymbol{x}_{k+1}\rangle)^2 \mid \mathbf{C}]$, the part 15:

$$P(\mathbf{C})\mathbb{E}_{\mathcal{S}_K}[\tilde{\pi}_\alpha(\langle \tilde{\boldsymbol{w}}_\alpha - \boldsymbol{w}_\alpha, \boldsymbol{x}_{k+1}\rangle)^2 \mid \mathbf{C}]$$

$$< P(\mathbf{C})\mathbb{E}_{\mathcal{S}_K}[\| - (\boldsymbol{I} - (\boldsymbol{I} + \Delta\bar{\Sigma}_{\boldsymbol{w}})^{-1})(\boldsymbol{w}_\alpha - \boldsymbol{w}^*) + (\boldsymbol{w}_\alpha - \boldsymbol{w}_\alpha)\|^2\|\boldsymbol{x}_{k+1}\|^2 \mid \mathbf{C}]$$

$$< \|\boldsymbol{w}_\alpha - \boldsymbol{w}^*\|^2 P(\mathbf{C}) \mathbb{E}_{\mathcal{S}_K}[\lambda_1^2(\boldsymbol{I} - (\boldsymbol{I} + \Delta\bar{\Sigma}_{\boldsymbol{w}})^{-1})\|\boldsymbol{x}_{k+1}\|^2 \mid \mathbf{C}]$$

(Let $\lambda_1(\boldsymbol{A})$ be the maximal eigenvalue of the matrix $\boldsymbol{A}$)

$$< 4P(\mathbf{C}) \mathbb{E}_{\mathcal{S}_K}[\lambda_1^2(\boldsymbol{I} - (\boldsymbol{I} + \Delta\bar{\Sigma}_{\boldsymbol{w}})^{-1})\|\boldsymbol{x}_{k+1}\|^2 \mid \mathbf{C}]$$

$$< 4P(\mathbf{C}) \mathbb{E}_{\mathcal{S}_K}[(1 - \frac{1}{1 + k\delta_{\boldsymbol{w}}\mathrm{U}})^2 \|\boldsymbol{x}_{k+1}\|^2 \mid \mathbf{C}]$$

$$< 4\mathbb{E}_{\mathcal{S}_K}[\|\boldsymbol{x}_{k+1}\|^2](1 - \frac{1}{1 + k\delta_{\boldsymbol{w}}\mathrm{U}})^2$$

$$= 4(1 + d\tau_x^2)(1 - \frac{1}{1 + k\delta_{\boldsymbol{w}}\mathrm{U}})^2$$

(Branch to purple for asymptotic bound or to orange for the bound for the U-shaped pattern.)

(Let $t = k^{\delta - \frac{1}{2}}$)

$$= 4(1 + d\tau_x^2)(1 - \frac{1}{k\delta_{\boldsymbol{w}}(1 + \tau_x^2)} + O(k^{\delta - \frac{3}{2}}))^2$$

$$= 4(1 + d\tau_x^2)(1 - \frac{2}{k\delta_{\boldsymbol{w}}(1 + \tau_x^2)}) + O(k^{\delta - \frac{3}{2}})$$

(Let $t = k^{-\frac{1}{4}}$, and assuming $\delta_{\boldsymbol{w}} \ll 1$, such that $\exists k \leq \frac{1}{\delta_{\boldsymbol{w}}\tau_x^2}$, s.t. $\mathrm{U} < 2(1 + \tau_x^2)$)

$$< 4(1 + d\tau_x^2)(\frac{k\delta_{\boldsymbol{w}}\mathrm{U}}{1 + k\delta_{\boldsymbol{w}}\mathrm{U}})^2$$

$$< 4(1 + d\tau_x^2)\max\{1, 4k^2\delta_{\boldsymbol{w}}^2(1 + \tau_x^2)^2\}$$

Finally for the third term $P(\neg\mathbf{C})\mathbb{E}_{\mathcal{S}_K}[\sum_{\beta=1}^M \tilde{\pi}_\beta(\langle\tilde{\boldsymbol{w}}_\beta - \boldsymbol{w}_\alpha, \boldsymbol{x}_{k+1}\rangle)^2 \mid \neg\mathbf{C}]$, the part 16:

$$P(\neg\mathbf{C})\mathbb{E}_{\mathcal{S}_K}[\sum_{\beta=1}^M \tilde{\pi}_\beta(\langle\tilde{\boldsymbol{w}}_\beta - \boldsymbol{w}_\alpha, \boldsymbol{x}_{k+1}\rangle)^2 \mid \neg\mathbf{C}]$$

$$= P(\neg\mathbf{C})\mathbb{E}_{\mathcal{S}_K}[\sum_{\beta=1}^M \tilde{\pi}_\beta\|(\boldsymbol{I} + \Delta\bar{\Sigma}_{\boldsymbol{w}})^{-1}(\boldsymbol{w}_\beta - \boldsymbol{w}^*) + \boldsymbol{w}^* - \boldsymbol{w}_\alpha\|^2\|\boldsymbol{x}_{k+1}\|^2 \mid \neg\mathbf{C}]$$

$$< P(\neg\mathbf{C})\mathbb{E}_{\mathcal{S}_K}[\sum_{\beta=1}^M \tilde{\pi}_\beta(2\|(\boldsymbol{I} + \Delta\bar{\Sigma}_{\boldsymbol{w}})^{-1}(\boldsymbol{w}_\beta - \boldsymbol{w}^*)\|^2 + 2\|\boldsymbol{w}^* - \boldsymbol{w}_\alpha\|^2)\|\boldsymbol{x}_{k+1}\|^2 \mid \neg\mathbf{C}]$$

$$< P(\neg\mathbf{C})\mathbb{E}_{\mathcal{S}_K}[\sum_{\beta=1}^M \tilde{\pi}_\beta(2 \cdot 4 + 2 \cdot 4)\|\boldsymbol{x}_{k+1}\|^2 \mid \neg\mathbf{C}]$$

$$= 16P(\neg\mathbf{C})\mathbb{E}_{\mathcal{S}_K}[\sum_{\beta=1}^M \tilde{\pi}_\beta\|\boldsymbol{x}_{k+1}\|^2 \mid \neg\mathbf{C}]$$

$$< 16P(\neg\mathbf{C})\mathbb{E}_{\mathcal{S}_K}[\|\boldsymbol{x}_{k+1}\|^2 \mid \neg\mathbf{C}]$$

$$< 16P(\neg\mathbf{C})\mathbb{E}_{\mathcal{S}_K}[\|\boldsymbol{x}_{k+1}\|^2]$$

$$< 16(1 + d\tau_x^2)P(\neg\mathbf{C})$$

$$< 48(1 + d\tau_x^2)\exp(-\frac{k^{2\delta}}{8})$$

Summarizing three terms, we have:

$$\mathbb{E}_{\mathcal{S}_k \oplus \boldsymbol{x}_{k+1}}[\mathcal{L}_k^{\mathrm{R}}]$$

(Branch to purple for asymptotic bound or to orange for the bound for the U-shaped pattern.)

$$< 16rMC_{k=0}\exp(\frac{-d_{\boldsymbol{\mu}}^2 + 4\tau_x\sqrt{d}k^{\frac{\delta}{2} - \frac{3}{4}} + O(k^{-1})}{2\sigma_\mu^2})\exp(-\frac{-d_{\boldsymbol{w}}^2 + \frac{4}{\delta_{\boldsymbol{w}}\tau_x^2}k^{-1} + O(k^{\delta - \frac{3}{2}})}{2\sigma_w^2}) +$$

$$4(1+d\tau_x^2)(1-\frac{2}{k\delta_{\boldsymbol{w}}(1+\tau_x^2)})+O(k^{\delta-\frac{3}{2}})+48(1+d\tau_x^2)\exp(-\frac{k^{2\delta}}{8})$$

$$<16rMC_{k=0}\exp(-(\frac{d_{\boldsymbol{\mu}}^2}{2\sigma_\mu^2}+\frac{d_{\boldsymbol{w}}^2}{2\sigma_w^2}))\exp(\frac{4\tau_x\sqrt{d}k^{\frac{\delta}{2}-\frac{3}{4}}}{2\sigma_\mu^2}+O(k^{-1}))+$$

$$4(1+d\tau_x^2)(1-\frac{2}{k\delta_{\boldsymbol{w}}(1+\tau_x^2)})+O(k^{\delta-\frac{3}{2}})$$

$$=16rMC_{k=0}\exp(-(\frac{d_{\boldsymbol{\mu}}^2}{2\sigma_\mu^2}+\frac{d_{\boldsymbol{w}}^2}{2\sigma_w^2}))(1+\frac{2\tau_x\sqrt{d}k^{\frac{\delta}{2}-\frac{3}{4}}}{\sigma_\mu^2})+4(1+d\tau_x^2)+O(k^{-1})$$

$$<16rMC_{k=0}\exp(-\frac{d_{\boldsymbol{\mu}}^2 k}{8\sigma_x^2})\exp(-\frac{u_{\boldsymbol{w}}^2\tau_x^2 k}{8\sigma_y^2})+$$

$$4(1+d\tau_x^2)\max\{1,4k^2\delta_{\boldsymbol{w}}^2(1+\tau_x^2)^2\}+48(1+d\tau_x^2)\exp(-\frac{k^{\frac{1}{2}}}{8})$$

The region for the orange formula are:

$$k\le\min\{\frac{1}{\delta_{\boldsymbol{\mu}}},\frac{1}{\delta_{\boldsymbol{w}}\tau_x^2}\}$$

$$4\tau_x\gamma\sqrt{1+k^{-\frac{1}{4}}})<\frac{d_{\boldsymbol{\mu}}^2}{2}$$

$$\text{L}\|\boldsymbol{w}_\beta-\boldsymbol{w}^*\|^2-\text{U}\|\boldsymbol{w}_\alpha-\boldsymbol{w}^*\|^2>\tau_x^2 u_{\boldsymbol{w}}^2/2$$

$$\text{U}<2(1+\tau_x^2)$$

$\square$

## H  PROOF OF LEMMA 5

In this subsection, we introduce the proof of Lemma 5. The proof techniques are very similar to the proof techniques for task retrieval in Sec. G.3.

*Proof.* We are using in-context samples following $\boldsymbol{x}_i\sim\mathcal{N}(\boldsymbol{\mu}^*,\tau_x^2\boldsymbol{I}),y_i=0$, *i.e.*, $\boldsymbol{w}^*=\boldsymbol{0}$, and we aim to have the prediction on $\mathcal{S}_k\oplus\boldsymbol{x}_{k+1}$ as $\langle\boldsymbol{x}_{k+1},\boldsymbol{w}_\alpha\rangle$, *i.e.*, to retrieve the prediction of the clean task $\alpha$. In order to have an upper bound on the loss, we consider $\boldsymbol{x}_i\sim\mathcal{N}(\boldsymbol{\mu}^*,\tau_x^2\boldsymbol{I})$ in two regions: (1) **C**: $\text{L}<\lambda_d(\frac{\sum_{i=1}^k\boldsymbol{x}_i\boldsymbol{x}_i^\top}{k})\le\lambda_1(\frac{\sum_{i=1}^k\boldsymbol{x}_i\boldsymbol{x}_i^\top}{k})<\text{U}$ (see Lemma 6 for L and U) and (2) $\neg\textbf{C}$: either the previous inequality does not hold. The probability of $\neg\textbf{C}$ is bounded by:

$$P(\neg\textbf{C})<3\exp(-\frac{kt^2}{8}).$$

Let $\mathcal{L}_k^{\text{R}}$ indicate the squared loss $(\mathcal{F}^*(\mathcal{S}_k\oplus\boldsymbol{x}_{k+1})-\langle\boldsymbol{x}_{k+1},\boldsymbol{w}_\alpha\rangle)^2$ on $\mathcal{S}_k\oplus\boldsymbol{x}_{k+1}$. With the help of Lemma 1 and Corollary 2, we can derive the expected squared loss on the prediction $\mathcal{F}^*(\mathcal{S}_k\oplus\boldsymbol{x}_{k+1})$, and then based on **C** and the target task $\alpha$, we split the expected squared loss into three parts similar to Sec. G.3:

$$\mathbb{E}_{\mathcal{S}_k\oplus\boldsymbol{x}_{k+1}}[\mathcal{L}_k^{\text{R}}]$$

$$<P(\textbf{C})\mathbb{E}_{\mathcal{S}_K}[\tilde{\pi}_\beta(\langle\tilde{\boldsymbol{w}}_\beta-\boldsymbol{w}_\alpha,\boldsymbol{x}_{k+1}\rangle)^2\mid\textbf{C}]+ \qquad (18)$$

$$P(\textbf{C})\mathbb{E}_{\mathcal{S}_K}[\tilde{\pi}_\alpha(\langle\tilde{\boldsymbol{w}}_\alpha-\boldsymbol{w}_\alpha,\boldsymbol{x}_{k+1}\rangle)^2\mid\textbf{C}]+ \qquad (19)$$

$$P(\neg\textbf{C})\mathbb{E}_{\mathcal{S}_K}[\sum_{\beta=1}^2\tilde{\pi}_\beta(\langle\tilde{\boldsymbol{w}}_\beta-\boldsymbol{w}_\alpha,\boldsymbol{x}_{k+1}\rangle)^2\mid\neg\textbf{C}] \qquad (20)$$

We firstly analyze the first term $P(\textbf{C})\mathbb{E}_{\mathcal{S}_K}[\tilde{\pi}_\beta(\langle\tilde{\boldsymbol{w}}_\beta-\boldsymbol{w}_\alpha,\boldsymbol{x}_{k+1}\rangle)^2\mid\textbf{C}]$ in Part. 18. Similar to Sec. G.3, we have:

$$P(\textbf{C})\mathbb{E}_{\mathcal{S}_K}[\tilde{\pi}_\beta(\langle\tilde{\boldsymbol{w}}_\beta-\boldsymbol{w}_\alpha,\boldsymbol{x}_{k+1}\rangle)^2\mid\textbf{C}]$$

$$<16P(\textbf{C})\mathbb{E}_{\mathcal{S}_K}[\frac{\tilde{\pi}_\beta}{\tilde{\pi}_\alpha}\|\boldsymbol{x}_{k+1}\|^2\mid\neg\textbf{C}]$$

Apply Eqs. 4, 5, and 6 and Assumption 3(b) to $\frac{\tilde{\pi}_\beta}{\tilde{\pi}_\alpha}$, we have a different results from Sec. G.3 since we have $\boldsymbol{w}_\beta = -\boldsymbol{w}_\alpha$ and $\boldsymbol{w}^* = \boldsymbol{0}$:

$$\mathbb{E}_{\mathcal{S}_K}[\sum_{\beta \neq \alpha} \frac{\tilde{\pi}_\beta}{\tilde{\pi}_\alpha} \|\boldsymbol{x}_{k+1}\|^2 \mid \mathbf{C}]$$

$$< 16 \mathbb{E}_{\mathcal{S}_K}[\sum_{\beta \neq \alpha} r \exp(\frac{-\sum_{i=1}^k \|\boldsymbol{\mu}_\beta - \boldsymbol{x}_i\|^2 + \sum_{i=1}^k \|\boldsymbol{\mu}_\alpha - \boldsymbol{x}_i\|^2}{2\sigma_x^2(1 + (k+1)\delta_{\boldsymbol{\mu}})}) \cdot$$

$$\exp(\frac{-\|\boldsymbol{w}_\beta - \boldsymbol{w}^*\|^2_{\boldsymbol{I}-(\boldsymbol{I}+\Delta\bar{\Sigma}_{\boldsymbol{w}})^{-1}} + \|\boldsymbol{w}_\alpha - \boldsymbol{w}^*\|^2_{\boldsymbol{I}-(\boldsymbol{I}+\Delta\bar{\Sigma}_{\boldsymbol{w}})^{-1}}}{2\sigma_w^2}) \cdot$$

$$\exp(\frac{-\|\boldsymbol{\mu}_\beta - \boldsymbol{x}_{k+1}\|^2 + \|\boldsymbol{\mu}_\alpha - \boldsymbol{x}_{k+1}\|^2}{2\sigma_x^2(1 + (k+1)\delta_{\boldsymbol{\mu}})}) \|\boldsymbol{x}_{k+1}\|^2 \mid \mathbf{C}]$$

(Notice $\boldsymbol{w}^* = \boldsymbol{0}, \boldsymbol{w}_\beta = -\boldsymbol{w}_\alpha$)

$$= 16r \sum_{\beta \neq \alpha} \mathbb{E}_{\mathcal{S}_K}[\exp(\frac{-\sum_{i=1}^k \|\boldsymbol{\mu}_\beta - \boldsymbol{x}_i\|^2 + \sum_{i=1}^k \|\boldsymbol{\mu}_\alpha - \boldsymbol{x}_i\|^2}{2\sigma_x^2(1 + (k+1)\delta_{\boldsymbol{\mu}})}) \cdot$$

$$\exp(\frac{-\|\boldsymbol{\mu}_\beta - \boldsymbol{x}_i\|^2 + \|\boldsymbol{\mu}_\alpha - \boldsymbol{x}_i\|^2}{2\sigma_x^2(1 + (k+1)\delta_{\boldsymbol{\mu}})}) \|\boldsymbol{x}_{k+1}\|^2 \mid \mathbf{C}]$$

Recall in case $\mathbf{C}$ we have:

$$\|\frac{\sum_{i=1}^k \boldsymbol{\epsilon}_i}{k}\| < \tau_x \gamma \sqrt{1+t}$$

Therefore, when conditioned on case $\mathbf{C}$, similar to Sec. G.3, we have:

$$\frac{\sum_{i=1}^k (-\|\boldsymbol{\mu}_\beta - \boldsymbol{x}_i\|^2 + \|\boldsymbol{\mu}_\alpha - \boldsymbol{x}_i\|^2)}{1 + (k+1)\delta_{\boldsymbol{\mu}}}$$

(Branch to purple for asymptotic bound or to orange for the bound for the U-shaped pattern.)

(Let $t = k^{\delta - \frac{1}{2}}$ and $\delta$ is small.)

$$< -\frac{d_{\boldsymbol{\mu}}^2}{\delta_{\boldsymbol{\mu}}} + \frac{4\tau_x \sqrt{d}}{\delta_{\boldsymbol{\mu}}} k^{\frac{\delta}{2} - \frac{3}{4}} + O(k^{-1})$$

(let $t = k^{-\frac{1}{4}}$, When $\delta_{\boldsymbol{\mu}} \ll 1$, such that $\exists k \leq \frac{1}{\delta_{\boldsymbol{\mu}}}$, s.t. $\frac{d_{\boldsymbol{\mu}}^2}{2} > 4\tau_x \gamma \sqrt{1 + k^{-\frac{1}{4}}}$)

$$< -\frac{d_{\boldsymbol{\mu}}^2}{4}$$

And similar to Sec. G.3 we have:

$$P(\mathbf{C}) \mathbb{E}_{\mathcal{S}_K}[\sum_{\beta \neq \alpha} \frac{\tilde{\pi}_\beta}{\tilde{\pi}_\alpha} \|\boldsymbol{x}_{k+1}\|^2 \mid \mathbf{C}]$$

(Branch to purple for asymptotic bound or to orange for the bound for the U-shaped pattern.)

$$< 16r \sum_{\beta \neq \alpha} \exp(\frac{-\frac{d_{\boldsymbol{\mu}}^2}{\delta_{\boldsymbol{\mu}}} + \frac{4\tau_x \sqrt{d}}{\delta_{\boldsymbol{\mu}}} k^{\frac{\delta}{2} - \frac{3}{4}} + O(k^{-1})}{2\sigma_x^2}) C_{k=0}$$

$$= 16r C_{k=0} \exp(\frac{-d_{\boldsymbol{\mu}}^2 + 4\tau_x \sqrt{d} k^{\frac{\delta}{2} - \frac{3}{4}} + O(k^{-1})}{2\sigma_{\boldsymbol{\mu}}^2})$$

$$< 16r C_{k=0} \exp(-\frac{d_{\boldsymbol{\mu}}^2 k}{8\sigma_x^2})$$

The analysis for the second term $P(\mathbf{C}) \mathbb{E}_{\mathcal{S}_K}[\tilde{\pi}_\alpha(\langle \tilde{\boldsymbol{w}}_\beta - \boldsymbol{w}_\alpha, \boldsymbol{x}_{k+1}\rangle)^2 \mid \mathbf{C}]$, the part 19 and the third term $P(\neg\mathbf{C}) \mathbb{E}_{\mathcal{S}_K}[\sum_{\beta=1}^2 \tilde{\pi}_\beta(\langle \tilde{\boldsymbol{w}}_\beta - \boldsymbol{w}_\alpha, \boldsymbol{x}_{k+1}\rangle)^2 \mid \neg\mathbf{C}]$, the part 20, are the same as Sec. G.3.

Summarizing three terms, we have:

$$\mathbb{E}_{\mathcal{S}_k \oplus \boldsymbol{x}_{k+1}}[\mathcal{L}_k^{\mathrm{R}}]$$

(Branch to purple for asymptotic bound or to orange for the bound for the U-shaped pattern.)

$$< 16rC_{k=0} \exp\left(\frac{-d_{\boldsymbol{\mu}}^2 + 4\tau_x \sqrt{d} k^{\frac{\delta}{2} - \frac{3}{4}} + O(k^{-1})}{2\sigma_{\mu}^2}\right) +$$

$$4(1 + d\tau_x^2)(1 - \frac{2}{k\delta_{\boldsymbol{w}}(1 + \tau_x^2)}) + O(k^{\delta - \frac{3}{2}}) + 48(1 + d\tau_x^2)\exp(-\frac{k^{2\delta}}{8})$$

$$< 16rC_{k=0} \exp(-\frac{d_{\boldsymbol{\mu}}^2}{2\sigma_{\mu}^2})\exp\left(\frac{4\tau_x \sqrt{d} k^{\frac{\delta}{2} - \frac{3}{4}}}{2\sigma_{\mu}^2} + O(k^{-1})\right) +$$

$$4(1 + d\tau_x^2)(1 - \frac{2}{k\delta_{\boldsymbol{w}}(1 + \tau_x^2)}) + O(k^{\delta - \frac{3}{2}})$$

$$= 16rC_{k=0} \exp(-\frac{d_{\boldsymbol{\mu}}^2}{2\sigma_{\mu}^2})(1 + \frac{2\tau_x \sqrt{d} k^{\frac{\delta}{2} - \frac{3}{4}}}{\sigma_{\mu}^2}) + 4(1 + d\tau_x^2) + O(k^{-1})$$

$$< 16rC_{k=0} \exp(-\frac{d_{\boldsymbol{\mu}}^2 k}{8\sigma_x^2}) + 4(1 + d\tau_x^2)\max\{1, 4k^2\delta_{\boldsymbol{w}}^2(1 + \tau_x^2)^2\} + 48(1 + d\tau_x^2)\exp(-\frac{k^{\frac{1}{2}}}{8})$$

$\square$

# I    DEMO PROBLEM AS A WARMUP

We study how demonstrations of ICL affect the prediction of a pretrained LM, and how the pretraining distribution affects this phenomenon. In other words, the LM denoted as $f$ is initially pretrained on a dataset distribution to produce the minimum risk minimizer $f^*$, and then the pretrained $f^*$ is used to predict the $y$ value of the input $x$. However, instead of directly inferencing via $f(x)$, we consider inferencing with additional $k$ demonstrations $\{x_i\}_{i=1}^k$ via the format $\hat{f}([x_1, \ldots, x_k, x])$. We aim to theoretically examine the effect of demonstrations $\{x_i\}_{i=1}^k$ on the prediction $\hat{f}([x_1, \ldots, x_k, x])$. Before going to the formal problem setting, we introduce this demo section to illustrate the basic phenomenon for better delivering our work.

The following demo subsections are organized as follows. We first introduce the problem setting in Sec. I.1. We then connect ICL with Bayesian inference in Sec. I.2. Further, we introduce the assumptions for the pretrained dataset in Sec. I.3. And finally, we derive a closed-form posterior and introduce two phenomena "Topic Shifting" and "Topic Re-weighting" in Sec. I.4.

## I.1    DEMO: PROBLEM SETTING

ICL (In-context learning) involves two important components: the pretraining dataset, and the LM (language model) supporting varied input lengths. We assume the LM $f : \cup_{k \in \{0, \ldots, K-1\}} \mathcal{R}^{k \times 1} \to \mathcal{R}^{1 \times 1}$ can fit the pretraining distribution exactly with enough data and expressivity. Each sample is generated from firstly picking a task $\mu$ from underlying task distribution $\mathcal{D}_\mu$, and then we generate context/tokens from a distribution $\mathcal{D}_x(\mu)$ based on the task $\boldsymbol{\mu}$. The sample generation process is described below:

**Assumption 6** (Demo Generation Process). *Given a task **prior** distribution $\mathcal{D}_\mu$, and a conditioned $x$ sampler $\mathcal{D}_x(\mu)$ conditioned on task $\mu$, the process of generating a sequence/sample $S_K = [x_1, x_2, \ldots, x_K]$ with length $K$ follows:*

(a) ***Sample a Task $\mu$ from the Prior:*** *$\mu \sim \mathcal{D}_\mu$, and the probability of $\mu$ is indicated by $P(\mu)$;*

(b) ***Sample $x$ from the Conditioned $x$ Sampler:*** *For $i \in \{1, 2, \ldots, K\}$, $x_i \sim \mathcal{D}_x(\mu)$, and the probability of $x_i = x$ is indicated by $P(x|\mu)$;*

(c) ***Define a Sequence:*** *For capital $K$, $S_K = [x_1, \ldots, x_K]$; and for lowercase $k$, the sequence of the first $k$ demonstrations of $S_K$ is indicated by $S_k = [x_1, \ldots, x_k]$, e.g., $S_2 = [x_1, x_2]$.*

**Remark 1.** *The generation process is related to real-world scenarios via two points: (1) For sampling step (i), the LM is trained on varied tasks; (2) For sampling step (ii), when one person/agent produces texts for one task, the generated text could be varied. An instance of the two points is that, given several tasks such as describing the fruit market, describing a football game, and describing the world environment, etc, one person picks the task of describing the fruit category, and he potentially has multiple ways to describe it.*

Now we consider training the LM $f(\cdot)$ using the sample $S_K$ generated via above generation process 6 via squared loss:

$$\mathcal{L}(f) = \mathop{\mathbb{E}}_{\mu \sim \mathcal{D}_\mu} [ \mathop{\mathbb{E}}_{\substack{x_i \sim \mathcal{D}(\mu), \\ i \in \{1,\dots,K\}}} [\sum_{k=0}^{K-1} (f(S_k) - x_{k+1})^2 | \mu]]. \tag{21}$$

How is this prediction related to Bayesian inference? We show the relationship between the prediction and Bayesian inference in the next Sec. I.2.

## I.2 Demo: Connecting ICL with Bayesian Inference

In this subsection, we try to connect ICL to Bayesian inference. Starting from Eq. 21, we can disentangle the loss function for inputs with different token lengths as follows:

$$\mathcal{L}(f) = \mathop{\mathbb{E}}_{\mu \sim \mathcal{D}_\mu} [ \mathop{\mathbb{E}}_{\substack{x_i \sim \mathcal{D}(\mu), \\ i \in \{1,\dots,K\}}} [\sum_{k=0}^{K-1} (f(S_k) - x_{k+1})^2 | \mu]]$$

$$= \mathop{\mathbb{E}}_{S_K} [\sum_{k=0}^{K-1} (f(S_k) - x_{k+1})^2]$$

$$= \sum_{k=0}^{K-1} \mathop{\mathbb{E}}_{S_K} [(f(S_k) - x_{k+1})^2].$$

Thus when the model $f$ has enough expressivity, the optimization problem $\mathrm{argmin}_f \mathcal{L}(f)$ of minimization of the loss function $\mathcal{L}(f)$ could be regarded as $K$ different minimization tasks:

$$\mathop{\mathrm{argmin}}_f \mathop{\mathbb{E}}_{S_K} [(f(S_k) - x_{k+1})^2], k \in \{0, \dots, K-1\}.$$

Thus the solution for each $k$ is a minimum mean square error (MMSE) estimator. Assuming $f$ has enough expressivity, exists $f^*$ that satisfy $f^* \in \mathrm{argmin}_f \mathop{\mathbb{E}}_{S_K} [(f(S_k) - x_{k+1})^2], k \in \{0, \dots, K-1\}$ simultaneously, and the prediction of $f^*(S_k)$ satisfies:

$$f^*(S_k) = \mathop{\mathbb{E}}_{S_K} [x_{k+1} | S_k] = \mathop{\mathbb{E}}_{\mu \sim \mathcal{D}_\mu} [ \mathop{\mathbb{E}}_{\substack{x_i \sim \mathcal{D}(\mu), \\ i \in \{1,\dots,K\}}} [x_{k+1} | \mu, S_k] | S_k] = \mathop{\mathbb{E}}_{\mu \sim \mathcal{D}_\mu} [ \mathop{\mathbb{E}}_{x_{k+1} \sim \mathcal{D}(\mu)} [x_{k+1} | \mu] | S_k] \tag{22}$$

Where the prediction is the expectation of $\mathop{\mathbb{E}}_{x_{k+1} \sim \mathcal{D}(\mu)} [x_{k+1} | \mu]$ under the posterior of $\mathcal{D}_\mu$ after observing $S_k$.

## I.3 Demo: Assumptions

Now in the previous Sec. I.2, we connect ICL with Bayesian inference. From Eq. 22 we observe that the prediction $f^*(S_k)$ depends on the posterior. We are interested in how the demonstrations affect the prediction of ICL, *i.e.*, how observation affects the posterior/prediction of Bayesian inference. In order to make further derivation on the posterior, we make further assumptions on the pretraining dataset as follows:

**Assumption 7** (Demo Prior Distribution). *Assumptions on the task **prior** distribution $\mathcal{D}_\mu$, and the conditioned token sampler $\mathcal{D}_x(\mu)$ conditioned on task $\mu$:*
*(a) The distribution $\mathcal{D}_\mu$ is a mixture of $M$ Gaussian distributions: $\mathcal{D}_\mu = \sum_{\beta=1}^M \pi_\beta T_\beta = \sum_{\beta=1}^M \pi_\beta \mathcal{N}(\mu_\beta, \sigma^2)$, where $\mu_\beta$ is the center of the mixture component $T_\beta$, and all $M$ mixture*

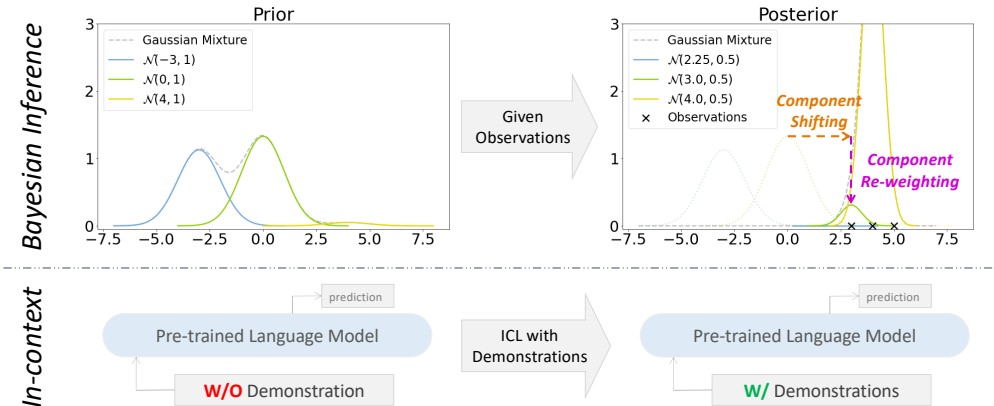

Figure 10: The left part of the figure indicates the LM is pretrained on samples generated from the prior distribution according to Assumption 7, and without demonstration, the pretrained LM predicts based on the prior. The right part of the figure indicates that with demonstrations, the pretrained language model predicts based on posterior, regarding the demonstrations as observed samples.

components share the same variance $\sigma^2$;
*(b) The distribution $\mathcal{D}_x(\mu)$ is a Gaussian distribution: $\mathcal{N}(\mu, \tau^2)$, where $\tau^2$ is the variance of the Gaussian and could be regarded as the noise scale when sampling $\mu$;*

**Remark 2.** *In our setting, we train the LM on $M$ tasks, mirroring real-world LM pretrained on varied topics including environment, market, movie, sports, etc. These tasks have text sequences from diverse sources like individuals, agents, and websites. Given that each source interprets tasks uniquely, they effectively provide "noisy" versions of the same task. We model this using a Gaussian mixture for the task prior. Each component's mean represents a specific task, while its variance captures the interpretive noises. Consequently, our invariant generator produces tokens based on these "noisy" task interpretations.*

### I.4  DEMO: DERIVATION OF POSTERIOR

With further Assumption 7 on the prior, *i.e.*, the pretraining distribution of ICL in Sec. I.3, we can make further derivation on the posterior:

$$P(\mu|S_k) \propto \sum_{m=1}^{M} \tilde{\pi}_\beta \mathcal{N}(\mu|\tilde{\mu}_\beta, \tilde{\sigma}^2) \tag{23}$$

$$(\tilde{\pi}_\beta = \pi_\beta \exp(\frac{(\mu_\beta - \frac{\sum_{i=1}^{k} x_i}{k})^2}{2(\tau^2 + k\sigma^2)}), \tilde{\mu}_\beta = \frac{\tau^2 \mu_\beta + \sigma^2 \sum_{i=1}^{k} x_i}{\tau^2 + k\sigma^2})^2, \tilde{\sigma}^2 = \frac{\tau^2 \sigma^2}{\tau^2 + k\sigma^2})$$

From Eq. 23, we observe two factors when comparing the posterior with the prior in Assumption 7: (i) Topic-Shifting: after observing $S_k = [x_1, x_2, \ldots, x_k]$, each mixture component's center is shifted to $\frac{\tau^2 \mu_\beta + \sigma^2 \sum_{i=1}^{k} x_i}{\tau^2 + k\sigma^2}$; (ii) Topic Re-weighting: each mixture component's mixture weight $\pi_\beta$ is re-weighted by multiplying $\exp(\frac{(\mu_\beta - \frac{\sum_{i=1}^{k} x_i}{k})^2}{2(\tau^2 + k\sigma^2)})$ (which needs to be further normalized so that re-weighted mixture weights sum to 1). Fig. 10 illustrates the phenomena of Topic Shifting and Topic Re-weighting when observing samples transferring prior to posterior.

## J  PROOF OF POSTERIOR DERIVATION IN DEMO

In this section, we give a detailed derivation of the posterior in Eq. 23 in Sec. I.4:

$$P(\mu \mid S_k) \propto P(\mu, S_k)$$
$$= P(S_k \mid \mu)P(\mu)$$

$$= (\Pi_{i=1}^k P(x_i \mid \mu))P(\mu)$$

$$= (\Pi_{i=1}^k \mathcal{N}(x_i \mid \mu, \tau^2)) \sum_{m=1}^M \pi_\beta \mathcal{N}(\mu \mid \mu_\beta, \sigma^2)$$

$$\propto (\Pi_{i=1}^k \exp(-\frac{(x_i - \mu)^2}{2\tau^2})) \sum_{m=1}^M \pi_\beta \exp(-\frac{(\mu - \mu_\beta)^2}{2\sigma^2})$$

$$= \exp(-\frac{\sum_{i=1}^k (x_i - \mu)^2}{2\tau^2}) \sum_{m=1}^M \pi_\beta \exp(-\frac{(\mu - \mu_\beta)^2}{2\sigma^2})$$

$$= \sum_{\beta=1}^M \pi_\beta \exp(-\frac{\tau^2(\mu - \mu_\beta)^2 + \sigma^2 \sum_{i=1}^k (x_i - \mu)^2}{2\tau^2\sigma^2})$$

$$= \sum_{m=1}^M \pi_\beta \exp(-\frac{\mu^2(\tau^2 + k\sigma^2) - 2\mu(\tau^2\mu_\beta + \sigma^2 \sum x_i) + (\tau^2\mu_\beta^2 + \sigma^2 \sum x_i^2)}{2\tau^2\sigma^2})$$

$$= \sum_{m=1}^M \pi_\beta \exp(-\frac{(\mu - \frac{\tau^2\mu_\beta + \sigma^2 \sum x_i}{\tau^2 + k\sigma^2})^2 + \frac{\tau^2\mu_\beta^2 + \sigma^2 \sum x_i^2}{\tau^2 + k\sigma^2} - (\frac{\tau^2\mu_\beta + \sigma^2 \sum x_i}{\tau^2 + k\sigma^2})^2}{2\frac{\tau^2\sigma^2}{\tau^2 + k\sigma^2}})$$

$$\propto \sum_{m=1}^M \pi_\beta \exp(\frac{(\mu_\beta - \frac{\sum_{i=1}^k x_i}{k})^2}{2(\tau^2 + k\sigma^2)}) \exp(-\frac{(\mu - \frac{\tau^2\mu_\beta + \sigma^2 \sum_{i=1}^k x_i}{\tau^2 + k\sigma^2})^2}{2\frac{\tau^2\sigma^2}{\tau^2 + k\sigma^2}})$$

$$\propto \sum_{m=1}^M \tilde{\pi}_\beta \mathcal{N}(\mu \mid \tilde{\mu}_\beta, \tilde{\sigma}^2)$$

$$(\tilde{\pi}_\beta = \pi_\beta \exp(\frac{(\mu_\beta - \frac{\sum_{i=1}^k x_i}{k})^2}{2(\tau^2 + k\sigma^2)}), \tilde{\mu}_\beta = \frac{\tau^2\mu_\beta + \sigma^2 \sum_{i=1}^k x_i}{\tau^2 + k\sigma^2})^2, \tilde{\sigma}^2 = \frac{\tau^2\sigma^2}{\tau^2 + k\sigma^2})$$

