# ADDITIONAL EXPERIMENTS FOR REBUTTAL

## 1  U-SHAPED LOSS IN GPT-4

The experiment is designed as follows:

| Setting | Desciption |
|---|---|
| LLM | GPT-4 |
| System Massage | You are a mathematician. Consider the following math problem and follow the exact instruction. |
| Prompt | You are given examples. Each example has two integers as input and one integer as output. Please provide an answer for the last problems in the math exercise: 
 $a_1(?)b_1=c_1$ 
 ... 
 $a_k(?)b_k=c_2$ 
 $a_{k+1}(?)b_{k+1}=$ 
 Provide your answer directly. |
| $a_i$, $b_i$, and $c_i$ | $a_i$ and $b_i$ are uniformly sampled from $[10, 99]$, and $c_i = a_i + b_i + 1$. |
| Goal of Task Retrieval | Aiming to retrieve the "addition" task, a(?)b=(a+b). However, the in-context samples are provided with a slightly different task "add-1 addition", a(?)b=(a+b+1). |
| Goal of Task Learning | Aiming to learn the "add-1 addition" task, a(?)b=(a+b+1), with in-context samples following the same "add-1 addition" task, a(?)b=(a+b+1). |

Under the same prompt setting in the table, as the number of in-context samples increases, we observe the error rate for task learning constantly decreases while the error rate for task retrieval follows a U-shaped curve, shown in the following table.

| Number of In-context Samples ($k$) | 0 | 1 | 2 | 4 | 8 | 16 |
|---|---|---|---|---|---|---|
| Error Rate of Task Retrieval | 75.0% | 36.2% | **33.9%** | 49.3% | 79.3% | 85.1% |
| Error Rate of Task Learning | 100.0% | 98.3% | 95.9% | 60.5% | 24.4% | **16.8%** |

We further randomly sample four pairs of prompt and prediction for $k = 0, 2, 8$ in the following tables for reference:

Table 1: 0 in-context samples, $k = 0$. Prediction is colored red if it is correct for task retrieval $(a(?)b = (a + b))$, and colored blue if it is correct for task learning $(a(?)b = (a + b + 1))$.

| prompt | ...
51(?)36=
... | ...
27(?)15=
... | ...
76(?)82=
... | ...
55(?)15=
... |
|---|---|---|---|---|
| results | Without knowing the operation or rule that connects the two input integers to the output integer in the examples, it's impossible to provide a correct answer. Please provide the examples or the rule. | Sorry, but your questionis not clear. Could you please provide more information about the operation between the two numbers? | Your question seems to be missing some information. Could you please provide the examples you mentioned? They are necessary to understand the relationship between the two input integers and the output integer. | 70 |

Table 2: 2 in-context samples, $k = 2$. Prediction is colored red if it is correct for task retrieval $(a(?)b = (a + b))$, and colored blue if it is correct for task learning $(a(?)b = (a + b + 1))$.

| prompt | ...
73(?)80=154
59(?)22=82
54(?)97=
... | ...
48(?)73=122
78(?)80=159
21(?)33=
... | ...
21(?)28=50
69(?)29=99
47(?)10=
... | ...
94(?)43=138
98(?)70=169
96(?)41=
... |
|---|---|---|---|---|
| results | 151 | 54 | 57 | 187 |

Table 3: 8 in-context samples, $k = 8$. Prediction is colored red if it is correct for task retrieval $(a(?)b = (a + b))$, and colored blue if it is correct for task learning $(a(?)b = (a + b + 1))$.

| prompt | ...
37(?)70=108
41(?)18=60
19(?)12=32
82(?)67=150
42(?)13=56
26(?)41=68
80(?)39=120
58(?)23=82
40(?)90=
... | ...
60(?)76=137
69(?)26=96
72(?)85=158
39(?)10=50
50(?)47=98
19(?)63=83
45(?)95=141
69(?)41=111
81(?)36=
... | ...
66(?)40=107
46(?)81=128
63(?)31=95
41(?)24=66
70(?)43=114
89(?)84=174
76(?)82=159
46(?)28=75
49(?)46=
... | ...
68(?)88=157
34(?)18=53
70(?)70=141
13(?)35=49
52(?)50=103
72(?)32=105
98(?)82=181
55(?)51=107
50(?)31=
... |
|---|---|---|---|---|
| results | 130 | 118 | 96 | 82 |

## 2  FLIPPED U-SHAPED LOSS UNDER PROPOSED ICL SETTING

Xie et al. (2022) observed that as the number of in-context samples increases, the performance of ICL first decreases and then increases as shown in the figure and caption below clipped and copied from the original paper.

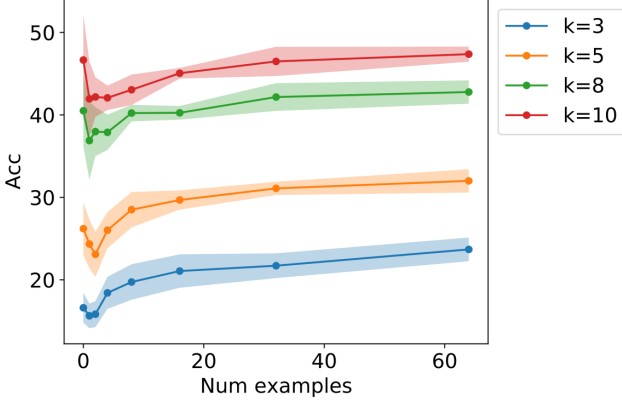

Figure 1: Zero-shot performance can be higher than one/few-shot performance in some settings in GINC, mirroring the behavior of GPT3 on some datasets such as LAMBADA (Brown et al., 2020). The few-shot setting introduces the distracting prompt structure, which can initially lower accuracy.

We show under our framework, we are able to mimic the performance tendency in the following figure which has a flipped U-shaped curve:

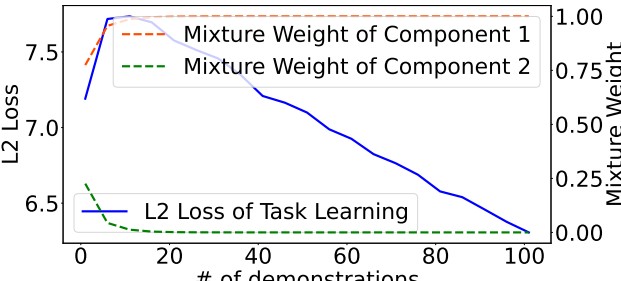

Figure 2: ICL aims to learn the prediction following in-context samples. However, due to the $x$ distribution of in-context samples aligning with center 1, as the increasing number of in-context samples, center 1 is retrieved first which causes higher loss. Further, with a large number of in-context samples, component shifting takes effect, making the prediction align with in-context samples.

We further summarize the setting of the prior distribution and in-context task in the following table which leads to the phenomenon in Figure 2:

|  | Mixture Weight | $\mu$ | $w$ | $\sigma_\mu$ | $\sigma_w$ | $\sigma_x$ | $\sigma_y$ |
|---|---|---|---|---|---|---|---|
| Component 1 | 1/2 | $\mu_1 = [+1]$ | $w_1 = [-1]$ | 0.05 | 0.05 | 1 | 2 |
| Component 2 | 1/2 | $\mu_2 = [-1]$ | $w_2 = [+1]$ | 0.05 | 0.05 | 1 | 2 |
| In-context Task | / | $\mu^* = [+1]$ | $w^* = [+1]$ | / | / | 1 | 0 |

Table 4: The pretraining task prior has two components, component 1 and component 2. We are aiming to use the in-context samples to learn the prediction of the in-context task, or equivalently in this case, to retrieve the prediction of the center of component 2, (notice $w_2$ of component 2 is equal to $w^*$ of the in-context task). In-context task has $\|\mu^* - \mu_1\| < \|\mu^* - \mu_2\|$ and $\|w^* - w_1\| > \|w^* - w_2\|$, *i.e.*, in-context samples has similar $x$ distribution to the center of component 1 while has similar $x \rightarrow y$ mapping to the center of component 2.