# OpenReview forum: "A Theoretical Analysis of In-context Task Retrieval and Learning"
_ICLR.cc/2024/Conference — Submitted to ICLR 2024_

### Official Review · Reviewer_v8Fu · 2023-10-21

**Soundness:** 2 fair
**Presentation:** 2 fair
**Contribution:** 2 fair
**Rating:** 3
**Confidence:** 3

**Summary:**

They assume that the pretraining data and the downstream task data is generated from a Gaussian/Linear generation process.  They then theoretically analyze the Bayesian-optimal predictions in the in-context learning setting. They divide in-context learning for downstream tasks into two categories depending on the parameters of their generation processes: (1) task retrieval, and (2) task learning. They derive error bounds for these two categories. Their theoretical results show that having more in-context examples may hurt the performance of the model.

**Strengths:**

1. A clear/rigorous/mathematical definition of two types of in-context learning (task retrieval and task learning) may be helpful for future works.
2. They derive the error bounds for the given setting that decreases quadratically.

**Weaknesses:**

Because Xie et al. (2022) has proposed to explain in-context learning with a latent variable model, I would expect new studies, if also adopt a latent variable model, to propose some refinement on the data generation process which should be more realistic. However, in my opinion, the date generation process in this work is not more realistic for the following reasons:

1. Firstly, it’s a gaussian model, so it’s very different from the discrete case of NLP. Because it’s not discrete, it is even less realistic than the HMM model used by Xie et al.
2. Secondly, this work assumes that the generation process of the pretraining data is the same as the downstream task. Again, this is worse than Xie et al., because Xie et al. discuss the distribution mismatch problem between pretraining and downstream tasks at least to some extent.

Additionally, it’s not clear to me what the main takeaway of this paper is. Indeed, the authors derive the error bounds for the two kinds of downstream tasks, however

1. It’s not clear to me how the definition of the two kinds of tasks is relevant in the real-world scenario.
2. Empirically, we do not observe that having more examples hurt the performance.
3. And again, it’s not clear how the generation process is related to the real-world data.

**Questions:**

I suggest the authors elaborate more on the implications of the bounds they prove.

---

> ### Author Response · Authors · 2023-11-21
>
> We would like to express our gratitude to the reviewer for dedicating your valuable time to provide constructive feedback on our work. We hope our responses have addressed your concerns and answered your questions. If our responses have resolved your concerns, we kindly request you to consider increasing your score.
> ___
> (1) “I would expect new studies” & “It’s not clear to me what the main takeaway of this paper”
>
> Our paper makes two major novel contributions.
>
> Firstly, we are the pioneering work that formally defines and analyzes task learning and retrieval. This can assist future researchers in reconciling seemingly conflicting observations made with in-context learning in the past, as they can be interpreted as non-conflicting results within the two different regimes.
>
> Secondly, we are the first to precisely predict two non-monotonic performance patterns of ICL in real-world LLMs as the number of samples increases: the U-shaped curve we predicted in the original submission, and the flipped one we presented in the rebuttal. The U-shaped curve was confirmed by our additional experiments. The flipped curve was initially presented in the original GPT3 paper (and later was re-confirmed by Xie et al.’s work too!) and explained via experiment under our framework. Experiments are described in “supplementary material”.
>
> These two contributions lead us to strongly believe that our paper offers substantial contributions to the research community, particularly theoretical analysis of ICL, and to some extent, practical LLM research.
> ___
> (2) (paraphrased) “data generation process should be more realistic” & “It’s not clear how the generation process is related to the real-world data”
>
> We deliberately selected the linear regression mixture model, which, as the reviewer correctly pointed out, is slightly less realistic (non-discrete) than the HMM model. Our choice was motivated by the model's analytical tractability, which provides a precise theoretical understanding of how task retrieval and task learning operate.
>
> Numerous theoretical studies adopt the same linear regression mixture model inluding recent works  [1-5] (see these papers in “To AC and All Reviewers” due to the character's length). Although the linear regression mixture model deviates from the actual pretraining data model, the insights it offers may still predict real-world phenomena.
> ___
> (3) “The generation process of the pretraining data is the same as the downstream task”
>
> We do *not* make such an assumption. Although we assume that the pretraining task parameters are drawn from a probability distribution whose support spans the entire task parameter space, this does not imply that every task is included in a finite pretraining dataset. In fact, since our task parameters are continuous random variables, with probability 1, the downstream task is *not* included in a finite pretraining dataset.
> ___
> (4) “How is the definition of the two kinds of tasks relevant in the real-world scenario?”
>
> Our work is largely motivated by various observations made with real-world LLMs that appear to be conflicting. Here are the most prominent examples:
>
> Paper "Rethinking the Role of Demonstrations: What Makes In-Context Learning Work?" demonstrates that in-context learning with random labels sometimes outperforms in-context learning with true labels. How can this be explained?
>
> The GPT3 paper and the paper by Xie et al. both show that zero-shot performance is often higher than few-shot performance in various tasks. Notably, these findings are all based on ground truth labels. How can this be explained?
>
> From a more traditional supervised-learning perspective, these two observations seem counterintuitive. The general expectation is that the more samples provided, the better the performance should be. Indeed, most theoretical ICL papers assume this setting and prove that ICL performance should monotonically increase as the number of samples increases. We refer to this setting as "task learning".
>
> To explain these phenomena, we introduce "task retrieval", which happens when in-context examples are drawn from a different task than the target task (This can also be viewed as a form of concept shift, using distribution shift terminology). This might happen with label noise or when there is a subtle mismatch between the way labels are created for in-context samples versus evaluation samples. In this regime, we can rigorously show that when the number of samples is relatively small, the performance may not be monotonic.
> ___
> (5) “Empirically, we do not observe that having more examples hurt the performance.”
>
> In response to your insightful suggestion, we conducted experiments with GPT-4 and discovered that the U-shaped loss curve is observable in practice. The experiment arise described in “supplemental material”. Another related non-monotonic performance was reported in the original GPT-3 paper and reconfirmed by the paper by Xie et al. (please refer to Figure 7 therein).

---

### Official Review · Reviewer_ZoJM · 2023-10-29

**Soundness:** 3 good
**Presentation:** 3 good
**Contribution:** 3 good
**Rating:** 6
**Confidence:** 3

**Summary:**

NLP tasks like LLMs can be better with in-context examples, and in-context learning (ICL) in the community approaches explanation of the area. In this work, authors summarize two modes of ICL - in-context task learning and in-context task retrieval, which are able to learn a new task or retrieve related tasks during pre-training LMs. In the past, theorists somehow ignore the importance of pre-training distribution. Hence, authors propose a data generative approach based on the distribution to explain those two modes.

**Strengths:**

1. As a theoretical work, beyond upper bounds for risks of two modes, this work provides evidences based on numerical computations and conducts experiments with Transformer. That would be helpful for practitioners in the future work.
2. Visual illustration like Figure 2 helps to understand the prior distribution.

**Weaknesses:**

1. I observed that Lemma 1 has been used a lot in the manuscript and is lack of proof. Please at least provide high-level idea of the proofing before using it.
2. Texts and figures are overlapped in Page 8.
3. Similarly, for Theorem 3,4 and Lemma 5, please provide at least few sentences about proofing instead of pointing the appendix.

**Questions:**

N/A

---

> ### Author Response · Authors · 2023-11-21
>
> Thank you for your strong support of our work!
>
> We also would like to express our sincere gratitude to the reviewer for dedicating your valuable time to provide constructive feedback on our work. We added experiments to show the U-shaped loss curve actually exists in real-world LLM (GPT-4) and our framework can also be used to explain a flipped U-shaped loss curve researchers have observed in real-world LLM. We share the experiments in "supplementary material". We kindly request you to consider increasing your score and championing our paper if these new experiments significantly improve the quality and confidence of this paper.
> ___
> (1) “High level proof idea for lemma, theory”
>
> We will incorporate additional sentences to describe the high-level idea behind the proof.
> ___
> (2) “Texts and figures are overlapped in Page 8”
>
> Thanks! fixed.

---

### Official Review · Reviewer_KzaL · 2023-11-07

**Soundness:** 3 good
**Presentation:** 3 good
**Contribution:** 2 fair
**Rating:** 5
**Confidence:** 4

**Summary:**

The authors study in context learning (ICL) for task retrieval and task learning. The ICL is modeled using a mixture of linear Gaussian tasks. Using component shifting and component reweighting they derive closed form expressions for posterior distribution given k observatios. This is leveraged to derive risk bounds for the two tasks, retrieval and learning, under squared loss. The theoretical results are backed up by numerical, and experimental results.

**Strengths:**

- The authors highlight the task retrieval and task learning aspects of ICL.
- The authors provide risk bounds for both the setup under a mixture of linear Gaussian generative model.

**Weaknesses:**

- The generative model is simplistic as it is limited to linear Gaussian mixture. In contrast HMM based ICL is already studied in other works already mentioned in the paper. The authors should explain the importance of studying this model.

- The authors leave out some area of works that are related, e.g. meta-learning and retrieval augmented learning. A few recent examples of the latter are -  'A Statistical Perspective on Retrieval-Based Models' by Basu et al., 'Generalization and stability in in-context learning' Li  et al.

- The dependence of U shaped risk bound on component reweighting and shifting is not discussed properly. Furtheremore, U shape is also observed through the bias-variance tradeoff in optimization literature. The authors don't connect the U shaped mention here with the bias-variance tradeoff.

- The novelty in deriving the posterior distributions for the mixture of Gaussian distribution is unclear to me.

**Questions:**

Please look at the weakness section.

---

> ### Author Response · Authors · 2023-11-21
>
> We would like to express our gratitude to the reviewer for dedicating your valuable time to provide constructive feedback on our work. We hope our responses have addressed your concerns and answered your questions. If our responses have resolved your concerns, we kindly request you to consider increasing your score.
> ___
> (1) “Explain the importance of studying this model”
>
> We selected the linear regression mixture model, which is slightly less realistic (non-discrete) than the HMM model. Our choice was motivated by the model's analytical tractability, which provides a precise theoretical understanding of how task retrieval and task learning operate.
>
> Numerous theoretical studies adopt the same linear regression mixture model. Notable recent works include [1] through [5]. All theoretical work necessitates some form of simplified model to comprehend complex real-world phenomena. Although the linear regression mixture model deviates from the actual pretraining data model, the insights it offers may still predict real-world phenomena.
>
> [1] What can transformers learn in-context? a case study of simple function classes
>
> [2] What learning ¨ algorithm is in-context learning? investigations with linear models
>
> [3] Transformers as algorithms: Generalization and implicit model selection in in-context learning
>
> [4] Transformers learn in-context by gradient descent
>
> [5] Transformers can optimally learn regression mixture models
> ___
> (2) “Some area of works that are related are left”
>
> Thank you for sharing these papers! We will incorporate these papers into the related works section.
> ___
> (3) “The dependence of U shaped risk bound on component reweighting and shifting is not discussed properly.”
>
> We apologize if our current explanation is unclear. We will revise it to improve clarity.
>
> The U-shape is observed under conditions of low prior task variance, and we aim to retrieve a prior task center that is closest to the in-context task. When k is small, due to the low prior task variance, component reweighting takes effect faster than component shifting. Thus, when all prior centers are nearly unchanged, the task center closest to the in-context task receives a high weight close to 1, thereby reducing task retrieval loss. However, as k continues to increase and becomes large, component shifting causes all posterior task centers to shift towards the in-context task. Therefore, when the in-context task differs from the target task, this results in an increase in task retrieval loss.
> ___
> (4) “Connect the U shaped with the bias-variance tradeoff.”
>
> Our U-shaped curve can **not** be directly explained by the bias-variance tradeoff. This is because the bias-variance tradeoff is for increasing model complexity, while our curve is for increasing the number of samples. Under the standard supervised learning setting, our U-shaped curve is not observed. Instead, our task retrieval setting is closely related to the distribution shift setting, particularly the concept shift.
> ___
> (5) “The novelty in deriving the posterior distributions for the mixture of Gaussian distribution is unclear to me”
>
> We have **not** claimed novelty in the calculation of the posterior distribution (although we believe it forms part of our contribution, given the extensive calculations required). Instead, our paper presents two major contributions in terms of novelty.
>
> Firstly, we are the pioneering work that formally defines and analyzes the two regimes of in-context learning: task learning and retrieval. This can assist future researchers in reconciling seemingly conflicting observations made with in-context learning in the past, as they can be interpreted as non-conflicting results within the two distinct regimes.
>
> Secondly, we are the first to precisely predict two non-monotonic performance patterns of ICL in real-world LLMs as the number of samples increases: the U-shaped curve we presented in the original submission, and the flipped one we presented in the rebuttal. The U-shaped curve was first predicted by our theory and then confirmed by our additional experiment with GPT4. The flipped curve was initially presented in the original GPT3 paper (and later was re-confirmed by Xie et al.’s work too!) and explained via experiment under our framework. The experiments are provided in “supplementary material”
>
> These two contributions lead us to strongly believe that our paper offers substantial contributions to the research community, particularly those focused on the theoretical analysis of ICL, and to some extent, practical LLM research.

---

### Official Review · Reviewer_4TUM · 2023-11-07

**Soundness:** 3 good
**Presentation:** 3 good
**Contribution:** 3 good
**Rating:** 6
**Confidence:** 2

**Summary:**

This paper investigates in-context learning for task retrieval and task learning from the lens of generative models for pre-training data and in-context samples. A Gaussian mixture model is proposed for the pre-training data generation process. The paper first demonstrates that in-context prediction with a Bayes-optimal next-token predictor corresponds to the posterior mean of the label given the in-context samples. The paper establishes upper bounds for both task retrieval and learning risks, revealing a quadratic decrease in the risk bound of task learning and a U-shaped pattern for the risk bound of task retrieval. The findings are validated through numerical simulations, and with experiments using Transformers.

**Strengths:**

- The theoretical analysis using data generative models using a Gaussian mixture model is novel, and presents a new approach to investigating in-context learning.
- The theoretical results are valid, and are backed by simulations with the data generated from a Gaussian mixture model with Transformers.
- The U-shaped pattern for the risk bound of task retrieval is shown for the first time, which has not been observed in previous works.

**Weaknesses:**

- The claim regarding the similarity to real-world settings is not quite accurate, as the setting studied in the paper is quite different from in-context learning in NLP tasks with practical LLMs.
- The assumptions in Section 3.3 are restrictive, and the scope of the theoretical analysis is limited to a very specific setup based on a particular generative model.

**Questions:**

- "A highly expressive F can be viewed as K separate models F0, . . . , FK−1, where Fk takes exactly 2k + 1 tokens as input. Thus, pretraining can be decomposed into K separate optimization problems." Can the authors provide more explanation regarding this statement?
- Can the authors provide more details about the Transformer? The code seems to suggest the architecture used is GPT-2, and NanoGPT, but these details should be provided in the paper.

Suggestions:
- The experimental results provided in Appendix A with Figures 6,7, and 8 are central results for the paper, and are not quite accessible for readers. It would be beneficial to add more experiments in the main paper, and move some of the analysis to the supplementary material.
- There are typographical issues and formatting issues (e.g. text overlaps in Figure 5) that should be fixed.

---

> ### Author Response · Authors · 2023-11-21
>
> Thank you for your strong support of our work!
>
> We also would like to express our sincere gratitude to the reviewer for dedicating your valuable time to provide constructive feedback on our work. We hope our responses have addressed your concerns and answered your questions. If our responses have resolved your concerns, we kindly request you to consider increasing your score and championing our paper.
> ___
> (1) “the similarity to real-world settings is not quite accurate, as the setting studied in the paper is quite different from in-context learning in NLP tasks” &  “assumptions in Section 3.3 are restrictive, and the scope of the theoretical analysis is limited to a very specific setup”
>
> We deliberately selected the linear regression mixture model, which, as the reviewer correctly pointed out, is slightly less realistic (non-discrete) than the HMM model. Our choice was motivated by the model's analytical tractability, which provides a precise theoretical understanding of how task retrieval and task learning operate. Numerous theoretical studies adopt the same linear regression mixture model. Notable recent works include [1] through [5].
>
> All theoretical work necessitates some form of simplified model and strong assumptions to comprehend complex real-world phenomena. Although the linear regression mixture model and our strong assumptions may not look very realistic, the analysis we performed with this model still predict real-world phenomena.
>
> [1] What can transformers learn in-context? a case study of simple function classes
> [2] What learning ¨ algorithm is in-context learning? investigations with linear models
> [3] Transformers as algorithms: Generalization and implicit model selection in in-context learning
> [4] Transformers learn in-context by gradient descent
> [5] Transformers can optimally learn regression mixture models
> ___
> (2) “A highly expressive F can be viewed as K separate models F0, . . . , FK−1, where Fk takes exactly 2k + 1 tokens as input. Can the authors provide more explanation regarding this statement?"
>
> F is a function that takes variable-length input. Thus, F can be fully characterized by specifying F's behavior for each input length. That is,
>
> F = F_0 * 1(# of in-context examples = 0) + F_1 * 1(# of in-context examples = 1) + …
>
> Now, consider a sequence of tokens [x1, y1, x2, y2]. Pretraining can be viewed as
>
> min_F L(F([x1]),y1) + L(F([x1,y1,x2]),y2).
>
> By the above input-length decomposition, this is equivalent to
>
> min_{F_0, F_1} L(F_0([x1]),y1) + L(F_1([x1,y1,x2]),y2).
>
> Thus, this becomes two separate optimization problems. Now, if F is highly expressive, then F_0 and F_1 are also highly expressive. Thus, each F_i can become the Bayes-optimal solution.
> ___
> (3) “The code seems to suggest the architecture used is GPT-2, and NanoGPT, but these details should be provided in the paper.”
>
> The code is constructed on the GPT2Model from the transformers package supported by Hugging Face. The number of attention heads is set to 8, the depth is set to 10, and the embedding dimension is set to 1024. We use the Adam optimizer with a learning rate and weight decay both set to 0.00001. The batch size is set to 256. We train the model for 3 epochs, with each epoch consisting of 10,000 steps.
> ___
> (4) “Figures 6,7, and 8 are central results for the paper, and are not quite accessible for readers. It would be beneficial to add more experiments in the main paper,” & “text overlaps in Figure 5”
>
> Thank you for your insightful suggestions! We will incorporate these changes into our revision.

---

### Official Review · Reviewer_8Pgy · 2023-11-09

**Soundness:** 4 excellent
**Presentation:** 2 fair
**Contribution:** 3 good
**Rating:** 6
**Confidence:** 3

**Summary:**

The paper presents a theoretical analysis of in-context learning. This work separates in-context learning into task retrieval and task learning modes. The authors formalise in-context learning as Bayesian inference over pre-trained tasks using proposed generative models. Closed-form posterior distribution and its component re-weighting and shifting mechanisms are presented clearly. They prove task learning risk decreases quadratically as number of examples increases and task retrieval risk follows a U-shaped curve, decreasing then increasing with more examples. They validate analysis theoretically and via simulations on Transformer models.
In summary, the paper provides a formal understanding of in-context learning grounded in Bayesian principles. The analysis reveals unique insights into the distinct behaviors of task retrieval versus learning based on modification of the posterior distribution over pre-trained tasks.

**Strengths:**

Originality:
Principled Bayesian framework unifying task retrieval and learning modes of in-context learning.
Interesting generative model provides formal grounding for the analysis.
Derives unique insights such as U-shaped bound revealing limitations of task retrieval.

Quality:
Good mathematical analysis and detailed proofs.
Interesting experiments supporting the theory.
Code provided for reproducibility.

Clarity:
Clearly explains concepts like component re-weighting and shifting.
Delineates assumptions underlying the analysis.

Significance:
Helps advances understanding of in-context learning in large language models.
Formal analysis provides basis for improving real-world in-context performance.
Insights like U-shaped bound have significant implications for practical model.

**Weaknesses:**

The generative modeling makes strong simplifying assumptions like Gaussian distributions that limits applicability to complex real-world textual data. Expanding the analysis to more realistic data distributions will strengthen it. Or perhaps even showing that any of their conclusions hold in LLMs, for example does the u shaped phenomena occur in LLMs?

The empirical validation relies heavily on synthetic data simulated from the assumed generative process. Evaluating on real-world NLP tasks would better assess wider applicability.

The criteria for determining the number of examples k for optimal retrieval remains unspecified. Providing more precise theoretical or empirical guidance could improve utility.

Risk is evaluated using squared loss. Evaluating other loss functions (Cross entropy) could expand usefulness across different domains.
As task retrieval becomes more mainstream in LLMs, societal impacts of failures in retrieved tasks can be added. Characterizing the potential downsides and suggesting caution.

**Questions:**

Please see the weaknesses

---

> ### Author Response · Authors · 2023-11-21
>
> Thank you for your strong support of our work!
>
> We also would like to express our sincere gratitude to the reviewer for dedicating your valuable time to provide constructive feedback on our work. We hope our responses have addressed your concerns and answered your questions. If there are any remaining questions, please do not hesitate to let us know. If our responses have resolved your concerns, we kindly request you to consider increasing your score and championing our paper.
> ___
> (1) “The generative modeling makes strong simplifying assumptions like Gaussian distributions that limits applicability to complex real-world textual data. Expanding the analysis to more realistic data distributions will strengthen it. Or perhaps even showing that any of their conclusions hold in LLMs, for example does the u shaped phenomena occur in LLMs?”
>
> In response to your valuable suggestion, we conducted additional experiments to demonstrate that the U-shaped curve can indeed be observed with a real-world LLM (GPT-4). Please refer to Section 1 of the supplemental file for details.
>
> Moreover, we demonstrate that under a slightly different setting, our task learning analysis can explain another intriguing phenomenon, which we term the 'flipped U curve' – a pattern where the loss increases and then decreases as the number of samples increases. This effectively explains the phenomenon reported in the original GPT-3 paper as well as in subsequent research. Please refer to Section 2 of the supplemental file for details.
>
> Thus, even if our linear regression mixture model deviates from the actual pretraining data model, the insights it offers can predict real-world phenomena!
> ___
> (2) ”optimal retrieval remains unspecified”
>
> The question essentially requires finding the minimizer of $f(k) = a \exp(-bk) + c \exp(-dk^{1/2}) + e k^2$, which is equivalent to solving $f’(k)=0$. The author believes that there is no closed-form solution for this.
> ___
> (3) “Evaluating other loss functions (Cross entropy)”
>
> Thank you for the excellent suggestion. We are currently exploring it; however, it's challenging to find a suitable prior for classification that is akin to the Gaussian mixture used in the linear regression setting. The Gaussian mixture provides flexibility for mathematical calculation and exploration of the prior's properties. We have added evaluations on GPT-4 which is trained with cross-entropy to reproduce the U-shaped loss curve, in the Section 1 of the supplemental file.

---

### Author Response · Authors · 2023-11-20
**Author Responsel to All Reviewers -- Two Primary Concerns**

We thank the reviewers for providing valuable suggestions that helped us improve our paper.

We are particularly encouraged that the reviewers have found the strengths of the paper:
1. Significance: advancing understanding of in-context learning in large language models, helpful to the future. (8Pgy, ZoJM, v8Fu)
2. Originality-Framework: a Bayesian framework unifying task retrieval and learning of in-context learning, a new approach to investigate in-context learning. (8Pgy, 4TUM)
3. Originality-Analysis: formal definition of task retrieval and learning, and unique insights such as U-shaped bound revealing limitations of task retrieval. (8Pgy, 4TUM, KzaL, v8Fu)
4. Quality-Theory: good mathematical analysis and detailed proofs, theoretical results are valid (8Pgy, 4TUM).
5. Quality-Expriment: interesting experiments supporting the theory. (8Pgy, 4TUM, ZoJM29).

The reviewers have raised two primary concerns: (1) the practical observability of the U-shaped curve our task retrieval analysis predicts, and (2) the practical relevance of linear regression with L2 loss. We address these concerns as follows.

(1) The U-shaped curve is observable in practice.

***We have conducted additional experiments with GPT-4, demonstrating the U-shaped loss curve of task retrieval !!!*** Moreover, under slightly modified conditions, ***our task retrieval analysis can be utilized to explain a similar phenomenon (which we term the 'flipped U-shaped loss curve')*** reported in existing literature, such as the original GPT3 paper. Therefore, we assert that our theoretical analysis holds significant practical relevance, as it aids in better understanding the behaviors of real-world LLMs. The two experiments are described in the PDF of "Supplementary Material".

(2) Linear regression mixture models are the current standard for theoretical understanding of in-context learning.

Numerous theoretical studies adopt the same linear regression mixture model. Notable recent works include [1] through [5]. All theoretical work necessitates some form of simplified model to comprehend complex real-world phenomena. Although the linear regression mixture model deviates considerably from the actual pretraining data model, the insights it offers may still predict real-world phenomena, as we have previously mentioned.

 [1] What can transformers learn in-context? a case study of simple function classes

 [2] What learning ¨ algorithm is in-context learning? investigations with linear models

 [3] Transformers as algorithms: Generalization and implicit model selection in in-context learning

 [4] Transformers learn in-context by gradient descent

 [5] Transformers can optimally learn regression mixture models

===

Below, we offer a detailed response to each reviewer, addressing your respective concerns. We believe we have adequately responded to the issues raised. If any aspect remains unclear, we are more than willing to provide further clarification. We kindly request the reviewers to consider revising your scores upward if you find our responses satisfactory. We appreciate your time and effort in reviewing our work and look forward to your feedback.

---

### Meta-Review · Area_Chair_Lozq · 2023-12-30

**Metareview:**

The paper attempts to improve our understanding of in context learning (ICL) in pretrained models. In this regards, authors propose a generative models for pretraining data ( a mixture of linear Gaussian tasks) to explain two phenomenons in ICL: task retrieval and learning. Using component shifting and component reweighting they derive closed form expressions for posterior distribution given k observations. This is leveraged to derive risk bounds for the two tasks, retrieval and learning, under squared loss. Finally experiments are conducted to demonstrate correlation to theoretical findings. The reviewers had a lukewarm to the paper and raised concerns about more simplistic model than prior work, connection of the two setting to real world, limited novelty in theoretical result proofs involving deriving posteriors for Gaussians, missing details and other presentation issues. We thank the authors for providing detailed responses and conducting additional experiments based on the reviews to make the paper better. Despite this, none of the reviewers are willing to champion for the paper. The u-shaped in LLMs can be attributed to many things like bias-variance trade-off [1] or sequence length issues [2]. Thus more discussion is needed to connect the proposed theoretical framework to practical setting. Thus, unfortunately the paper cannot be accepted to ICLR in its current form.

[1] Transformers as Algorithms: Generalization and Stability in In-context Learning
[2] Lost in the Middle: How Language Models Use Long Contexts

**Justification For Why Not Higher Score:**

- Considering more simplified setting than prior art.
- Limited novelty in theoretical results, could enhance by considering any one of: more loss functions, being closer to real settings, deriving suggestions for practitioners like optimal number of in-context samples, tighter risk bounds, etc.
- Limited connection to real practical setting: more discussion/experiments are needed on how proposed bayesian framework connects to observed u-shape in LLMs and its not due to numerous other factors.

**Justification For Why Not Lower Score:**

N/A

---

### Decision · Program_Chairs · 2024-01-16

Reject